# T2N as a new tool for robust electrophysiological modeling demonstrated for mature and adult-born dentate granule cells

Marcel Beining[1,2,3,4]*, Lucas Alberto Mongiat[5], Stephan Wolfgang Schwarzacher[3], Hermann Cuntz[1,2]†*, Peter Jedlicka[3]†

[1]Ernst Strüngmann Institute (ESI) for Neuroscience in Cooperation with Max Planck Society, Frankfurt, Germany; [2]Frankfurt Institute for Advanced Studies, Frankfurt, Germany; [3]Institute of Clinical Neuroanatomy, Neuroscience Center, Goethe University, Frankfurt, Germany; [4]Faculty of Biosciences, Goethe University, Frankfurt, Germany; [5]Instituto de Investigación en Biodiversidad y Medioambiente, Universidad Nacional del Comahue-CONICET, San Carlos de Bariloche, Argentina

**Abstract** Compartmental models are the theoretical tool of choice for understanding single neuron computations. However, many models are incomplete, built ad hoc and require tuning for each novel condition rendering them of limited usability. Here, we present T2N, a powerful interface to control NEURON with Matlab and TREES toolbox, which supports generating models stable over a broad range of reconstructed and synthetic morphologies. We illustrate this for a novel, highly detailed active model of dentate granule cells (GCs) replicating a wide palette of experiments from various labs. By implementing known differences in ion channel composition and morphology, our model reproduces data from mouse or rat, mature or adult-born GCs as well as pharmacological interventions and epileptic conditions. This work sets a new benchmark for detailed compartmental modeling. T2N is suitable for creating robust models useful for large-scale networks that could lead to novel predictions. We discuss possible T2N application in degeneracy studies.

DOI: https://doi.org/10.7554/eLife.26517.001

*For correspondence:
beining@fias.uni-frankfurt.de (MB);
cuntz@fias.uni-frankfurt.de (HC)

†These authors contributed equally to this work

**Competing interests:** The authors declare that no competing interests exist.

## Introduction

Traditionally, neurons have long been interpreted as passive integrators of input signals that fire action potentials when a threshold is reached (*Knight, 1972*). This paradigm has meanwhile changed as the output of neurons was shown to depend on many intrinsic cellular mechanisms (e.g. voltage-gated channels, dendritic architecture, synaptic plasticity, active dendrites, axon initial segment) indicating that single neuron computation is rather complex (*Softky and Koch, 1993*; *Brunel et al., 2014*; *Volgushev, 2016*). Consequently, detailed compartmental models have found their way into the set of tools for neuroscientists to understand, test, or predict mechanisms underlying neuronal function (*de Schutter, 1994*). Compartmental models are easy to manipulate and models of cellular mechanisms such as ion channels and synapses can be incorporated in arbitrary detail. Many recent models include reconstructed morphologies, which are often available online through specialized databases. As these models seem to become more and more realistic, the hope arises that one will soon be able to simulate entire circuits or even the brain itself simply by including more and more details (*Markram, 2006*, *2012*; *Markram et al., 2015*; *Hawrylycz et al., 2016*). However, most published models behave poorly when used outside of the scope for which they were created. There are

several reasons for this, such as bona fide adaptation of other models without knowing their limitations, too few target constraints because of using a low number of target parameters, no scientific rationale for setting the parameters, missing axon initial segments and a lack of data from pharmacology to fit and test the model's individual components (*Almog and Korngreen, 2016*). Furthermore, model neurons often contain a mix of constraints from very different experimental conditions and animals, combined in sometimes inconsistent ways.

To make addressing these issues easier and to help build consistent and robust models, we developed *T2N*, a new software interface to control compartmental modeling package *NEURON* (*Carnevale and Hines, 2006*) using *Matlab* and the *TREES toolbox* (*Cuntz et al., 2010*; *2011*). *T2N* enables to design detailed electrophysiology models on the basis not of single morphologies but rather on large datasets of reconstructed and synthetic morphologies. Such datasets have become more easily available from online databases such as *NeuroMorpho.Org* (*Ascoli et al., 2007*) and morphological modeling studies (*Cuntz et al., 2008*, *2010*; *Koene et al., 2009*; *Beining et al., 2017*). In addition, ion channel models from newly available databases (*Ranjan et al., 2011*; *Podlaski et al., 2016*; *McDougal et al., 2017*) can be directly incorporated into *T2N* models to consider the many new insights from recent studies on neuronal electrophysiology and protein expression. Through this tool, we aimed here to create novel compartmental models that (1) are solely based on ion channel isoforms known to exist in the neurons of interest, (2) are robust over many different real and synthetic dendritic morphologies and (3) reproduce experimental results from various studies. Although *T2N* can be used to create robust models for any neuron type, in this work we focused on hippocampal dentate granule cells, which play a crucial role in learning and memory and exhibit the unique feature that they integrate into the adult hippocampal network as newborn neurons throughout life.

## Results

### Development of *T2N*

Our novel modeling framework for creating compartmental models with realistic biophysical properties in multiple morphologies is depicted in *Figure 1*. In this framework, our *T2N* package interfaces between *TREES toolbox* and *NEURON*. Since the *TREES toolbox* (*Cuntz et al., 2010*, *2011*) is a recently established versatile tool for the analysis and modeling of 3D morphologies of dendrites, its coupling to *NEURON* (*Carnevale and Hines, 2006*) opens many new possibilities: (1) Biophysical mechanisms can be inserted not only into reconstructed but also into synthetic morphologies (e.g. created with *TREES toolbox*; *Cuntz et al., 2010*; *Schneider et al., 2014*; *Beining et al., 2017*; *Platschek et al., 2016*), which is important for the creation of a large set of realistic compartmental models capturing neuron-to-neuron variability of dendritic trees. For the insertion of biophysical mechanisms, *T2N* makes maximal use of region specifications that are available in *TREES toolbox*. Handling of section lists in *NEURON* is not necessary. (2) For a given set of biophysical mechanisms, *T2N* enables an easy and efficient switch among diverse morphologies from different species including any number of morphologies downloaded from for example, databases of reconstructed morphologies such as *NeuroMorpho.Org*. This facilitates the generalization of predictions from one dendritic tree type to other types and supports the search for universal principles valid for all dendritic morphologies. (3) *T2N* provides a simple and clear set up and controls *NEURON* compartmental models with a direct subsequent analysis with *Matlab* and the *TREES toolbox* allowing for any morphology related analyses. This is a unique feature of *T2N*. By generating stereotyped *NEURON* scripts, *T2N* enhances the readability and compatibility of the code. (4) Multiple simulations are run automatically in parallel on different cores without the need of rewriting the *NEURON* code. When activated, T2N also supports parallel NEURON (*Migliore et al., 2006*; *Hines and Carnevale, 2008*) and distributes cells automatically on a given amount of cores, thereby increasing the speed of large-scale network simulations drastically. (5) By connecting NEURON to Matlab, T2N makes it easier to plot and visualize the results of simulations and their analysis. In summary, by coupling morphological software and compartmental simulations, *T2N* provides users with powerful tools for an in-depth analysis of structure-function relationships in neurons. In the following, we show on the

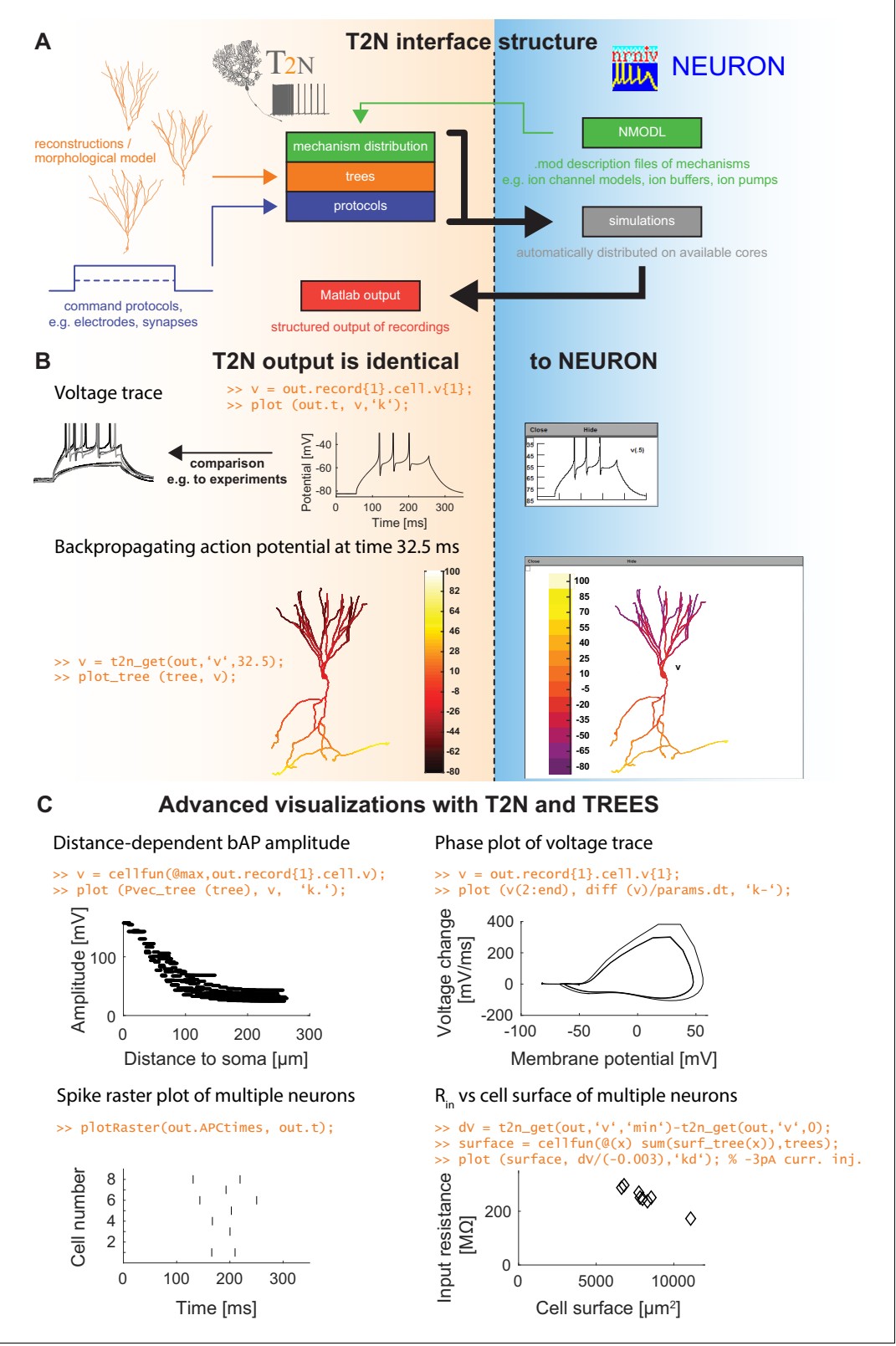

**Figure 1.** TREES-to-NEURON (T2N) interface linking compartmental modeling environment NEURON with morphology modeling and analysis tools of Matlab and TREES toolbox. *T2N* enables fast and simple incorporation of many diverse morphologies in compartmental simulations facilitating the search for morphologically robust biophysical models. (**A**) Illustration of *T2N* workflow. *T2N* allows for setting up a full compartmental model in *Matlab* by importing reconstructed or synthetic morphologies (orange; e.g. from NeuroMorpho.org) and by distributing subcellular channel mechanisms

*Figure 1 continued on next page*

*Figure 1 continued*

(green; mod files generated with *NEURON*'s *NMODL* or obtained from databases such as IonChannelGenealogy or Channelpedia). In addition, *T2N* enables setting up full simulation control by attaching stimulation and recording electrodes and specifying simulation conditions (e.g. stimulation protocols; blue). *T2N* then automatically produces stereotyped *NEURON* hoc code, initializes and runs simulations and returns recorded data in a structured output format (red). (**B**) A comparison of two example results in *NEURON* and *T2N* validates *T2N* simulation output. The orange script shows sample code for visualizing the output. Upper row: somatic voltage trace during a current injection. Lower row: membrane voltage at each dendrite location at a single time point. (**C**) Examples of using *T2N* for a simple and fast analysis and visualization of simulation results. (Code for creating the panels is shown in orange; code for the specific labels is omitted).

DOI: https://doi.org/10.7554/eLife.26517.002

example of the dentate GC (see also Appendix 2) how to build a robust compartmental model using *T2N*.

## *T2N* facilitates creation of compartmental models with detailed channel composition

*T2N* simplifies distributing dendritic, somatic and axonal ion channels in layer- or branch-specific manner. We illustrate this for an experimentally well constrained set of GC ion channels that we identified and modeled based on extensive literature search (see Appendix 2 for more details). Importantly, we included only those channel isoforms, which were described for GCs. Moreover, we carefully implemented compartment-specific distributions of the channels according to immunohistochemical labeling and light or electron microscopy as well as electrophysiological data (*Table 1* and *Figure 2A*). Available tools of NEURON have limitations with regard to specifying layer- or branch-specific biophysical properties in a large dataset of morphologies. T2N makes it easy to insert ion channels in selected regions because it maps the nodes, branches and regions of the TREES toolbox (*Cuntz et al., 2010*, *2011*) onto sections and segments in NEURON (see Tutorial 1 in Appendix 1). Of note, for cell types or compartments where channel expression data is not available, T2N can be used in a more exploratory manner, for example such as mapping model responses on single cell current sweep data.

## *T2N* facilitates use of synthetic morphologies based on optimal wiring principles

T2N allows users to investigate electrophysiological properties of morphological models created with TREES toolbox. The underlying morphological modeling algorithm (minimum spanning tree (MST) algorithm) finds optimal weighted solution for connecting dendritic target points considering a fundamental trade-off between cable length and conduction times (*Cuntz et al., 2010*, *2011*; *Cuntz et al., 2012*). Realistic morphological models of dendrites, created by the MST algorithm, can be easily imported into NEURON via T2N (see Tutorial 2 in Appendix 1). In this way, users can test whether their compartmental models are stable over a large set of dendritic morphologies. To create a set of GC synthetic morphologies, we took our previously published morphological model of mature rat GCs (*Beining et al., 2017*) and adapted it for mature mouse GCs (*Figure 2B*). The resulting synthetic dendritic trees were morphologically comparable to the reconstructed trees of mouse GCs from *Schmidt-Hieber et al. (2007)* (*Figure 2C*). These synthetic neurons were introduced into the compartmental model as a further validation of the fitted passive and active properties (see below).

## T2N allows for an easy switch between real and synthetic morphologies and facilitates comparison of simulation results with experimental data

To illustrate the flexibility and versatility of T2N, we used it to fit the GC model equipped with ion channels from *Figure 2A* (see also *Table 1*) to experimental data. The channels were inserted into reconstructed (*Figure 2A*) or synthetic (*Figure 2B*) morphologies. Our goal was to replicate electrophysiological recordings from mature GCs including voltage clamp and current clamp experiments. For this purpose we used raw experimental traces from published data (*Mongiat et al., 2009*). Tutorials 3–5 (Appendix 1) explain how to use T2N to define and run simulations, especially how to generate I-V and spiking frequency vs. current (f-I) curves. *Figure 3* (middle column) and *Table 2* show the results of such simulations in morphologies of mature GCs (for details see Appendix 2) indicating

**Table 1.** Summary of all ion channel models and densities implemented in the mouse mature GC model.

Categorial values of the ion channel expression profiles: 0 = not existent or very weak, 1 = weak, 2 = moderate, 3 = strong. Conductances [$mS/cm^2$] for each ion channel used in the model are given in the gray fields.

| Name | Soma | Axon | AIS | GCL | IML | MML | OML | Reference | Ion channel model |
|---|---|---|---|---|---|---|---|---|---|
| $Na_v$ 1.1 | 3 | 0 | 0 | 0 | 0 | 0 | 0 | (*Westenbroek et al., 1989*; *Schmidt-Hieber and Bischofberger, 2010*) | 8-state model from (*Schmidt-Hieber and Bischofberger, 2010*). Inact. modified according to (*Rush et al., 2005*; *Schmidt-Hieber and Bischofberger, 2010*) (see text) |
| $Na_v$ 1.2 | 0 | 3 | 3 | 0 | 0 | 0 | 0 | | |
| $Na_v$ 1.6 | 0 | 1 | 3 | 0 | 0 | 0 | 0 | (*Kress et al., 2010*; *Schmidt-Hieber and Bischofberger, 2010*) | |
| | 88.128 | 88.1280 | 518.400 | - | - | - | - | | |
| K2Ps (passive) | 3 | 1 | 1 | 2 | 2 | 2 | 2 | (*Lesage et al., 1997*; *Hervieu et al., 2001*; *Talley et al., 2001*; *Gabriel et al., 2002*; *Aller and Wisden, 2008*; *Yarishkin et al., 2014*) | |
| | 0.014 | 0.007 | 0.007 | 0.014 | 0.014 | 0.014 | 0.014 | | |
| Kir 2.x | 3 | 1 | 1 | 2 | 2 | 2 | 2 | (*Karschin et al., 1996*; *Miyashita and Kubo, 1997*; *Stonehouse et al., 1999*; *Prüss et al., 2003*) | 6-state model, modification see Appendix 2. |
| | 0.1416 | 0.0674 | 0.0674 | 0.1416 | 0.1416 | 0.1416 | 0.1416 | | |
| HCN1-3 | 0 | 0 | 0 | 0 | 2 | 2 | 2 | (*Notomi and Shigemoto, 2004*) | 2-state model, from (*Stegen et al., 2012*); activation −10 mV, added cAMP-sens. and slow comp. of act. |
| | - | - | - | - | 0.004 | 0.004 | 0.004 | | |
| $K_v$ 1.1 | 0 | 3 | 3 | 0 | 0 | 0 | 0 | (*Rhodes et al., 1997*; *Grosse et al., 2000*; *Monaghan et al., 2001*) | nh model from (*Christie et al., 1989*) |
| | - | 0.25 | 0.25 | - | - | - | - | | |
| $K_v$ 1.4 | 0 | 3 | 3 | 0 | 0 | 0 | 0 | (*Rhodes et al., 1997*; *Cooper et al., 1998*; *Grosse et al., 2000*; *Monaghan et al., 2001*) | $n^4h$ model from (*Wissmann et al., 2003*) |
| | - | 1 | 1 | - | - | - | - | | |
| $K_v$ 2.1 | 3 | 0 | 0 | 0 | 0 | 0 | 0 | (*Rhodes et al., 1997*; *Murakoshi and Trimmer, 1999*) | mh model, fitted using (*VanDongen et al., 1990*; *Kramer et al., 1998*; *Kerschensteiner and Stocker, 1999*; *McCrossan et al., 2003*; *Gordon et al., 2006*) |
| | 7.09 | - | - | - | - | - | - | | |
| $K_v$ 3.3/ 3.4 | 0 | 2 | 3 | 0 | 0 | 0 | 0 | (*Weiser et al., 1994*; *Chang et al., 2007*) | mh model, fitted using (*Rudy et al., 1991*; *Schröter et al., 1991*; *Rettig et al., 1992*; *Miera et al., 1992*; *Riazanski et al., 2001*; *Desai et al., 2008*) |
| | - | 7.6562 | 30.7813 | - | - | - | - | | |
| $K_v$ 4.2/3 +KChIP/ DPP6 | 0 | 0 | 0 | 1 | 2 | 3 | 3 | (*Rhodes et al., 2004*; *Zagha et al., 2005*; *Menegola and Trimmer, 2006*) | 13-state model from (*Barghaan et al., 2008*); activation −20 mV according to (*Barghaan et al., 2008*; Figure S1A) and (*Jerng et al., 1999*; *An et al., 2000*; *Bähring et al., 2001*; *Patel et al., 2004*; *Jerng et al., 2005*; *Rüschenschmidt et al., 2006*; *Kaulin et al., 2008*; *Kim et al., 2008*) |
| | - | - | - | 2.1750 | 4.35 | 4.35 | 4.35 | | |
| $K_v$ 7.2/3 (KCNQ2 and 3) | 0 | 2 | 3 | 0 | 0 | 0 | 0 | (*Cooper et al., 2001*; *Klinger et al., 2011*; *Martinello et al., 2015*) | mh model from (*Mateos-Aparicio et al., 2014*) (η = 0.5, see Tab. S1 in that publication) |
| | - | 1.3400 | 6.7000 | - | - | - | - | | |
| $Ca_v$ 1.2 (L-type) | 3 | 0 | 1 | 1 | 2 | 2 | 2 | (*Tippens et al., 2008*; *Leitch et al., 2009*) | $mh_1h_2$ model from GENESIS (*Evans et al., 2013*), added $Ca^{2+}$-dep. inactivation (h2) |
| | 0.0200 | - | 0.0100 | 0.0100 | 0.0400 | 0.0400 | 0.0400 | | |
| $Ca_v$ 1.3 | 3 | 1 | 2 | 1 | 2 | 2 | 2 | (*Tippens et al., 2008*; *Leitch et al., 2009*) | $mh_1h_2$ model from GENESIS (*Evans et al., 2013*), added $Ca^{2+}$-dep. inactivation, modified after (*Bell et al., 2001*; *Koschak et al., 2001*) |
| | 0.0160 | 0.0040 | 0.0080 | 0.0040 | 0.0080 | 0.0080 | 0.0080 | | |
| $Ca_v$ 2.1/2 (N-/P/Q- type) | 3 | 2 | 2 | 1 | 1 | 1 | 1 | (*Day et al., 1996*; *Chung et al., 2001*; *Li et al., 2007*; *Xu et al., 2007*; *2010*) | $m^2h$ model from (*Fox et al., 1987*); set inact. time constant to 100 ms according to (*Fox et al., 1987*; *Huang et al., 2010*) |
| | 0.3000 | 0.0500 | 0.0500 | 0.0500 | 0.0500 | 0.0500 | 0.0500 | | |
| $Ca_v$ 3.2 (T-type) | 3 | 1 | 1 | 2 | 2 | 2 | 2 | (*Craig et al., 1999*; *McKay et al., 2006*; *Martinello et al., 2015*) | 8-state model from (*Burgess et al., 2002*) |
| | 0.0220 | 0.0080 | 0.0080 | 0.0220 | 0.0220 | 0.0220 | 0.0220 | | |

*Table 1 continued on next page*

*Table 1 continued*

| Name | Soma | Axon | AIS | GCL | IML | MML | OML | Reference | Ion channel model |
|---|---|---|---|---|---|---|---|---|---|
| BK (slo1) α αβ | 2 | 3 | 3 | 0 | 0 | 0 | 0 | (*Knaus et al., 1996*; *Misonou et al., 2006*; *Sailer et al., 2006*; *Kaufmann et al., 2010*) | Model from (*Jaffe et al., 2011*); modification see Appendix 2 |
| | 15.6 3.9 | 62.4 15.6 | 62.4 15.6 | - | - | - | - | | |
| SK2 | 0 | 2 | 3 | 0 | 1 | 1 | 1 | (*Obermair et al., 2003*; *Sailer et al., 2004*; *Maciaszek et al., 2012*; *Ballesteros-Merino et al., 2014*) | Model from (*Solinas et al., 2007*) based on (*Hirschberg et al., 1998*; *1999*) |
| | 0.001 | 0.013 | 0.083 | 0.002 | 0.004 | 0.004 | 0.004 | | |

DOI: https://doi.org/10.7554/eLife.26517.004

that our model is able to reproduce passive properties, steady state currents as well as AP shape and spiking behavior observed in patch-clamp experiments (*Figure 3*, left column; *Mongiat et al., 2009*). Importantly, the GC model remained stable and continued to generate realistic electrophysiological traces even after replacing one set of GC morphologies (reconstructed dendrites) by a different set of morphologies (synthetic dendrites; *Figure 3*, right column; see also Appendix 2) while keeping all biophysical mechanisms unaltered. Interestingly, the insertion of different morphologies introduced certain amount of variability in electrophysiological behavior (see Appendix 2 for details). Thus, some of the variance observed in electrophysiological recordings might be explained by the morphological variability of GCs.

## T2N facilitates the use of real or synthetic morphologies from different species

To test whether the ion channels from *Figure 2A* can account for mature rat GC electrophysiology, we used T2N to simulate rat I-V and f-I curves simply by replacing the mouse with rat GC morphologies. For this we used reconstructed and synthetic mature rat GC morphologies (*Figure 4A*), which we have recently published (*Beining et al., 2017*). Interestingly, increasing the Kir conductance (see Appendix 2 for details) was sufficient to replicate mature rat GC I-V recordings (*Pourbadie et al., 2015*) using rat morphologies (*Figure 4B*). Also, after the adjustment of the Kir conductance, active channel properties and densities from mouse GCs (*Table 1*) reproduced the spiking behavior of rat GCs (*Figure 4C–D*). This result indicates that both rodent species might share a similar GC ionic channel density pattern, conferring to these neurons their electrophysiological identity. Similarly to the mouse GC model, we could interchange real and synthetic rat morphologies without affecting the spiking behavior (*Figure 4B–D*, left vs. right). This shows an inherent robustness of our active model and validates its usefulness for large-scale network simulations of the rat DG.

## T2N simplifies analysis of dendritic voltage propagation and Ca$^{2+}$ signaling across different morphologies and species

T2N supports simulations, efficient analysis and visualization of distance-dependent changes of dendritic voltage and Ca$^{2+}$ spread. In Tutorial 6 (Appendix 1), we show how a few lines of code are sufficient for plotting bAP amplitudes on the shape of neuronal trees (shape plot: see insets in *Figure 5A*) and for creating bAP amplitude vs. distance plots (*Figure 5B*). In Tutorial 7 (Appendix 1), we explain how to evaluate Ca$^{2+}$ dynamics in different compartments of a model using the T2N tools.

In the specific case of GCs, we used T2N and our active GC models from *Figures 3* and *4*, to compare backpropagating APs (bAPs) in mouse and rat (for details see Appendix 2). The rat GC model was able to reproduce bAP attenuation (*Figure 5A*, left; see Appendix 2 for details) determined from dendritic patch clamp experiments (*Krueppel et al., 2011*). Interestingly, for a realistic delay of the bAP peak, we had to adjust the specific axial resistance R$_a$ and the passive membrane conductance to the higher temperature of 33°C (*Figure 5B*, left) that was used in experiments (*Krueppel et al., 2011*). This provides further evidence for the consistency of our model with experimental data. Moreover, we used a well-tuned phenomenological Ca$^{2+}$ buffer model (see Appendix 2 for details), which generated realistic intracellular Ca$^{2+}$ signals induced by bAPs (*Figure 5C*, left). Ca$^{2+}$ dynamics in synthetic morphologies (*Figure 5C*, green bars) matched Ca$^{2+}$ signals from

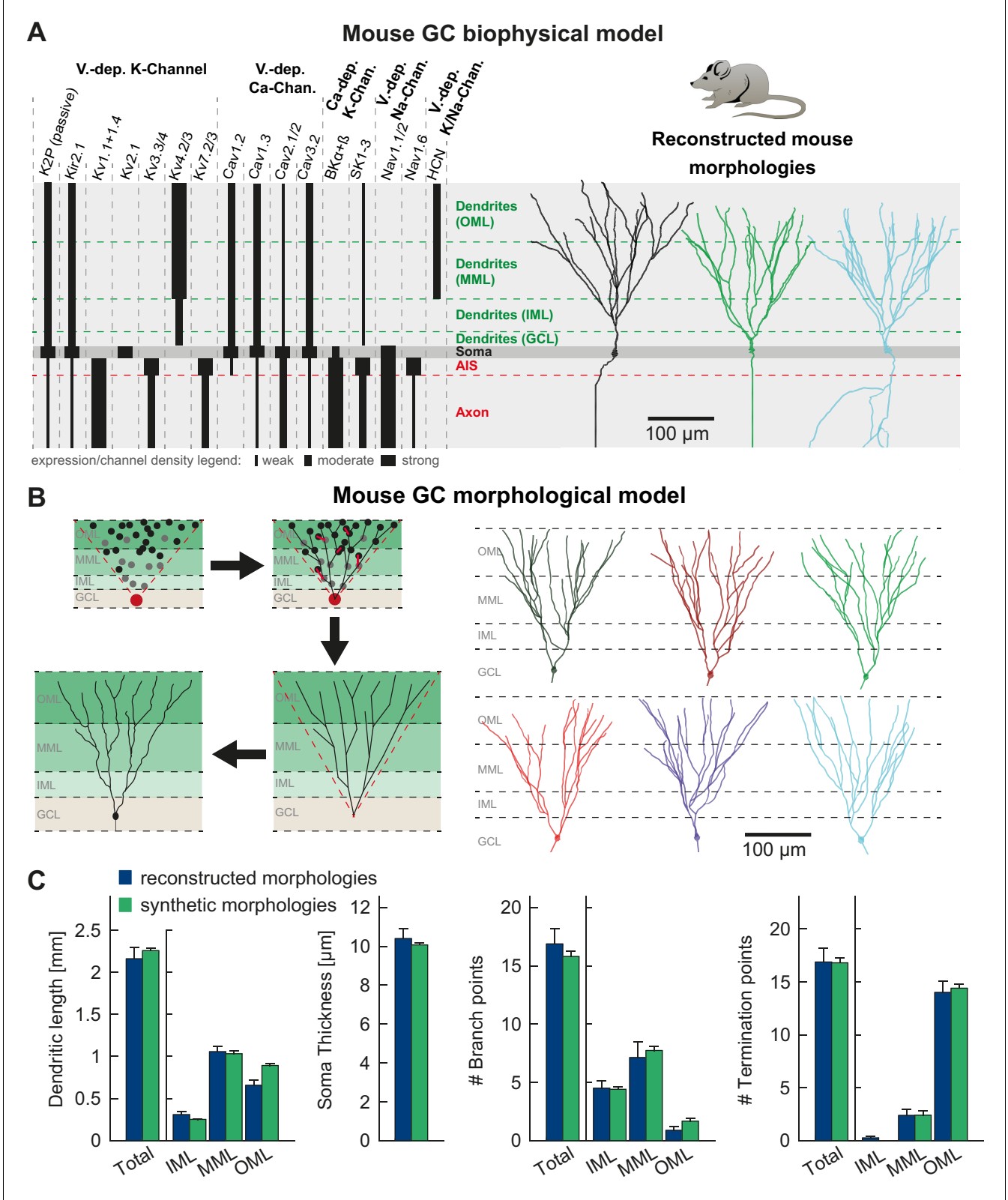

**Figure 2.** T2N supports incorporation of realistic ion channels and synthetic morphologies. (**A**) Ion channel composition of the mouse dentate granule cell (GC) model. Left: Passive and active ion channels with their specific distribution in six different regions: outer molecular layer (OML), middle molecular layer (MML), inner molecular layer (IML), soma, axon initial segment (AIS) and axon. The relative spatial distribution of voltage-dependent (V.-dep.) and calcium-dependent (Ca$^{2+}$-dep.) channels is in line with an extensive amount of data from the literature (see **Table 1**, Appendix 2 and

*Figure 2 continued on next page*

*Figure 2 continued*

Materials and methods for details). Right: Three exemplary morphologies out of eight reconstructed mouse GCs (*Schmidt-Hieber et al., 2007*) used for compartmental modeling of mouse GCs. (B) Schematic of the morphological model used to generate synthetic mouse morphologies which is analogous to the previously reported rat model (*Beining et al., 2017*; see Material and methods there for details). Upper left: A synthetic 3D young dentate gyrus (DG) was created comprising different layers (GCL, IML, MML, and OML, from bottom to top). A soma (red dot) was defined and random target points (black dots) were distributed within a 3D cone (red dashed lines). These points were complemented by directed target points (gray dots) that were placed automatically between clusters of target points and the soma. Upper right: The target points were connected by a minimum spanning tree algorithm (*Cuntz et al., 2010*) and terminal dendritic segments shorter than 20 µm were pruned off (red segments, see *Beining et al., 2017*). Lower right: The young DG and the dendritic tree have been stretched to their mature size (see *Beining et al., 2017* for more information). Lower left: Adding a somatic diameter profile, a synthetic axon, applying jittering and dendritic diameter taper (not shown for visualization purposes) to the dendrites results in realistic synthetic GC morphologies suitable for compartmental modeling. (C) Six out of 15 synthetic morphologies created by the morphological model and used for compartmental modeling with their anatomical borders (gray dashed lines). (D) General and layer-specific structural comparison of the reconstructed (blue, *Schmidt-Hieber et al., 2007*) and synthetic (green) mouse GC morphologies.

DOI: https://doi.org/10.7554/eLife.26517.003

experiments (black bars) as well as those from reconstructed morphologies (blue bars). Thus, our GC model generated realistic and stable intracellular $Ca^{2+}$ dynamics over a broad range of different morphologies in rat and mouse.

After validating the rat GC model for bAPs, we computed an experimentally testable prediction for bAP attenuation in mouse GCs (*Figure 5A,B*, right), for which no experimental data on bAPs exist so far. The model predicted that bAP attenuation was smaller in mouse GCs than in rat GCs. This prediction can be tested by dendritic patch clamp recordings in mouse GCs. We also computed a prediction for bAP-induced intracellular $Ca^{2+}$ changes in mouse morphologies (*Figure 5C*, right). $Ca^{2+}$ levels in mouse GCs were comparable to $Ca^{2+}$ levels in rat GCs. Provided that $Ca^{2+}$ buffering and extrusion mechanisms are comparable between mouse and rat GCs (*Stocca et al., 2008*), this suggests that dendritic $Ca^{2+}$ signaling is relatively similar in both species despite differences in back-propagating dendritic voltage spread.

## Example of sensitivity analysis performed with T2N revealing critical ion channels in mature mouse and rat GCs

T2N helps identify crucial parameters affecting electrophysiological behavior of compartmental models because it supports flexible whole cell as well region- and layer-specific manipulations of ion channel properties. In Tutorial 8 (Appendix 1), we show how to use a T2N function to upregulate or downregulate individual or multiple channels in defined regions of a dendritic tree. We applied these T2N features to perform a sensitivity analysis of the GC model predicting the effects of a reduction (*Figure 6A*) or an increase (*Figure 6—figure supplement 1*) of model parameter values. In addition, we used the *T2N* channel block function (Tutorial 8, Appendix 1) to completely turn off individual ion channels ($K_v$3, BK, SK and Kv7) and explore their impact on AP repolarization and spike adaptation (*Figure 6B,C*). In Appendix 2, we provide a summary of the results with detailed information on key GC ion channels and other factors (e.g. temperature) involved in the regulation of GC excitability, action potential (AP) properties, voltage propagation and output firing. These results describe single parameter sensitivity analyses. However, T2N can be used also for analyzing the impact of any combinations of parameters, thus contributing to assessments of degeneracy in compartmental models (see Discussion).

## T2N supports prediction of clinically relevant ion channel alterations in multiple neuronal morphologies

T2N's strength relies in its capability of handling and manipulating compartmental simulations in many morphologies. Therefore, it is suitable to predict the consequences of ion channel changes not only in healthy cells but also under pathological conditions. We exemplify this by using our mouse GC model to calculate the effects of compensatory ion channel alterations observed during temporal lobe epilepsy (TLE). As shown previously in experiments, protective upregulation of HCN and Kir (*Young et al., 2009*; *Stegen et al., 2012*) or $K_v$1.1 channels (*Kirchheim et al., 2013*) decreases GC excitability under epilepsy conditions. In line with these data, our model GCs exhibited similar changes (*Figure 6D,E*; see also *Figure 6—figure supplement 2* and Appendix 2 for further details).

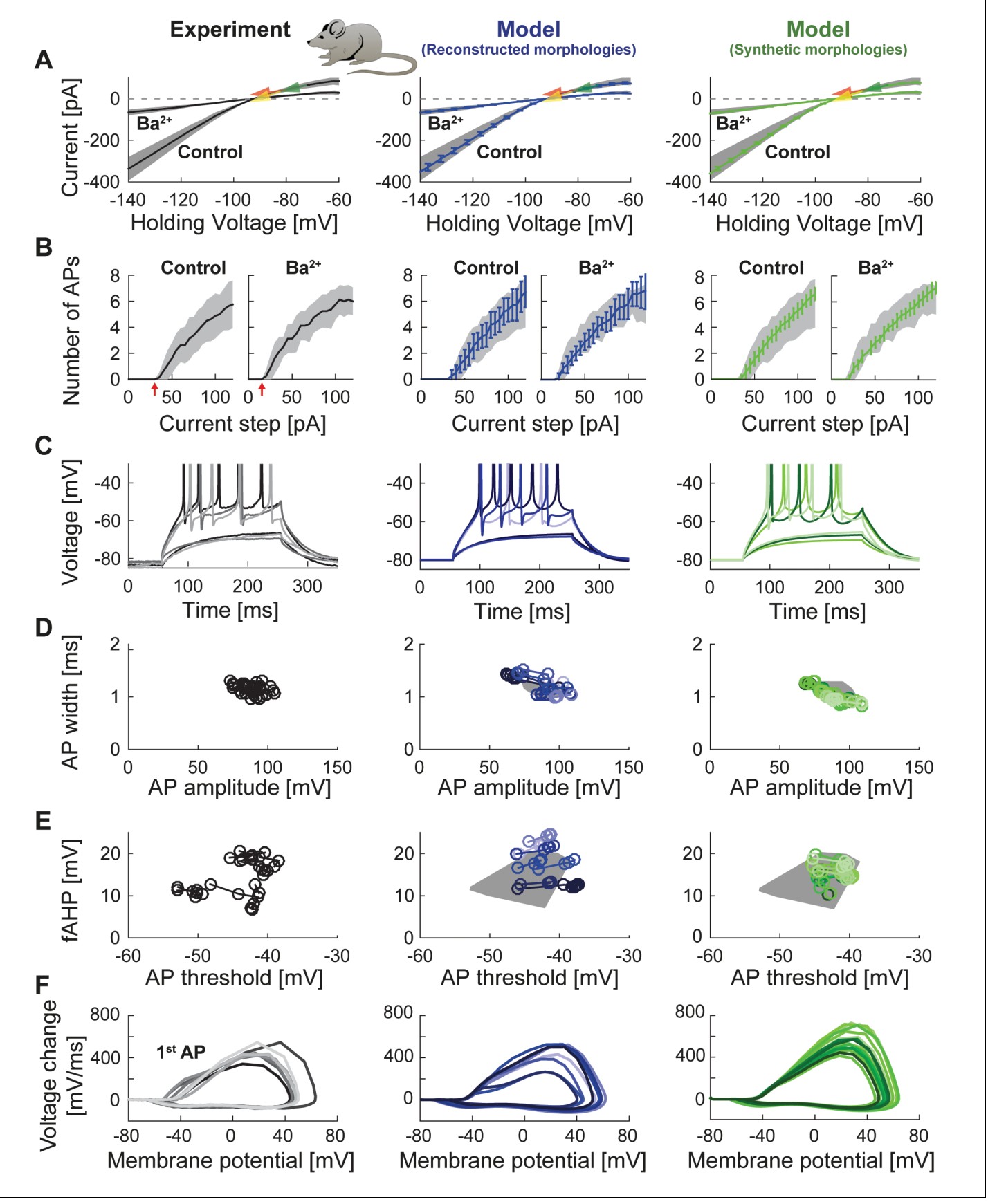

**Figure 3.** Passive and active properties of the mature mouse GC model. Comparison of electrophysiological features between experimental data (left column, grayish colors) (*Mongiat et al., 2009*), GC model with reconstructed morphologies (middle column, blueish colors) and GC model with synthetic morphologies (right column, greenish colors). (**A**) Current-voltage (I–V) relationships before and after application of 200 μM $Ba^{2+}$. Simulations (blue and green curves) are compared to experimental data (mean and s.e.m. from raw traces (*Mongiat et al., 2009*) as black curve and gray patch; arrows are average values reported from further literature: red (*Brenner et al., 2005*), yellow (*Mongiat et al., 2009*), green (*Schmidt-Hieber et al., 2007*)). $Ba^{2+}$ simulations correspond to 99% Kir2 and 30 % K2P channel blockade. (**B**) Number of spikes elicited by 200 ms current steps (F-I relationship) from a holding potential of −80 mV. Right subgraph shows F-I relation after adding $Ba^{2+}$. Experimental standard deviation is shown as gray patches in all columns. Red arrows point to the rheobase, which is different between control and $BaCl_2$ application. (**C**) Exemplary spiking traces from control condition in (**B**) (200 ms, 30 and 75 pA somatic current injections). (**D–E**) Action potential (AP) features of the first AP (90 pA somatic step current injection, 200 ms). Convex hulls around experimental data are shown in all columns as gray patches. (**D**) AP width vs. AP amplitude. (**E**) Amplitude of fast afterhyperpolarisation (fAHP) vs. AP threshold. (**F**) Phase plots of the first AP (dV/V curve, 90 pA current step, 200 ms).
DOI: https://doi.org/10.7554/eLife.26517.005

The following figure supplements are available for figure 3:

**Figure supplement 1.** Performance of a widely used GC model with reconstructed and synthetic mouse morphologies.
DOI: https://doi.org/10.7554/eLife.26517.006
**Figure supplement 2.** Influence of morphology on electrophysiological properties in the mature mouse GC model.
DOI: https://doi.org/10.7554/eLife.26517.007
**Figure supplement 3.** Current dynamics during voltage clamp in mature mice GCs.
DOI: https://doi.org/10.7554/eLife.26517.008
**Figure supplement 4.** Maximal rate of voltage change during an AP in the mature mouse GC model.
DOI: https://doi.org/10.7554/eLife.26517.009

This shows that T2N can be used to estimate effects of pathology-related alterations, which are robust across multiple non-identical single-cell morphologies. Moreover, by providing and exploiting powerful morphological modeling tools from the TREES toolbox, T2N creates a unique opportunity for making clinically relevant cell-type models with hundreds to thousands of distinct morphologies that can be inserted into network models to study neuronal pathology on the level of microcircuits or large circuits.

## Example of using T2N for building a data-driven young adult-born GC model

To demonstrate the flexibility of T2N in building compartmental models, we used it to create the first model of young adult-born GCs (abGCs). During a critical period (starting around the 4th week of cell age), abGCs exhibit increased excitability as compared to older abGCs or mature GCs (*Mongiat et al., 2009*). Our goal was to reproduce the electrophysiology of these young (28 days

**Table 2.** Electrophysiology in mature mouse GCs – experiment vs. model.

| Intrinsic properties | Experiment | Model reconstr. morphologies | Model synth. morphologies |
| --- | --- | --- | --- |
| $R_{in}$ [MΩ] (@ −82.1 mV) | 289.5 ± 34.9 | 287.0 ± 14.7 | 279.6 ± 6.9 |
| $c_m$ [pF] | 48.9 ± 5.3 | 55.7 ± 2.8 | 61.2 ± 1.6 |
| tau [ms] | 34.0 ± 2.0 | 31.4 ± 0.2 | 31.6 ± 0.1 |
| $V_{rest}$ [mV] | −92.7 ± 0.5 * | −88.7 ± 0.1 | −88.6 ± 0.0 |
| $I_{threshold}$ [pA] | 47.5 ± 4.5 | 52.5 ± 3.7 | 50.3 ± 1.6 |
| $V_{threshold}$ [mV] | −46.3 ± 1.6 * | −44.9 ± 0.3 | −43.8 ± 0.2 |
| AP amplitude [mV] | 95.6 ± 2.1 | 96.3 ± 2.9 | 97.7 ± 1.7 |
| AP width [ms] | 1.03 ± 0.02 | 1.00 ± 0.04 | 0.93 ± 0.02 |
| fAHP [mV] | 15.7 ± 1.4 | 17.5 ± 1.7 | 15.8 ± 0.8 |
| Interspike interval [ms] | 36.3 ± 4.9 | 36.2 ± 3.2 | 34.5 ± 1.1 |
| Max. spike slope [V/s] | 450.1 ± 23.7 | 428.0 ± 39.5 | 519.7 ± 24.9 |
| gKir [nS] | 5.46 ± 1.31 | 5.90 ± 0.89 | 5.97 ± 0.6 |

*after subtraction of a calculated liquid junction potential of 12.1 mV.
DOI: https://doi.org/10.7554/eLife.26517.010

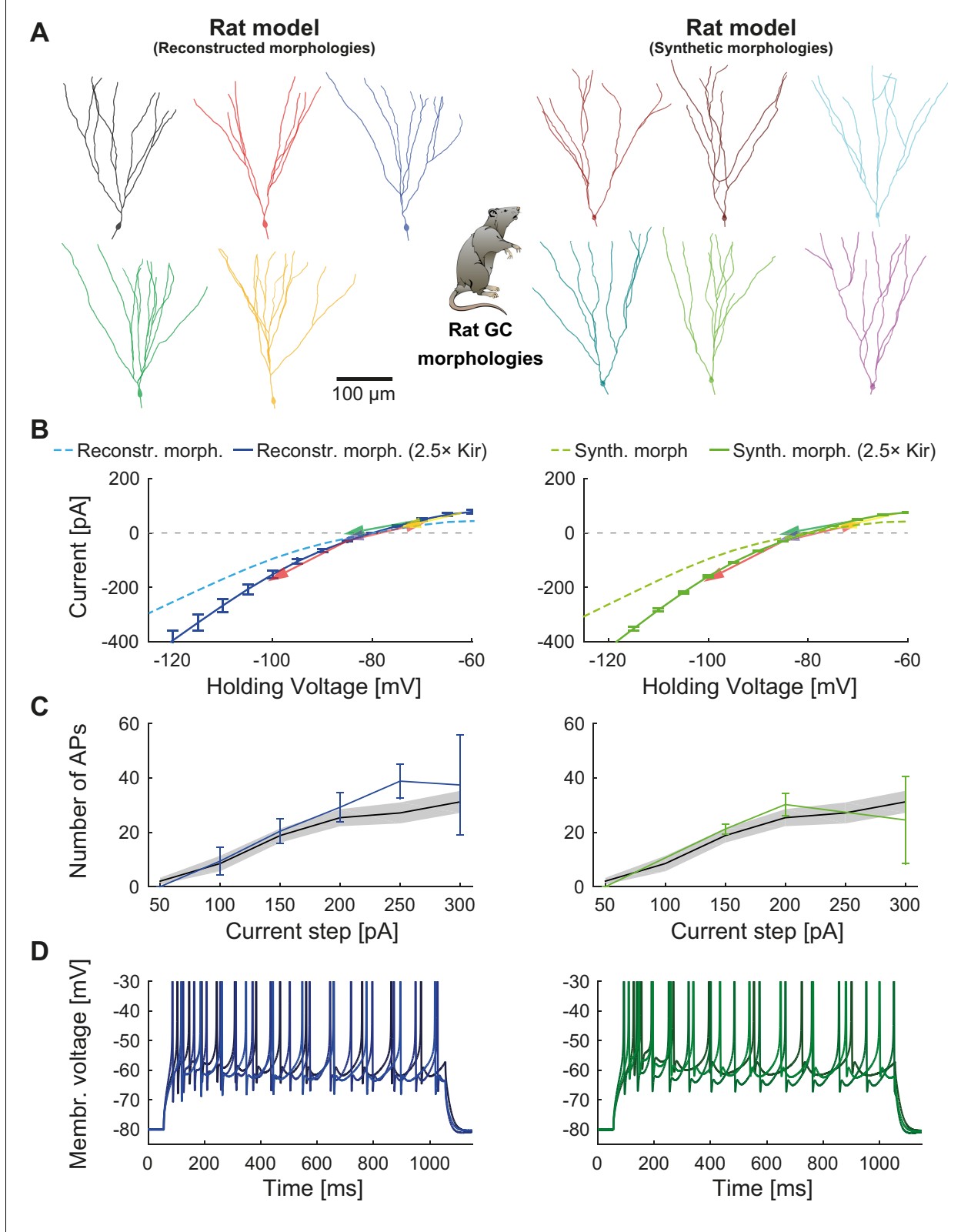

**Figure 4.** Mature rat GC model. Comparison of electrophysiological features between GC model with reconstructed morphologies (left column, blueish colors) and GC model with synthetic morphologies (right column, greenish colors) as it was adapted for reproducing rat data. (**A**) Illustration of reconstructed (left) and synthetic (right) rat morphologies used for simulations of rat GCs, from (***Beining et al., 2017***). (**B**) I-V relationship of the model with (dark solid lines) or without (bright dashed lines) adjustment of passive conductance to experimental rat data (indicated by arrows: red

*Figure 4 continued on next page*

*Figure 4 continued*

(*Staley et al., 1992*), yellow (*Mateos-Aparicio et al., 2014*), green (*Pourbadie et al., 2015*), violet (*Schmidt-Hieber et al., 2004*). (C) F-I relationship of the model compared to data (black line and standard deviation as gray patch) from *Pourbadie et al., 2015*. (D) Exemplary spiking traces simulated during a 1 s current injection of 200 pA.

DOI: https://doi.org/10.7554/eLife.26517.011

The following figure supplement is available for figure 4:

**Figure supplement 1.** Performance of the classical GC model with reconstructed and synthetic rat morphologies.

DOI: https://doi.org/10.7554/eLife.26517.012

old) abGCs (*Mongiat et al., 2009*). For this purpose, we adapted our mature GC model by modifying its biophysics according to ion channel data from abGCs and postnatal developing GCs (*Table 3*, *Figure 7*; see Appendix 2 for details). We did not change the morphology of dendrites since our previous study in rat showed that dendritic trees of young and mature adult-born GCs are similar (*Beining et al., 2017*). In line with experimental findings (*Mongiat et al., 2009*), changing the expression of Kir2 channels and other channel types (*Table 3*) led to altered I-V curves and increased excitability in young abGCs as compared to mature GCs (*Figure 7*). These results indicate that the robust compartmental models generated using T2N can easily be adapted for exploring varying electrophysiological states of the same cell type, for example, during adult neurogenesis and potentially also during development.

## T2N simplifies modeling of synaptic drive and facilitates making experimental predictions

T2N makes it easier to equip compartmental models with layer-specific synaptic inputs and connect them to spike generators. Tutorial 9 (Appendix 1) provides a step-by-step description of T2N-assisted insertion of AMPA synapses modeled as exponential rise and decay of synaptic conductance upon receiving spikes from a spike train generator (artificial presynaptic cell). The tutorial also explains how to generate random (Poisson) spike train to drive presynaptic spike generators. Because generating random spike streams in NEURON is not trivial (see https://www.neuron.yale.edu/neuron/node/60), T2N allows users to employ random number generators of Matlab to simplify this process. This illustrates one strength of T2N, which relies in providing Matlab functions not only for analyzing simulation results but also for setting up models as well as their instrumentation and control.

We employed above-mentioned functions of T2N to generate experimentally testable predictions for synaptic integration of abGCs and mature GCs (mGCs; *Figure 8*, see Appendix 2 for details). Four-week-old abGCs are known to have a lower number of excitatory synapses as reflected by lower spine densities (*Zhao et al., 2006*) and decreased frequency of miniature excitatory post-synaptic currents (*Mongiat et al., 2009*). Importantly, when abGCs were driven by a smaller number of synapses, they exhibited similar synaptic input/output relationships as mGCs (*Figure 8A*). This suggests that higher intrinsic excitability of abGCs (see also *Figure 7C*) compensates for their lower numbers of synaptic inputs. Moreover, our modeling indicates that both young abGCs and mature GCs are tuned to follow input frequencies in the theta range (<10 Hz; *Figure 8B*). This result is consistent with studies showing that diminished glutamatergic input is compensated by the enhanced excitability when GABAergic inhibition is blocked (*Mongiat et al., 2009*; *Pardi et al., 2015*).

Finally, we used the model to test synaptic integration of abGCs for temporally shifted synaptic inputs. Our model predicts that at low frequencies in the theta range, young abGCs were able to integrate synaptic inputs with a broader time window than mGCs (*Figure 8B* and Appendix 2). This is in line with the proposed special role of abGCs in hippocampal pattern separation and integration due to their broader tuning to the activity of synaptic inputs (*Aimone et al., 2010*; *Johnston et al., 2016*; *Rangel et al., 2013*). In conclusion, our model reproduces and predicts the activation patterns of young and mature granule cells under those conditions when inhibition is not present.

## Discussion

In this work, we developed *T2N*, a novel software tool for linking morphological with compartmental modeling and analysis. *T2N* allows communicating seamlessly between the *TREES toolbox* in Matlab

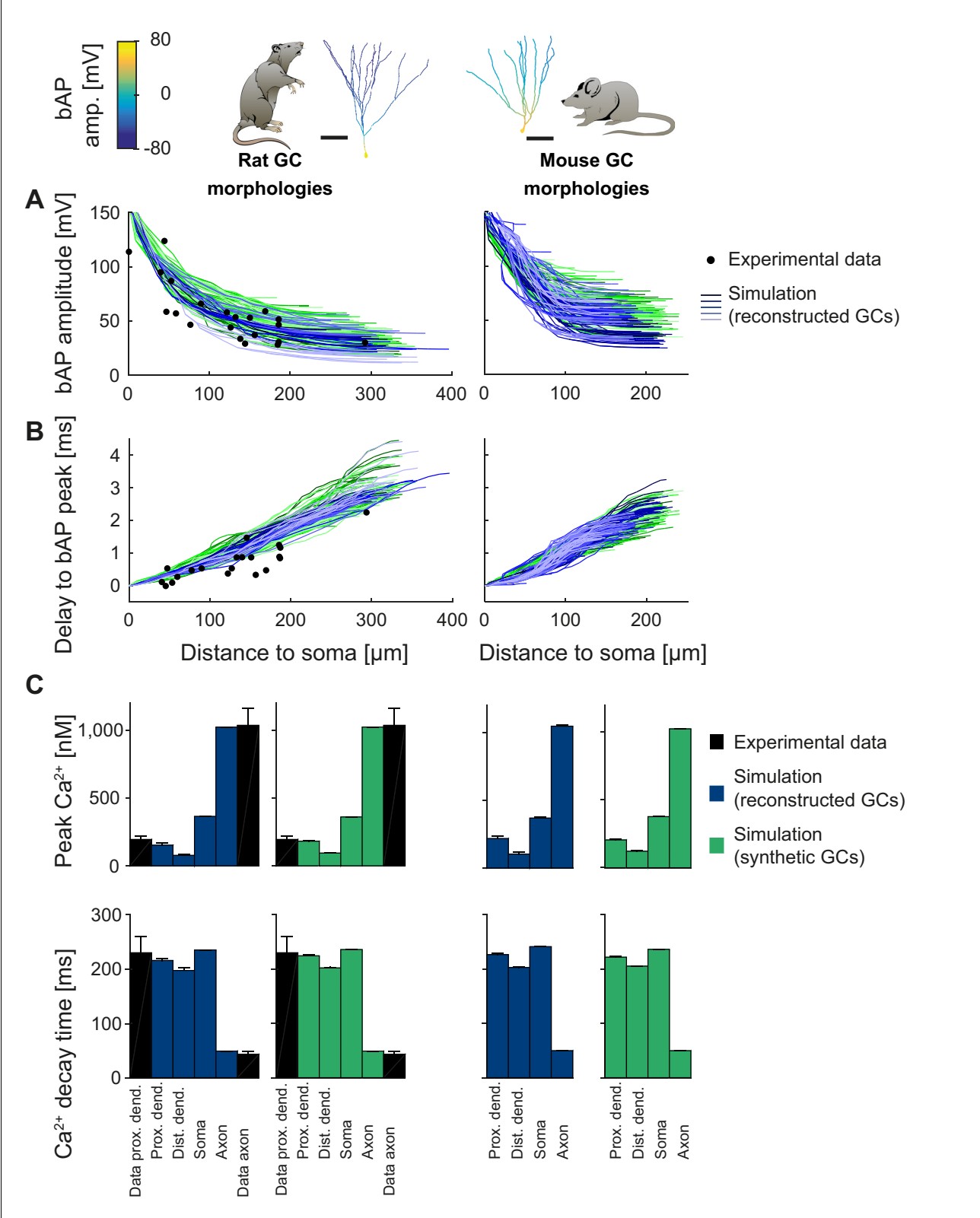

**Figure 5.** Backpropagating action potentials (bAPs) in mature mouse and rat GC models. bAP characteristics at 33°C (experiment and simulation), elicited in the soma by a brief current injection. Inset: Exemplary rat and mouse GC morphology with local maximum voltage amplitudes. (**A**) Maximal voltage amplitude as a function of Euclidean distance from the soma. Black data points are experimental data from rat (*Krueppel et al., 2011*). There are no available data on bAP characteristics for mouse GCs. (**B**) Corresponding delay of the maximal bAP amplitude in the model compared to

*Figure 5 continued on next page*

*Figure 5 continued*
experimental rat data (black dots) (***Krueppel et al., 2011***). (**C**) Peak Ca²⁺ amplitudes at room temperature following an AP measured at different locations in the rat (left) and mouse (right) GC model using reconstructed (blue) and synthetic (green) morphologies. Experimental rat data measured in proximal dendrites (***Stocca et al., 2008***) and axonal mossy fiber boutons (MFBs) (***Jackson and Redman, 2003***) are added as black bars. There are no available data on bAP characteristics for mouse GCs. (**D**) Ca²⁺ decay time constants analogous to C.
DOI: https://doi.org/10.7554/eLife.26517.013
The following figure supplement is available for figure 5:

**Figure supplement 1.** Backpropagating action potentials (bAPs) in the classical GC model.
DOI: https://doi.org/10.7554/eLife.26517.014

(***Cuntz et al., 2010***, ***2011***) and the software package *NEURON* (***Carnevale and Hines, 2006***). *T2N* enables to fit models directly on any population of morphologies including those from morphological models. In this way, we provide tools to generate the kind of robust models for which we presented one example for dentate granule cells (GCs). *T2N* as well as the new GC model are freely available online (http://www.treestoolbox.org/T2N.html; senselab.med.yale.edu/modeldb/, accession # 231862; we also uploaded a pure NEURON version of the GC model including all morphologies and biophysics but only two protocols on ModelDB under the accession # 231818) as a resource for scientists working with detailed biophysical compartmental models.

What is the strength of *T2N*? What are its unique features? *T2N* provides user-friendly definition and control of *NEURON* compartmental models (morphologies, channel distributions, simulations etc.) as well as a subsequent analysis with *Matlab* and the *TREES toolbox*. Moreover, the automatic parallelization of multiple simulation runs (e.g. to create an F-I relationship) and the parallelization option using *NEURON*'s parallel computing feature (***Migliore et al., 2006***) for single simulations that include a large amount of cells (e.g. large-scale networks) reduces simulation time considerably. The clear structure of the definitions of a model's biophysical features as well as the automatically produced stereotyped *NEURON* code improves reading the model scripts and merging of different models developed with *T2N*. Sensitivity analyses, plots and visualizations are much easier to do with T2N than other commonly used software tools. In addition to simulations in reconstructed morphologies, *T2N* easily allows running simulations using synthetic morphologies from morphological models thereby facilitating the generation of biophysically and morphologically realistic large-scale network models. By enabling the use of diverse reconstructed and synthetic dendritic trees, *T2N* makes it possible to generalize the predictions of compartmental simulations to any morphology and supports the search for universal principles valid across different species and cell types. By supporting the inclusion of variable morphologies and precise incorporation of ion channels, *T2N* will allow users to more fully harness the resources from online databases such as NeuroMorpho (***Ascoli et al., 2007***) and IonChannelGenealogy (***Podlaski et al., 2016***) or Channelpedia (***Ranjan et al., 2011***). In summary, *T2N* is a versatile and adaptable tool for extensive in silico structure-function analyses in *NEURON*.

## New robust GC model

Using *T2N* we developed a new compartmental model that mimics the detailed electrophysiological behavior of mature GCs and young abGCs in mouse and rat. The model has five important advantages and improvements when compared to previously published models: (1) Our model is the first compartmental GC model – and one of the first neuron models overall – which remains robust across a wide variety of reconstructed and synthetic morphologies. (2) The model contains only conductances of channel isoforms that are currently known to exist in GCs and accurately implements their kinetics. The model is based on information from more than 220 publications (see the Reference list) that were required to cover the full extent of the biological detail in our model, rendering its development an in-depth quantitative review of the electrophysiology of granule cells. (3) The model is capable of reproducing findings and experiments from many different studies. (4) After adjustment of Kir2 channel density, the model reproduced electrophysiological behavior of both rat and mouse mature GCs indicating that these species might share similar active channels. (5) The adapted model for young abGCs represents the first available data-driven compartmental model of these neurons. With this consistent model at hand, we were able to reproduce the effects of compensatory ion channel changes under epileptic conditions in mature GCs. Furthermore, the model predicted the

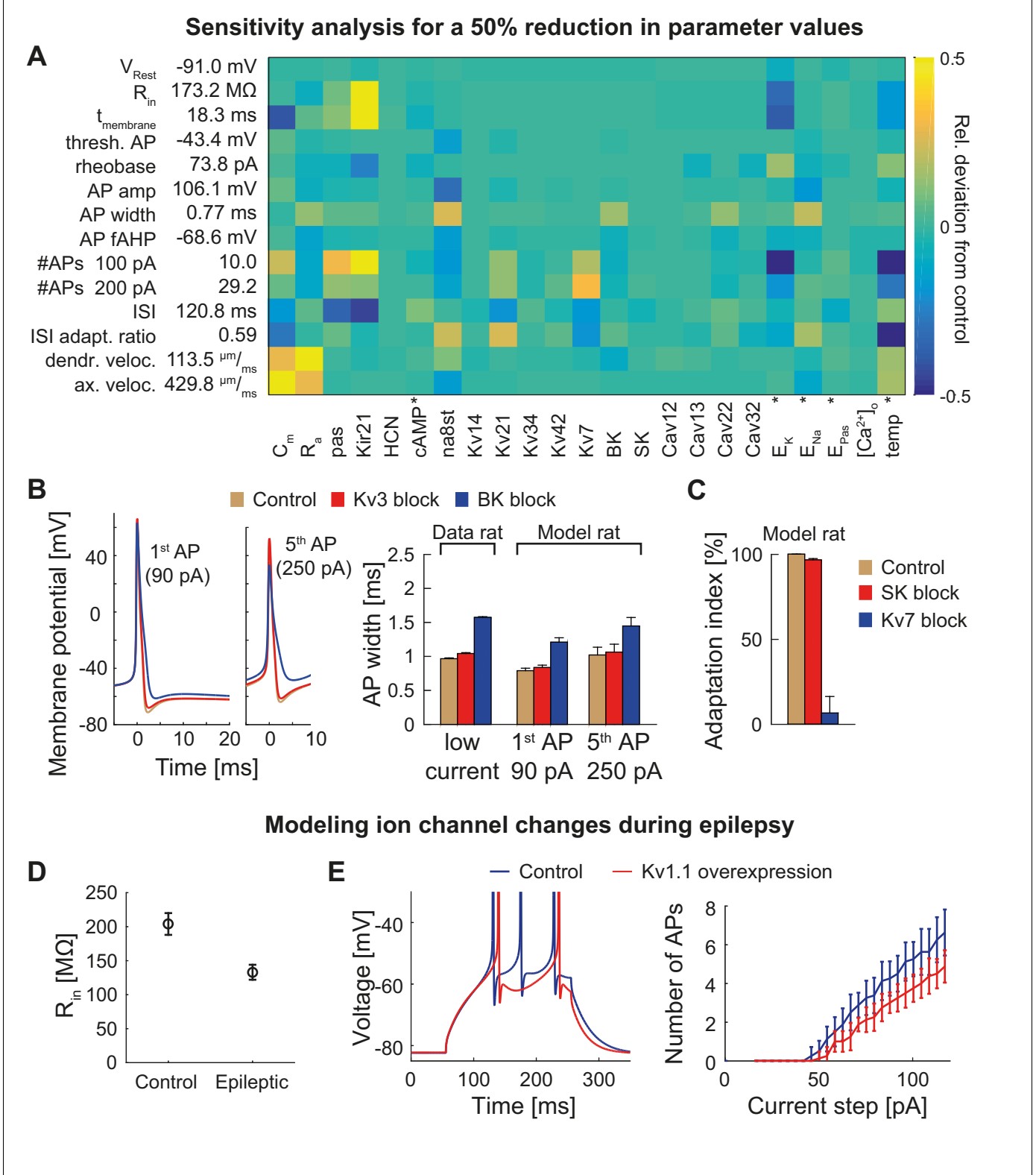

**Figure 6.** Dependence of the model on specific channels and parameters. (**A**) Sensitivity matrix showing the relative change (color-coded) in electrophysiological parameters (y-axis) in the mature rat GC model following a 50% reduction in ion channel densities or other model parameters (x-axis), except for the cases marked with an asterix (*): the reversal potential of potassium $E_K$ as well as the passive reversal potential $E_{Pas}$ were raised by +10 mV (to reduce ionic drive) and $E_{Na}$ was lowered by −20 mV. The temperature was raised by +10°C. cAMP concentration (influencing HCN

*Figure 6 continued on next page*

*Figure 6 continued*

channels in the model) was raised from 0 to 1 µM. (B) Left: Exemplary voltage traces during 1 s current injection of 90 pA (left, first AP) or 250 pA (right, fifth AP) under control (black lines), K$_v$3.4 block (red lines) or BK block (blue lines) conditions in the mature rat GC model. Right: Half-amplitude AP widths compared to experimental data that used paxilline to block BK (***Brenner et al., 2005***; ***Müller et al., 2007***) or BDS-I to block K$_v$3.4 channels (***Riazanski et al., 2001***). (C) Impact of the blockade of SK and K$_v$7 channels on spike frequency adaptation in the mature rat GC model. (D) Input resistance measurements in the rat GC model in the control case and when post-epileptic conditions are modeled (doubled Kir2 and HCN channel conductance). (E) A reported overexpression of K$_v$1.1 following an in vivo approach to elicit temporal lobe epilepsy in mice (***Kirchheim et al., 2013***) was mimicked in silico by a three-fold increase of K$_v$1.1 channel density in the mature mouse GC model. Left graph illustrates increased spiking delay, whereas the right plot shows the reduced excitability.

DOI: https://doi.org/10.7554/eLife.26517.015

The following figure supplements are available for figure 6:

**Figure supplement 1.** Sensitivity analysis for a doubling of parameter values in the mature rat GC model.
DOI: https://doi.org/10.7554/eLife.26517.016
**Figure supplement 2.** Test for resonance in the rat GC model.
DOI: https://doi.org/10.7554/eLife.26517.017

impact of differences in intrinsic properties between young abGCs and mature GCs on the temporal summation of synaptic input. We found that the higher intrinsic excitability allows young abGCs to integrate synaptic inputs in a broader time window compared to mature GCs. Altogether, this suggests a universal nature of the stability of the model. To sum up, our granule cell simulations provide important insights and tools for the hippocampus research field in general and the adult neurogenesis field in particular. Our study builds the cornerstone for future GC modeling approaches, by providing a model with which hypotheses on the impact of structural and functional alterations can be tested and further mechanisms such as synaptic plasticity and inhibition can be added at will. Our study further underlines the importance of biological soundness and the appropriate level and amount of detail for realistic modeling.

## Morphologically robust compartmental modeling

Many existing GC compartmental models were based on a very simplified representation of morphology comprising two cylinders in place of realistic dendrites (e.g. *Jedlicka et al., 2015*; *Santhakumar et al., 2005*). Furthermore, models that did not use such simplified compartments were mostly tested in single morphologies (*Aradi and Holmes, 1999*; *Ferrante et al., 2009*). Therefore, there was a need for a new biophysical model, which would be transferable to further morphologies. We found previous biophysical models of hippocampal GCs to be unstable across different

**Table 3.** Ion channels or currents that were reported to be less expressed in immature GCs and were downregulated in the young GC model

| Channel name | Cell type and Reference | Downregulation in the model [%] |
| --- | --- | --- |
| Kir 2.x | Young adult-born GCs (***Mongiat et al., 2009***) | 73 |
| K$_v$1.4 | Young postnatal GCs (***Maletic-Savatic et al., 1995***; ***Guan et al., 2011***) | 0 |
| K$_v$2.1 | Young postnatal GCs (***Maletic-Savatic et al., 1995***; ***Antonucci et al., 2001***; ***Guan et al., 2011***) | 50 |
| K$_v$3.4 | Young postnatal GCs (***Riazanski et al., 2001***) | 0 |
| K$_v$4.2/4.3 +KChIP/DPP6 | Young postnatal GCs (***Maletic-Savatic et al., 1995***; ***Riazanski et al., 2001***) | 50 |
| K$_v$7.2 and 7.3 (KCNQ2 and 3) | Young postnatal GCs (***Tinel et al., 1998***; ***Smith et al., 2001***; ***Geiger et al., 2006***; ***Safiulina et al., 2008***) | 50 |
| Na$_v$1.2/6 | Young postnatal GCs (***Liu et al., 1996***; ***Pedroni et al., 2014***) | 25 |
| Ca$_v$1.2 | Young postnatal GCs (***Jones et al., 1997***) | 0 |
| Ca$_v$1.3 (L-type) | Young postnatal GCs (***Kramer et al., 2012***) | 50 |
| BK-α/BK-β4 | Young postnatal GCs (***MacDonald et al., 2006***; ***Xu et al., 2015***) | 40/100 |

DOI: https://doi.org/10.7554/eLife.26517.019

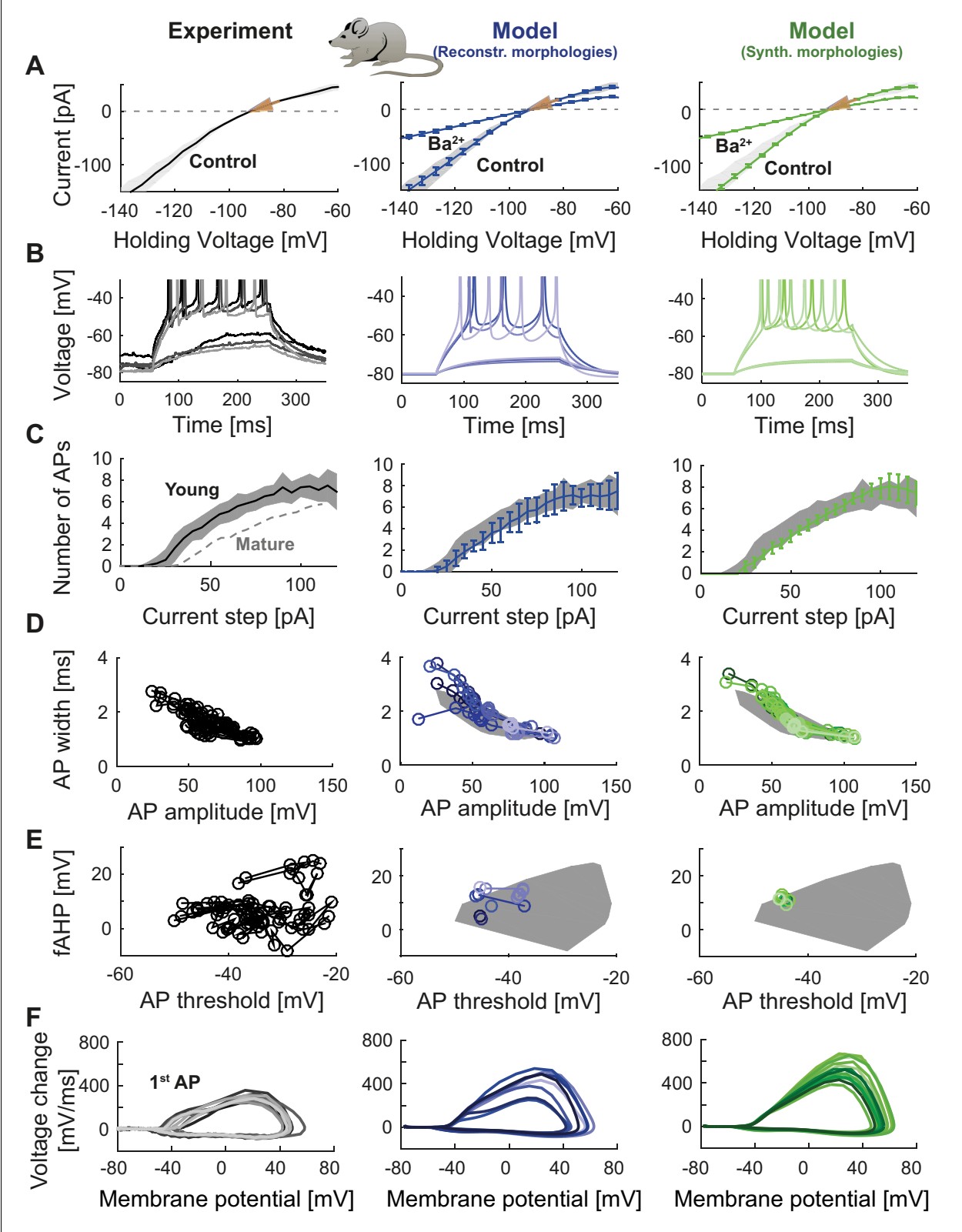

**Figure 7.** Model of young adult-born granule cells (abGCs) in mice. Panels are analogous to *Figure 3*, with comparison of electrophysiological features between experimental data (left column, grayish colors), GC model with reconstructed morphologies (middle column, blueish colors) and GC model with synthetic morphologies (right column, greenish colors). The experimental data of young abGCs at a cell age of 28 dpi is from *Mongiat et al. (2009)*. The model was obtained by a reduction of several ion channels (see *Table 3*). (**A**) Current-voltage (I–V) relationships before and after

*Figure 7 continued on next page*

*Figure 7 continued*

application of 200 µM Ba$^{2+}$; Ba$^{2+}$ simulations correspond to 99% Kir2 and 30 % K2P channel blockade. Experimental measurements of R$_{in}$ in 28 dpi old abGCs from further literature are indicated by arrows (red [*Mongiat et al., 2009*], green [*Piatti et al., 2011*], pink [*Yang et al., 2015*]). (B) Exemplary spiking traces (200 ms, 10 and 50 pA somatic current injections). (C) Number of spikes elicited by 200 ms current steps (F-I relationship). Experimental standard deviation is shown as gray patches in all columns and the F-I curve of mature GCs is plotted in the left column (gray dashed line) for comparison. (D–E) Action potential (AP) features (90 pA somatic step current injection, 200 ms). Convex hulls around experimental data are shown in all columns as gray patches. (D) AP width vs. AP amplitude. (E) Amplitude of fast afterhyperpolarisation (fAHP) vs. AP threshold. (F) Phase plots of the first AP (dV/V curve, 90 pA current step, 200 ms).
DOI: https://doi.org/10.7554/eLife.26517.018

dendritic morphologies (*Figure 3—figure supplement 1*, *Figure 4—figure supplement 1*, *Figure 5—figure supplement 1*, *Supplementary file 1*). In our study, we introduced electrophysiological variability (*Figure 3—figure supplement 2*) to the compartmental model by using diverse realistic and synthetic morphologies while keeping the channel densities the same. We developed a morphological mouse model capable of reproducing detailed morphological parameters of reconstructed mouse GCs (*Schmidt-Hieber et al., 2007*). We also created synthetic rat GCs using our recently published morphological model fitted on fully reconstructed rat morphologies (*Beining et al., 2017*). The morphological variability produced by each model was similar to the biological variability in the reconstructions. Interestingly, the resulting electrophysiological variability was in the range of experimental data indicating that morphological variability is able to account for a large part of electrophysiological variability. Hence, our model provides a valuable tool to create a DG network model with thousands of different but realistic GC morphologies (c.f. *Schneider et al., 2012*; *Schneider et al., 2014*) and data-driven GC spiking behavior.

Why was our compartmental model able to reliably reproduce electrophysiological data despite morphological variability of dendrites? One important reason is that our biophysical mechanisms were based on detailed, up-to-date knowledge of the ion channel distribution and kinetics. Second, both morphological as well as biophysical model parameters were determined in a species-, cell type- and cell-age-specific manner. Third, instead of using one morphology or simplified morphologies, we tuned the model using a large set of realistic dendritic trees. Fourth, because we implemented realistic intracellular Ca$^{2+}$ dynamics we did not have to use unrealistic Ca$^{2+}$- or Ca$^{2+}$-dependent channel densities (for details see Appendix 2). This is a significant amendment of previous GC models. Fifth, instead of using a single voltage trace or a single recording, we used several traces and datasets to tune the model. By reproducing numerous electrophysiological phenotypes rather than one phenotype, our approach was similar to a multiple objective approach of *Druckmann et al. (2007)*. Of note, we found a single solution that works across many morphologies, not a set of solutions with different parameter combinations for each morphology. However, we do not exclude the possibility that there exists such a set of solutions with distinct parameters for different morphologies. Taken together, our work suggests that morphologic robustness arises naturally in models in which parameters have been tuned using multiple different experiments and morphologies. This conclusion is also supported by a comparison of our biophysical model to an earlier widely used GC model from *Aradi and Holmes (1999)* (e.g. *Schneider et al., 2012*; *Liu et al., 2014*; *Mateos-Aparicio, Murphy and Storm, 2014*; *Jedlicka et al., 2015*; *Platschek et al., 2016*), which failed to reproduce electrophysiological data after transferring it to diverse mouse and rat GC morphologies (*Figure 3—figure supplement 1*, *Figure 4—figure supplement 1*, *Figure 5—figure supplement 1*, *Supplementary file 1*). However, the point of our new model is not to disregard valid predictions of previously published compartmental GC models. We rather emphasize the need for using diverse morphologies in combination with realistic channels for the improvement of GC models and compartmental models in general. We believe that now the community of computational neuroscientists should start to build models, which perform well outside of the scope, for which they were created (*Almog and Korngreen, 2016*). T2N provides a way to achieve this.

## Predictions of the GC model

Our results suggest that mature rat GCs display a reduced excitability due to incorporation of additional Kir channels. As an alternative, this could also be achieved by other leak channels such as K2P channels; however, the rat I-V curve from experimental data in *Figure 4B* showed pronounced

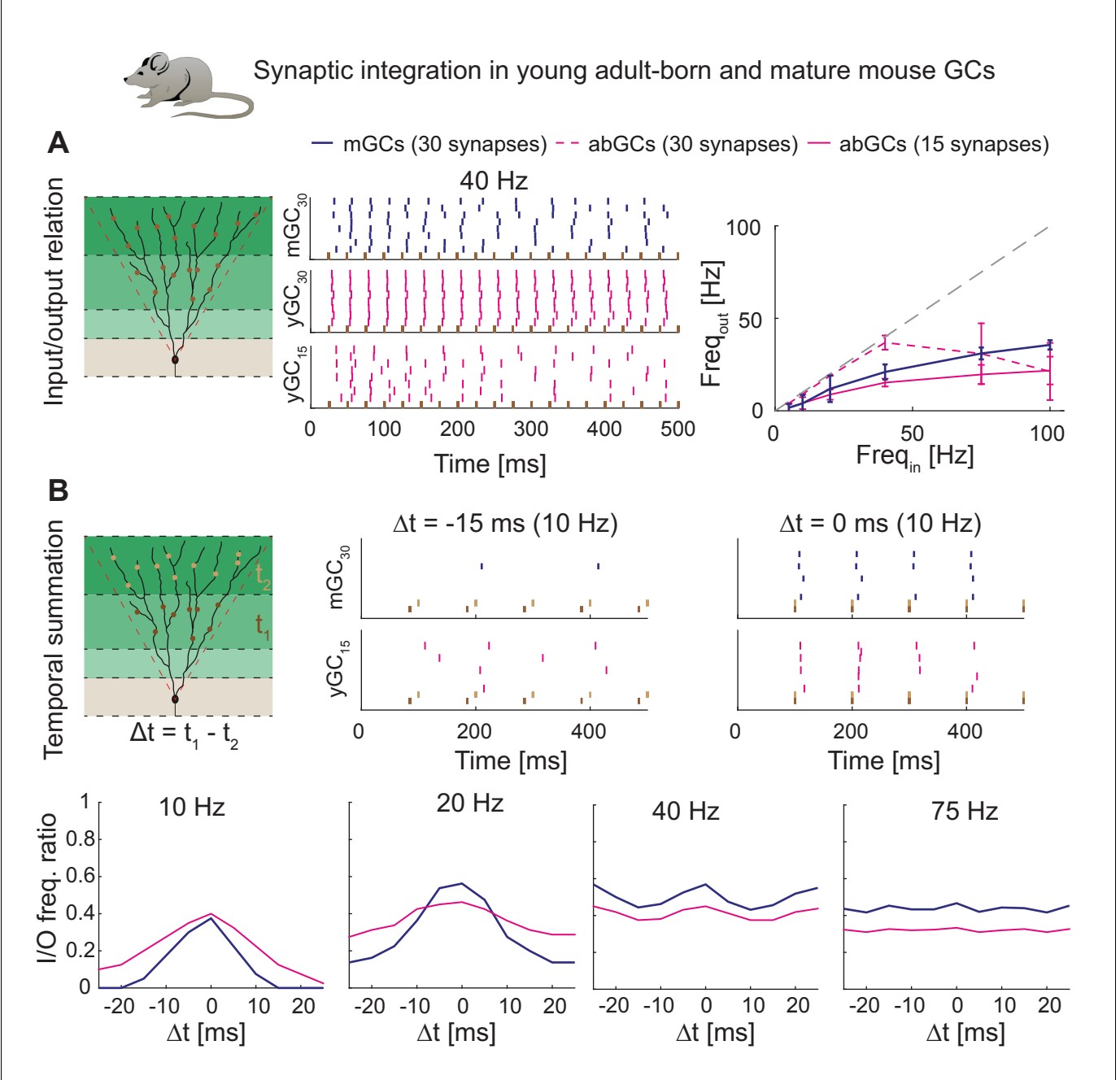

**Figure 8.** Synaptic integration in young abGCs vs. mature GCs. (**A**) Left: Scheme of the simulation configuration with 15 synapses distributed in the MML and 15 in the OML. Middle: All synapses are activated synchronously at 40 Hz. Note that young abGCs (middle row) followed the input (black vertical lines) better than mature GCs (upper row), but performed similarly (lower row) when the biologically lower synapse number (15 synapses in total, yGC_15) was implemented. Right: Summary of the input/output relation at all tested frequencies (5, 10, 20, 40, 75, 100 Hz). Gray dashed line illustrates the theoretically perfect input/output ratio. (**B**) Upper left: Scheme of the simulation configuration when MML and OML synapses are activated with a delay of Δt to analyze temporal summation of inputs. Upper right: Note that young abGCs perform better than mature GCs at following the 10 Hz input when the MML and OML inputs are delayed (left, −15 ms) compared to synchronous activation (right, 0 ms). Lower row: Summary over all tested frequencies (10, 20, 40, 75 Hz) showing that young abGCs have a broader time window of temporal summation than mature GCs at low frequencies but perform slightly worse than mature GCs at high frequencies.

DOI: https://doi.org/10.7554/eLife.26517.020

inward rectification, further supporting Kir channels as an underlying mechanism. In line with this, the increased leak conductance in the rat GC model improved the fitting of simulated bAP attenuation to physiological recordings obtained from rat experiments (*Krueppel et al., 2011*), as the attenuation was too weak in the unmodified rat model (data not shown).

In our attempt to create the first compartmental model of abGCs, we focused on their special intrinsic, non-synaptic properties known to exist at the start of the critical time window, namely their increased input resistance and weaker Na/K peak conductance (*Mongiat et al., 2009*). To implement these changes we used data on ion channels which are known to be upregulated during postnatal development (*Table 3*) assuming that adult-born is similar to postnatal GC development (*Espósito et al., 2005*; *Zhao et al., 2006*; *Snyder et al., 2012*). Even though a lower expression (or alternative splicing) of BK channels is only visible at P14 or earlier, we also had to reduce BK channels in our young abGC model because the fast AHP, which is mainly regulated by BK channels in GCs, was reported to be reduced in young abGCs (*Yang et al., 2015*), an observation we also found in our raw traces (*Figure 7B*, left) from *Mongiat et al. (2009)*. The parameters of the young abGC model were fitted best when we reduced the beta4-subunit associated BK current (gabk) by 100%. Thus, the abGC model predicts that the beta4 subunit is less expressed or not associated with BK channels in young abGCs. Future improvements of the abGC model should focus on more realistic simulations of details in voltage traces including fAHP kinetics, which are reproduced qualitatively but not quantitatively by the current model (*Figure 7B*).

To investigate the impact of the special intrinsic properties of young abGCs on their synaptic integration, we subjected both young and mature GC models to a broad range of synaptic input stimulation frequencies ranging from 10 to 75 Hz. In line with experimental data (*Mongiat et al., 2009*; *Pardi et al., 2015*), we found that diminished glutamatergic input onto abGCs was compensated by their enhanced excitability when GABAergic inhibition was absent. Both populations of GCs responded in a similar fashion over a wide range of stimuli, which is also in agreement with electrophysiological recordings (*Pardi et al., 2015*). Furthermore, despite their weaker excitatory input, we found that young abGCs were more efficiently activated by temporally separated (>15 ms) incoming activity from medial and lateral perforant path inputs as compared to mature GCs. Of note, in our study we did not model very young abGCs with reduced dendrite arborization but only focused on 4 weeks old abGCs that do not display any further significant alterations in their dendritic morphology (see our dendrite analyses and modeling in *Beining et al. (2017)*. However, T2N can be used to model also younger abGC at various ages. By coupling different phases of dendrite development to corresponding biophysical models, T2N may reveal the principles, which support the maintenance of structural and functional integrity of real or synthetic morphologies as they mature or change during pathology (see *Narayanan and Chattarji, 2010*; *Dhupia et al., 2014*; *Bozelos et al., 2015*; *Platschek et al., 2016*; *Platschek et al., 2017*).

Most existing GC models did not implement specific ion channels but instead used equations describing ion currents (A-, M-, T-type, L-type, N-type, delayed rectifier etc.) that had been measured in GCs, but which are formed by the combined action of several differently distributed ion channels in the real cell (e.g. $K_v1$ and $K_v4$ form the A-type current in GCs but are localized in the axon or dendrite, respectively). By incorporating the contributions of different ion channel isoforms, our model can be used to analyze and predict the impact of different channelopathies or compensatory ion channel adaptations onto the cell's active and passive behavior. This might be of special interest since specific isoforms dynamically control excitability (e.g. $K_v2.1$; *Misonou et al., 2005*) and alter their expression under pathological conditions such as epilepsy (e.g. $K_v1.1$; *Kirchheim et al., 2013*; or Kir2.1 and HCN; *Stegen et al., 2012*) or oxidative stress (e.g. $K_v4$; *Rüschenschmidt et al., 2006*). Indeed, our model was able to reproduce qualitatively the effects of a compensatory upregulation of Kir, HCN and Kv1.1 channels reported in TLE (*Stegen et al., 2012*; *Kirchheim et al., 2013*) demonstrating its predictive power there. Thus, the model might further be used to predict single or combined effects of other TLE-induced hippocampal alterations such as the reduction of BK channels in GCs (*Pacheco Otalora et al., 2008*), the aberrant connectivity (see review by *Sharma et al., 2007*; and network model by *Santhakumar et al., 2005*), as well as the impact of therapeutic gene transfer approaches, such as the transfer of the K2P leak channel TREK-1 to ameliorate status epilepticus (*Dey et al., 2014*).

As we did not investigate the entire parameter space of our electrophysiological models (especially in the young abGC model), for example using a genetic algorithm, we cannot exclude that a

different channel density distribution would result in a similarly robust and successful reproduction of experimental results (*Achard and De Schutter, 2006*). We took great care to compare the expression and subcellular distribution data of ion channels in immunohistochemical studies with different studies by other labs or with electrophysiological evidence (e.g. pharmacological blockade). However, previous work in other cell types and animals has shown that similar electrical behavior might arise from different combinations and parameters of ion channels (*Achard and De Schutter, 2006*; *Günay et al., 2008*; *Prinz et al., 2004*; for review see *Marder, 2011*; *Marder and Goaillard, 2006*).

## T2N limitations and future directions

So far, T2N is specialized on handling neuronal morphologies and neuronal models. As other cellular interactions, such as astrocytic-neuronal contacts are emerging to play an important role, e.g. for meta-plasticity (*Abraham and Bear, 1996*; *Abraham, 2008*), future versions of T2N should make it possible to reconstruct, build and simulate astrocytes and astrocyte-neuron interactions. The set of T2N functions can be extended to simplify modeling of new experimental settings including simulations of nonlinear synaptic integration or synaptic and intrinsic plasticity as well as structural dendritic plasticity such as dendritic retraction or pruning of dendritic segments (*Beining et al., 2017*; *Platschek et al., 2017*). Including stochastic sampling algorithms would make T2N suitable to study degeneracy by supporting a search for distinct combinations of morphological and biophysical properties generating similar physiological outcomes.

## GC modeling and degeneracy

With our newly developed *T2N*, we were able to create a novel compartmental model of mature and adult-born mouse and mature rat GCs that is biologically and physiologically consistent. Therefore, it is of high predictive value for studies on the single-cell and network behavior of mature GCs and young abGCs, as well as under pathological conditions of synaptic, morphological or physiological alterations of GCs. As compared to more standardized methods with automated parameter fitting such as those used in the Allen Brain Project or Blue Brain Project (*Druckmann et al., 2007Druckmann et al., 2007*; *Hay et al., 2011*; *Markram et al., 2015*; *Shai et al., 2015*; *Van Geit et al., 2016*), our model resulted from a more traditional approach of incorporating as much biological data as possible. Nevertheless, our model satisfies the objective constraints from experiments (*Table 2*) and is robust to experimentally verifiable manipulations. We do not criticize automated parameter fitting, which is of great value. Our main point is the emphasis on using many different morphologies in combination with carefully, biologically constrained ion channel models. While one reason for the particular robustness of our model comes from the modeling approach using *T2N*, it is likely that the complete GC ion channel set offers redundancy and stability with respect to differences such as in morphology or species. Therefore, it will be interesting to further investigate whether the redundancy introduced by the set of existing ion channels is responsible for the robustness to morphological modifications in our GC model. Thus, our biophysical and morphological model provides a basis for future studies determining how cell-to-cell and animal-to-animal variability of ion channel expression combined with morphological and synaptic variability affects the robustness of GC passive and active behavior.

Our model is available on the ModelDB public database (http://senselab.med.yale.edu/ModelDB/default.asp) and can now be used to address the exciting question whether ion channel degeneracy in GCs exists in terms of compensatory interactions between multiple ion channels (*Drion et al., 2015*) and how it contributes to the homeostasis of GC function. Future work should also address the question whether variation of some other biological factors besides morphology or in addition to it would also lead to robust GC simulation results. As mentioned above, for this purpose, T2N could be extended by incorporating stochastic search algorithms (*Foster et al., 1993*; *Goldman et al., 2001*; *Weaver and Wearne, 2008*; *Rathour and Narayanan, 2014*; *Mishra and Narayanan, 2017*), which would allow users to generate multiple randomized models with different biophysical and morphological parameters leading to similar electrophysiological behavior. This would help to reveal which combinations of channel properties and dendritic arborization support robustness of GC function. A recent study (*Mishra and Narayanan, 2017*) addressed this issue using a large number of GC models with variable channel parameters in reduced morphologies. It would be interesting to employ a similar approach and use *T2N* and our new GC model with the updated

layer-specific composition of ionic channels to stochastically generate many biophysically distinct GC models with variable location-dependent channel expression as well as variable full dendritic morphologies (*Schneider et al., 2014*). Simulations and analyses in such large collections of detailed conductance-based GC models might contribute to the identification of subcellular mechanisms of degeneracy. Specifically, this approach would show whether disparate dendrite and channel parameters, including dendrite length and branching as well as gradients in channel densities, kinetics, voltage-dependence or intracellular milieu may lead to identical GC electrophysiology. *T2N* will be useful to generalize such analyses also to other cell types.

## Materials and methods

### Compartmental modeling with T2N

Compartmental modeling was done in the *NEURON* (*Carnevale and Hines, 2006*) environment (V7.4) controlled and run using our novel *T2N* interface. *T2N* was written as an extension of the freely-available *TREES toolbox* (*Cuntz et al., 2010*, *2011*) providing an interface between *Matlab* (Mathworks) and *NEURON*. It was developed on *Matlab* 2015b and it is recommended to use *Matlab* 2015b or higher. All ion channels, point processes, connections, morphologies and *NEURON* settings are directly set in a well-defined *Matlab* structure. For any morphology-related settings or manipulations, *T2N* uses the set of *TREES toolbox* functions (e.g. to create and handle reconstructed and synthetic dendritic morphologies). Neuronal morphologies including precise node locations are automatically translated into *NEURON* sections and segments. Multiple *NEURON* simulations (e.g. to simulate several cells or to create an f-I relationship) can be run in parallel as *T2N* is able to start separate *NEURON* instances on different cores, thus reducing simulation time. Even more important, T2N allows the use of the parallel NEURON environment (*Migliore et al., 2006*; *Hines and Carnevale, 2008*), thus drastically increasing performance of large-scale networks by distributing cells of single NEURON simulations on multiple cores using a round robin approach. Recorded variables are returned to *Matlab* in a well-ordered structure for further analysis. For more information, see the *T2N* manual, which is provided with the code (see Data sharing).

### Data analysis and visualization

The programming environment *Matlab* (Mathworks, version 2015b, some functions of the GC model are not functional in earlier Matlab versions) was used together with the *TREES toolbox* to analyze raw electrophysiological data from *Mongiat et al. (2009)*, as well as the output of the compartmental modeling simulations. Electrophysiological properties were measured as following: The input resistance $R_{in}$ was measured using the steady-state current during a depolarizing 10 mV voltage step (200 ms long, from a holding potential of −92.1 mV). In the raw data from *Mongiat et al. (2009)*, on which we fitted most of the active properties of our compartmental model, a liquid junction potential (LJP) of 12.1 mV existed for which we corrected the voltage traces and voltage commands. Hence, to compare Kir conductance in the raw data and the model as performed in *Mongiat et al. (2009)*, we calculated the slope conductance at hyperpolarized values (−152.1 to −122.1 mV) and subtracted the slope conductance at a potential range where Kir channels are largely closed (−82.1 to −62.1 mV). The cell capacitance was obtained from a −10 mV voltage step as the integral of the measured current (steady-state current subtracted) divided by the amplitude of the voltage step. The membrane time constant was measured as the exponential voltage decay following a 500 ms long hyperpolarizing current step (10 pA) from a holding potential of −80 mV. All action potential (AP) property measurements were done on the first AP of a 200 ms long 90 pA current step (from a −80 mV holding potential). The voltage threshold of an action potential (AP) was defined as the point when the voltage slope exceeded 15 mV/ms. The rheobase was the current step (5 mV intervals, 200 ms current step from −80 mV holding potential) at which the first AP occurred. The AP amplitude was defined as the difference between the absolute AP amplitude and the AP voltage threshold and AP width was the half-maximum width of this amplitude. The fast afterhyperpolarization potential (fAHP) was calculated as the difference between the voltage threshold and the minimum voltage between two consecutive APs, provided that the time difference between the voltage minimum and voltage threshold was less than 5 ms (larger intervals were assumed to be medium AHPs). The interspike interval (ISI) was the delay between two consecutive AP maxima, whereas the

ISI adaptation ratio was defined as one minus the first divided by the last ISI. The backpropagating AP (bAP) amplitude was the maximal amplitude at a specific dendritic location during an AP elicited at the soma and the bAP delay was the time delay between the somatic and the dendritic voltage maximum. The dendritic or axonal velocity was the inverse of this delay times the path distance between the soma and the dendritic/axonal location. The $Ca^{2+}$ amplitude was the local maximal amplitude following an AP elicited at the soma. The $Ca^{2+}$ decay time constant was obtained by fitting a biexponential curve to the $Ca^{2+}$ decay curve that followed an AP and calculating the weighted sum of the two time constants, as has been done in *Stocca et al. (2008)*.

Individual figure panels throughout the manuscript were generated with *Matlab* and combined in *Adobe Illustrator CS6*.

## Data sharing

All compartmental models along with all simulation protocols that have been performed in this study as well as the T2N software are available on the TREES homepage (http://www.treestoolbox.org/T2N.html) and on the ModelDB public database (http://senselab.med.yale.edu/ModelDB, accession # 231862. We also uploaded a pure NEURON version of the GC model including all morphologies and biophysics but only two simulation protocols on ModelDB under the accession # 231818)

## Models of mouse and rat mature GCs and young mouse abGCs

See Appendix 2, *Tables 1* and *3* for details of reconstructed and synthetic GC morphologies and ion channel properties used in compartmental simulations of GCs.

## Acknowledgements

We would like to thank J Kasper and S Krischok for performing preliminary analyses and A Bird, A Castro, LH Deters, M Mittag and H Röhr for testing the T2N software. Further we thank T Deller for useful discussions and continuous support, as well as the reviewers for comprehensive and detailed reviews leading to the improvement of the manuscript.

## Additional information

### Funding

| Funder | Grant reference number | Author |
| --- | --- | --- |
| Deutsche Forschungsgemeinschaft | CRC1080 | Stephan Wolfgang Schwarzacher |
| Bundesministerium für Bildung und Forschung | 01GQ1406 | Hermann Cuntz |
| Alzheimer Forschung Initiative | 15038 | Peter Jedlicka |
| Bundesministerium für Bildung und Forschung | 01GQ1203A | Peter Jedlicka |
| Agencia Nacional de Promoción Científica y Tecnológica | PICT2013-2056 | Lucas Alberto Mongiat |
| Deutsche Forschungsgemeinschaft | JE 528/6-1 | Peter Jedlicka |

The funders had no role in study design, data collection and interpretation, or the decision to submit the work for publication.

### Author contributions

Marcel Beining, Conceptualization, Resources, Data curation, Software, Formal analysis, Validation, Investigation, Visualization, Methodology, Writing—original draft, Project administration, Writing—review and editing; Lucas Alberto Mongiat, Resources, Data curation, Funding acquisition, Validation, Writing—review and editing; Stephan Wolfgang Schwarzacher, Conceptualization, Resources, Supervision, Funding acquisition, Validation, Project administration, Writing—review and editing;

Hermann Cuntz, Conceptualization, Resources, Supervision, Funding acquisition, Validation, Visualization, Methodology, Writing—original draft, Project administration, Writing—review and editing; Peter Jedlicka, Conceptualization, Resources, Data curation, Supervision, Funding acquisition, Validation, Visualization, Methodology, Writing—original draft, Project administration, Writing—review and editing

Author ORCIDs
Marcel Beining  http://orcid.org/0000-0002-6577-2648
Hermann Cuntz  http://orcid.org/0000-0001-5445-0507
Peter Jedlicka  http://orcid.org/0000-0001-6571-5742

Decision letter and Author response
Decision letter https://doi.org/10.7554/eLife.26517.029
Author response https://doi.org/10.7554/eLife.26517.030

## Additional files

### Supplementary files
• Supplementary file 1. Analogous to *Table 2*, this table compares electrophysiological properties of experimental data and simulations performed with the biophysical model of Aradi and Holmes (*Aradi and Holmes, 1999*) and reconstructed (middle column) or synthetic (right column) rat morphologies.
DOI: https://doi.org/10.7554/eLife.26517.021

• Transparent reporting form
DOI: https://doi.org/10.7554/eLife.26517.022

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

## Appendix 1

DOI: https://doi.org/10.7554/eLife.26517.023

## T2N Tutorials

This appendix lists all tutorials that are currently available for T2N. The tutorials are in the form of a Matlab live script (.mlx). If you cannot open Matlab live scripts (e.g. having no Matlab GUI or using a Matlab version below R2016a), please open the corresponding .m script files in the same folder and follow the comments/sections in the script.

To download T2N including the TREES toolbox and the GC model, go to https://senselab.med.yale.edu/modeldb/ShowModel.cshtml?model=231862

T2N requires NEURON to be installed (http://www.neuron.yale.edu/neuron/download).

After downloading and extracting the 'GC Model - full.zip', please run the 'runthisAfterUnzip.m' script file located in the main folder to automatically add all files to the Matlab search path.

Note for Linux/Mac users: Matlab has to be run from a Terminal for T2N to work properly.

### Tutorial 1 – How to distribute ion channels on a TREES morphology using T2N

Please open the Matlab live script located in '*T2Nfolder*/Tutorials/t2n_Tutorial_live.mlx' and follow section 'Tutorial A – initialize parameters and the neuron structure'. An even more detailed tutorial on how to distribute mechanism on morphologies can be found in the Matlab live script located in '*T2Nfolder*/Tutorials/t2n_distributeMechanisms_live.mlx'

### Tutorial 2 – How to load a morphology

Please open the Matlab live script located in '*T2Nfolder*/Tutorials/t2n_Tutorial_live.mlx' and follow section 'Tutorial B – loading a morphology' (after considering the initial note of the tutorial script). The last step ('t2n_writeTrees') converts the TREES toolbox morphology into a hoc template file which will be used by T2N in NEURON later (and of course could also be used in NEURON without T2N).

### Tutorial 3 – How to define a simulation (here somatic current injection)

Please open the Matlab live script located in '*T2Nfolder*/Tutorials/t2n_Tutorial_live.mlx' and follow section 'Tutorial C – simulation protocol: somatic current injection' (after considering the initial note of the tutorial script)

### Tutorial 4 – How to do a f-I relationship simulation run

Please open the Matlab live script located in '*T2Nfolder*/Tutorials/t2n_Tutorial_live.mlx' and follow section 'Tutorial D - simulation protocol: several simulations with different injected current amplitudes (f-I relationship)' (after considering the initial note of the tutorial script)

### Tutorial 5 – How to do a I-V relationship simulation run

Please open the Matlab live script located in '*T2Nfolder*/Tutorials/t2n_Tutorial_live.mlx' and follow section 'Tutorial E - simulation protocol: several voltage clamp steps (I-V relationship)' (after considering the initial note of the tutorial script)

## Tutorial 6 – How to map a bAP onto a morphology and plot its distance-dependence

Please open the Matlab live script located in '*T2Nfolder*/Tutorials/t2n_Tutorial_live.mlx' and follow section 'Tutorial F - Map the backpropagating AP onto the tree and plot its distance-dependence' (after considering the initial note of the tutorial script)

## Tutorial 7 – How to evaluate Ca$^{2+}$ dynamics in different compartments of a model

This is not part of the general T2N tutorial, as different models might comprise different Calcium buffer models with different parameter names. However, function aGC_CaDyn and aGC_plotCaDyn of the GC model folder does the job. Section 'Ca$^{2+}$ dynamics *Figure 5*, mimicking *Stocca et al. (2008)*' in GC_experiments.m of the GC model folder is an example on how to use these two functions.

## Tutorial 8 – How to systematically modify parameters

Please open the Matlab live script located in '*T2Nfolder*/Tutorials/t2n_Tutorial_live.mlx' and follow section 'Tutorial G - Do a parameter scan or how to modify mechanism parameters' (after considering the initial note of the tutorial script)

## Tutorial 9 – How to set up synapses, connections and networks and how to use the parallel NEURON feature

Please open the Matlab live script located in '*T2Nfolder*/Tutorials/t2n_Tutorial_live.mlx' and follow section 'Tutorial H - Synaptic stimulation, simple Alpha synapse', 'Tutorial I - Synaptic stimulation, Exp2Syn synapse and a NetStim' and 'Tutorial J - Small network with Poissonian input' (after considering the initial note of the tutorial script)

## Appendix 2

DOI: https://doi.org/10.7554/eLife.26517.024

# Supplementary information on the new GC compartmental model built with T2N

## Context

Poor results using compartmental models under conditions they were not tuned for (*Almog and Korngreen, 2016*) pose a general problem, including current models of dentate granule cells (GCs). Due to the central role of GCs in the hippocampal circuit, e.g. their function in transforming input information into sparse code (*Jung and McNaughton, 1993*), attempts to construct compartmental models of mature GCs have been made early on (*Yuen and Durand, 1991*; *Aradi and Holmes, 1999*). In the meantime, many GC models were generated and have been modified and extended multiple times (*Chiang et al., 2012*; *Ferrante et al., 2009*; *Jaffe et al., 2011*; *Jedlicka et al., 2015*; *Krueppel et al., 2011*; *Mateos-Aparicio et al., 2014*; *Platschek et al., 2016*; *Tejada et al., 2012*). These single-cell GC models have been instrumental for studying the large-scale network of the dentate gyrus (DG), for example, to analyze the origins of DG hyperexcitability in temporal lobe epilepsy (*Santhakumar et al., 2005*; *Dyhrfjeld-Johnsen et al., 2007*; *Vlachos et al., 2012*; *Tejada and Roque, 2014*; *Hendrickson et al., 2015*; see also *Winkels et al., 2009*; *Jedlicka et al., 2010*; *Jedlicka et al., 2011*). However, these models were based on the very first GC models, when simplified morphologies were used, detailed knowledge of the ion channel distribution and kinetics was lacking and missing parameters were often adopted from other cell types or species. (Of note, some implementations of the Aradi and Holmes channels in previous GC models contained bugs so readers should consult the readme.html files in ModelDB models with accession numbers 124513 and 185355 to find corrections and avoid using the bugs unknowingly.) Probably due to these factors, more recent GC models needed to be modified or created *ad hoc* to reproduce single experiments (*Jaffe et al., 2011*; *Chiang et al., 2012*; *Stegen et al., 2012*; *Mateos-Aparicio et al., 2014*; *Yim et al., 2015*; *Platschek et al., 2016*).

# Details of GC simulation results

## New comprehensive up-to-date collection of ion channels

We assembled a set of GC ion channels and their compartment-specific distributions that have been reliably characterized experimentally by immunohistochemical labeling with light or electron microscopy as well as electrophysiology and modeling (*Table 1* and *Figure 2A*, see below for a full description of all ion channels). To establish the biophysical model, corresponding ion channel models were obtained from literature or developed based on known channel kinetics and incorporated in reconstructed morphologies of eight mature mouse GCs (*Figure 2A*; *Schmidt-Hieber et al., 2007*). To achieve realistic activation of $Ca^{2+}$-dependent channels, we implemented a phenomenological $Ca^{2+}$ buffer model (see Materials and methods) that reproduced experimentally measured concentrations and kinetics of free $Ca^{2+}$ at different subcellular levels as well as measured GC $Ca^{2+}$ currents (*Eliot and Johnston, 1994*; *Jackson and Redman, 2003*; *Stocca et al., 2008*).

## I–V curves, passive properties, spiking behavior in the mouse mature GC model

The density of the channels, which are open at resting potential (the leak and the inward-rectifying Kir2 channel) were fitted to the qualitatively assessed channel expression pattern (see *Table 1*) and then fine-tuned by hand to fit experimentally measured steady-state currents in mature GCs during voltage clamp steps from −130 to −60 mV (I-V curve, *Figure 3A*) (*Mongiat et al., 2009*). We further used I–V curves obtained after application of

200 μM $BaCl_2$ to the extracellular medium to block Kir channels and thereby estimate their contribution to currents activated around resting potential (*Figure 3A*). Interestingly, a further block of the passive channel by 30% was necessary in the $Ba^{2+}$ simulations to match the data. This was consistent with the observed moderate $Ba^{2+}$ sensitivity in K2P channels (*Lesage et al., 1997*; *Meadows et al., 2000*; *Goldstein et al., 2005*; *Ma et al., 2011*). Of note, $Ba^{2+}$ is a relatively nonspecific blocker of ion channels. At 200 μM, $BaCl_2$ is also known to block A-type K+ channels (*Gasparini et al., 2007*; *Losonczy et al., 2008*), which were shown to alter the input resistance ($R_{in}$) in CA1 pyramidal cells (*Kim et al., 2005*). Since GCs also express Kv4 and Kv1.4 channels, it is possible that the A-type K+ channels contribute to $R_{in}$. The contribution of Kv4 and Kv1.4 channels to $R_{in}$ was low in our model (see sensitivity matrix in *Figure 6* and supplement) so they do not seem necessary to explain the $Ba^{2+}$ experiment, but see our discussion on degeneracy.

The specific membrane capacitance $c_m$ was fitted to reproduce subthreshold properties such as the membrane time constant and capacitance (*Table 2*). The input resistance ($R_{in}$) was consistent with electrophysiological measurements from literature (*Brenner et al., 2005*; *Schmidt-Hieber et al., 2008*; *Mongiat et al., 2009*). By considering the current or voltage steps applied in each of these studies (*Figure 3A*, colored arrows), we could show that due to the inward rectification, the measured $R_{in}$ could vary substantially depending on the holding voltage and the current or voltage step that was applied. Interestingly, in electrophysiological traces, the current dynamics of the cells showed a slowly activating outward current at holding voltages below −100 mV (*Figure 3—figure supplement 3*). This current was reproduced by our GC model as Kir channels were partly blocked at resting potential and had a slow recovery time when being unblocked by hyperpolarization.

In order to fit the spiking behavior of our mature GC model, we used raw traces from current clamp measurements of eight mature GCs (*Mongiat et al., 2009*) to reproduce action potential (AP) shape and spiking properties in detail. For this, the densities of all active ion channels were matched to the qualitatively assessed channel expression pattern (see *Table 1*) and then fine-tuned by hand. The spiking frequency vs. current (F–I) curve, relating somatic current injections to the amount of elicited APs is an important measure of a neuron's excitability, and therefore a crucial feature to be replicated. Our active model was able to reproduce the F–I curve in both conditions, control and pharmacological blockade of Kir channels with $BaCl_2$ (*Figure 3B*, experiment: gray lines on the left; model: blue lines in the middle column). The spiking behavior closely matched the experimental data (*Figure 3C*), which was validated by comparing the AP properties such as the AP width, voltage threshold, amplitude, fast afterhyperpolarization (AHP) as well as the voltage phase plot (*Figure 3D–F*). Importantly, replacing the reconstructed dendritic arborization with synthetic mouse morphologies (leaving all biophysical properties untouched), produced similar electrophysiological results (*Figure 3A–F* right column, model data shown in green) indicating a strong robustness of our model against morphological changes. Of note, it was not necessary to refit the passive properties for the synthetic morphologies. This further renders our morphological mouse model suitable for large-scale network modeling of the DG. Interestingly, the different morphologies exhibited different variants, e.g. of AHPs and depolarizing afterpotentials (DAPs), similarly to what was observed in experiments (*Althaus et al., 2015*). We analyzed the morphological origin of the electrophysiological variability and found a moderate negative correlation between dendritic surface and the cell's rheobase (correlation coefficient R = −0.42) or AP number (correlation coefficient R = −0.40) indicating reduced excitability in larger dendritic trees (*Figure 3—figure supplement 2*, left). This is consistent with previous experimental and modeling studies showing that larger dendritic trees exhibit reduced excitability due to the larger dendritic leak (*Krichmar et al., 2002*; *Šišková et al., 2014*; *Platschek et al., 2016*). The size of the soma was also of relevance since it shaped AP characteristics such as fast AHP (correlation coefficient R = −0.33) and AP width (correlation coefficient R = −0.45) in the GC models (*Figure 3—figure supplement 2*, right). This suggests that some of the electrophysiological variability might be accounted for by the variability of GC morphologies.

## Mature rat GC model

Since many GC experiments are performed in rats instead of mice, we further developed a rat GC model. In order to reproduce mature rat GC electrophysiology, we replaced the mouse morphologies with reconstructions of mature rat GCs (*Figure 4A*, left) as well as synthetic rat morphologies generated by our morphological model (*Figure 4A*, right), both published previously by our lab (*Beining et al., 2017*). Rat GCs have generally longer dendrites but only marginally higher total dendritic length since they are less branched than mouse GCs (*Beining et al., 2017*). Furthermore, rat GCs have a larger mean dendritic diameter. However, even with the altered morphologies, experimental rat GC $R_{in}$ measurements (average $218 \pm 30$ MΩ, *Figure 4B*, colored arrows) were only matched when increasing the Kir conductance by a factor of x2.5 ($204 \pm 16$ MΩ, solid blue curve, *Figure 4B*) while keeping other channel conductances equal to the values from the mature mouse GC model (*Table 1*). In contrast, the match was poor when all channel densities were kept as in the mouse model ($313 \pm 21$ MΩ dashed line, *Figure 4B*). Also, when the Kir conductance was not adjusted, ongoing spiking failed at high-current injections due to insufficient leak, which led to Nav channel inactivation (depolarization block). However, with the increased Kir channel conductance, our rat GC model reproduced the F–I curve reported from a study (*Pourbadie et al., 2015*), for which similar intracellular solutions were used as for the mouse GC electrophysiology (*Mongiat et al., 2009*; *Figure 4C*). These results indicate that a higher expression of Kir channels might account for species differences in passive and active electrophysiological behavior.

## bAP attenuation and Ca$^{2+}$ dynamics in rat and mouse GC models

To further validate our mouse and rat active models of GCs, we next investigated dendritic signal propagation and Ca$^{2+}$ signaling. Speed and attenuation of backpropagating APs (bAPs) that were previously measured in rats at various distances from the soma (*Krueppel et al., 2011*) were matched by our rat GC model (*Figure 5A*, left, blue curves) with a relative attenuation at 185 µm of $24.0 \pm 2.8\%$, compared to $24.5 \pm 3.6\%$ in experiments (*Figure 5A*, left, black dots). Since we did not implement voltage-gated sodium (Na$_v$) channels in the dendritic region of our morphologies, our simulations strengthen the notion that, as suggested previously (*Krueppel et al., 2011*), GC dendrites are virtually void of Na$_v$ channels. This is particularly important since earlier GC models comprised significant densities of dendritic Na$_v$ channels (*Aradi and Holmes, 1999*; *Santhakumar et al., 2005*; *Schmidt-Hieber and Bischofberger, 2010*; *Mateos-Aparicio et al., 2014*) thereby strongly affecting synaptic integration and calcium- or voltage-dependent plasticity. Although our modeling results indicate that GC dendrites do not express any Na$_v$ channels we must point out that there could be other possible solutions that fit the data. Indeed, pharmacological experiments (local application of TTX to block Na$_v$ channels in GC dendrites) combined with dual somatodendritic recordings and modeling (*Krueppel et al., 2011*) revealed that GCs may possess low densities of Na$^+$ channels (see their Figures 2 and 3). The delay between the somatic AP and the maximum bAP at distal dendrites could not be well reproduced without adjusting the specific axial resistance $R_a$ and the passive membrane conductance to the higher temperature of 33°C that was used in experiments by Krueppel and colleagues (*Figure 5B*, left) with Q10 values taken from *Trevelyan and Jack (2002)*. This indicates that faster dendritic voltage propagation at higher temperatures should be considered in models that simulate in vivo conditions. Applying the same protocols to the mouse GC model, for which no experimental data exist so far, revealed a significantly lower relative bAP attenuation of $35.6 \pm 3.2\%$ at 185 µm (p=0.028, Kruskal-Wallis test, n = 5 rat GCs vs. n = 8 mouse GCs) but a similar delay ($1.81 \pm 0.08$ ms in mouse vs. $1.68 \pm 0.05$ ms at 185 µm, p=0.188, Kruskal-Wallis test, n = 5 rat GCs vs. n = 8 mouse GCs) compared to the rat model (*Figure 5A,B*, right). This predicts species-specific differences in bAP attenuation that remain to be examined experimentally. As Ca$^{2+}$ is an important cellular signal, we also analyzed the bAP-induced Ca$^{2+}$ peak and decay in various compartments (axon, soma, proximal and distal dendrite) in the

mouse and rat model and compared it to known experimental data. The $Ca^{2+}$ peak levels were found to be slightly but not significantly higher in mouse (n = 8) than in rat (n = 5) dendrites (*Figure 5C*, proximal: 216.4 ± 16.5 µM in mouse vs. 152.0 ± 39.4 µM in rat, p=0.305 Kruskal-Wallis test; distal: 102.6 ± 12.1 µM in mouse vs 76.8 ± 11.9 µM in rat, p=0.305, Kruskal-Wallis test).

Of note, in contrast to our biophysical model, classical GC ion channels (*Aradi and Holmes, 1999*) used in most previous GC models were not able to generate realistic output when used in multiple mouse or rat morphologies (*Figure 3—figure supplement 1*, *Figure 4—figure supplement 1*, *Figure 5—figure supplement 1*, *Supplementary file 1*). This indicates the importance of using detailed data-driven ion channel composition, realistic $Ca^{2+}$ buffer models, diverse dendritic trees as well as multiple different electrophysiology experiments for tuning and generating compartmental models, which would be stable across many morphologies and conditions.

## Sensitivity analysis in mature rat GC model

Our GC model is biophysically highly realistic because it comprises exclusively ion channel isoforms that were demonstrated to exist in GCs. Therefore, it can be useful for making novel predictions and to identify parameters that critically influence neural computation in GCs. *Figure 6A* shows a sensitivity matrix in which the relative effects on physiological features of GCs were measured when ion channel conductances or other model parameters were up- or downregulated. A number of predictions can be drawn from this matrix. Reducing the passive leak and Kir2.1 channels critically influenced $R_{in}$, the membrane time constant, the interspike interval (ISI), and the number of APs at 100 pA. In contrast, the number of APs at 200 pA was only marginally affected, indicating that as the membrane is steadily depolarized at high current injections other channels such as $K_v2.1$ and $K_v7$ dominate the hyperpolarizing potassium influx, as Kir2.1 channels close. The 8-state sodium channel model (na8st) which represents $Na_v1.2$ and 1.6 was critically involved in excitability, AP shape and voltage threshold as well as in the propagation speed along the dendrites and axon. Furthermore, $Na_v$ channels, $Na^+$ concentration and temperature (besides anatomical properties such as diameter and branching) were the only components influencing propagation speed. Interestingly, $K_v2.1$ (delayed rectifier current) blockade reduced ISIs but increased ISI adaptation, as the higher number of APs recruited more $K_v7$ channels. The blockade of $K_v7$ shows that this channel gains relevance as membrane depolarizes, contributing more and more to the slow AHP. As expected, altering the reversal potential of the $Na^+$ and $K^+$ had complementary effects: A 10 mV increase of $E_K$ (from −90 to −80 mV) influenced mainly the passive properties such as $R_{in}$ and excitability, whereas a reduction of $E_{Na}$ by −20 mV reduced AP amplitude (and propagation along the axon) and therefore increased AP width due to reduced activation of repolarizing K-channels. Interestingly, corresponding increases (doubling) of each model parameter led to opposite effects (*Figure 6—figure supplement 1*), except that doubling $C_m$ partly showed paradoxical results arising from its effect on the dynamics of activation of dendritic channels.

Even though only five channels in our model had been modeled as being temperature-sensitive ($Ca_v3.2$, SK, $K_v4.2$, $K_v7.2/3$ and HCN), increasing or decreasing the temperature by 10° C from 24° C to 34° C or to 14° C had a significant influence on the cell's excitability and spiking behavior (34° C: *Figure 6A*; 14°C: *Figure 6—figure supplement 1*). From these predictions, we conclude that, as different recording temperatures are used in different studies, the influence of temperature on the contribution of specific channels to electrophysiological properties, for example, to the slow AHP, can therefore be quite important and should be considered.

## AP repolarization and spike adaptation in mature rat GCs

Finally, we tested some relationships of ion channels and AP features suggested previously in experiments. Firstly, we tested whether our model could reproduce the experimentally

documented dominant contribution of BK and $K_v3$ channels to AP repolarization in rat GCs (*Riazanski et al., 2001*; *Brenner et al., 2005*) with BK governing somatic (*Müller et al., 2007*) and $K_v3$ axonic repolarization (*Alle et al., 2011*). To this goal, we conducted simulations with low (90 pA) and high (250 pA) current injections to analyze the contribution of BK and $K_v3$ channels to AP repolarization. We found that $K_v3$ blocking had no major effects on somatic AP shape, whereas a blockade of BK channels increased AP width, especially when the cell had to elicit several short-interval APs (*Figures 6B*, 250 pA). For both stimulation intensities, the relative contribution of $K_v3.4$ and BK stayed the same in the model with BK contributing considerably more to somatic repolarization, consistent with literature (data from rat; *Müller et al., 2007*; *Riazanski et al., 2001*; compared with our rat GC model in *Figure 6B*).

Secondly, we found that in our GC model $K_v7$ (M-current) contributed to spiking adaptation whereas calcium-dependent SK channels did not (*Figure 6C*). This is different from experimental data measured in rat GCs (*Mateos-Aparicio et al., 2014*). This discrepancy might originate from the manually adjusted size of injected current during spiking adaptation measurements under different pharmacological blockers (*Mateos-Aparicio et al., 2014*). Another explanation might be the usage of a different charge-carrying anion: our data is retrieved from *Mongiat et al. (2009)*, who used Gluconate vs. $MeSO_4$ in their experiment, which is known to induce larger slow AHPs (*Zhang et al., 1994*) and to slowly reduce $R_{in}$ by up to 70% (*Kaczorowski et al., 2007*). In addition, this discrepancy might also arise from the putative partial permeability of SK for $Na^+$ which would drastically increase $E_{Na}$ to ~50 mV (*Shin et al., 2005*), or from the very complex interaction between $K_v7$ channels, KATP channels and neuronal $Ca^{2+}$ sensors such as hippocalcin (*Andrade et al., 2012*) of which all are further affected by temperature (note the 10°C higher temperature in *Mateos-Aparicio et al., 2014*), phosphorylation (*Andrade et al., 2012*) and ATP (*Baukrowitz and Fakler, 2000*).

Taken together, our sensitivity mapping of ion channel changes opens a number of insights that help understand the contribution of each ion channel isoform to the electrophysiological behavior of GCs (but see our discussion on degeneracy).

## Compensatory ion channel alterations in GCs during temporal lobe epilepsy

To test the predictive power of our model under pathological changes in the GC ion channel composition we analyzed the impact of ion channel changes reported to occur during temporal lobe epilepsy (TLE) (*Bender et al., 2003*; *Young et al., 2009*; *Stegen et al., 2012*; *Kirchheim et al., 2013*). Overexpression of HCN and Kir (by doubling the channel densities in the model) reduced the intrinsic excitability of our model rat GCs by decreasing $R_{in}$ (133 ± 11 MΩ vs. 204 ± 16 MΩ, *Figure 6D*). This was consistent with patch-clamp recordings from murine and human GCs in TLE revealing a compensatory reduction of GC excitability due to protective enhanced expression of HCN and Kir channels (*Young et al., 2009*; *Stegen et al., 2012*). In addition, in line with these experiments, our normal GC model showed a very low resonant behavior when oscillating currents between 1 and 15 Hz were injected (*Figure 6— figure supplement 2*), indicating that the low HCN channel density in GCs under normal conditions is not sufficient to induce resonant behavior as observed in CA1 pyramidal cells (*Stegen et al., 2012*). Similarly, we further showed that protective overexpression of $K_v1.1$ found in mouse models of TLE increased spike delays and decreased GC excitability in our mature mouse GC model (*Figure 6E*), although the spike delay effect in our model was not as prominent as in experiments (*Kirchheim et al., 2013*). This discrepancy could be explained by the use of different charge-carrying anion, as explained in the spike frequency adaptation experiments (Gluconate in the experiments used to fit our model vs. $MeSO_4$ in the experiment, see above). Thus, our model appears useful to analyze GC spiking behavior following compensatory channel regulation due to pathological conditions such as TLE and thus could be integrated in available dentate network models that aim to model such conditions (*Yim et al., 2015*).

## Young adult-born GC model

Dentate GCs are continuously produced throughout life, a process called adult neurogenesis. Young adult-born GCs (abGCs) possess unique electrophysiological features and numerous studies have pointed out their special role in hippocampal memory formation (*Ming and Song, 2011*; *Aimone et al., 2014*; *Johnston et al., 2016*). In particular, young abGCs display a critical phase starting at about 4 weeks of cell age when they exhibit increased excitability, enhanced synaptic and dendritic plasticity, and receive less inhibitory input (*Ge et al., 2007*; *Mongiat et al., 2009*; *Marín-Burgin et al., 2012*; *Bergami et al., 2015*; *Temprana et al., 2015*; *Beining et al., 2017*). As no compartmental model of young abGCs exists so far, we aimed to investigate the capability of our biophysical mature GC model to replicate the electrophysiology of young abGCs when differences in channel expression are considered and introduced by T2N.

In order to fit the passive and active properties of the young abGC model, we used raw voltage and current traces that have been acquired under the same conditions as in the mature mouse GC experiments (*Mongiat et al., 2009*). The I-V relationship and passive properties were well matched by reducing the Kir2 channel density by 73% (*Figure 7A*). This was consistent with previously reported reduced Kir channel currents and increased excitability in young abGCs (*Mongiat et al., 2009*). Since other channels have not been studied in young abGCs so far, altered channel expression for our abGC model was inspired from channel distribution studies in postnatal developing GCs (*Table 3*) as their development was reported to be similar to that of abGCs (*Laplagne et al., 2006*; *Urbán and Guillemot, 2014*).

Since young abGCs have a much lower membrane capacitance consistent with a lower spine density at 28 dpi (*Mongiat et al., 2009*; *Yang et al., 2015*), we reduced the spine scaling factor (see Materials and methods) by a factor of x0.3 compared to the mature GC model. Even with this low value, the membrane capacitance of our model (46.95 ± 3.21 pF) did not match the experimental values of 30.6 ± 1.0 pF (*Mongiat et al., 2009*). This resulted in longer spike delays compared to exemplary spike traces (compare columns in *Figure 7B*). However, the acute slice recordings might have slightly underestimated the capacitance values due to ineffective voltage clamping of distal dendritic regions or due to dendritic branches that were cut during slice preparation. Moreover, other studies showed capacitances for young abGCs of around 40 pF (*Piatti et al., 2011*; *Yang et al., 2015*). Despite the differences in capacitance, our young abGC model was capable of reproducing the experimental F–I relationship (*Figure 7C*), AP characteristics (*Figure 7D–E*), and AP dynamics (*Figure 7F*) when spine densities and expression of ion channels were reduced as mentioned above.

## Synaptic signal integration in young abGCs

Young abGCs have been attributed a special role in hippocampal pattern separation and integration due to their broader tuning to input activity (see reviews *Aimone et al., 2010*; *Rangel et al., 2013*; *Johnston et al., 2016*). Therefore, to further test our model and to generate quantitative predictions concerning synaptic integration of abGCs, we compared the temporal processing of synaptic inputs in the young abGC and the mature GC model considering only intrinsic differences (i.e. no additional inhibitory input). Using T2N, we first randomly and equally distributed 30 excitatory synapses over the middle (MML) and outer molecular layer (OML), the termination side of the major afferent input from the medial and lateral entorhinal cortex, respectively. This number was chosen to reliably drive GCs at theta frequency without saturating them and is comparable to literature, since *Krueppel et al. (2011)* estimated that about 55 synchronously active distal synapses are necessary to drive a dentate GC in rat. Next, we synaptically activated the GCs synchronously at different frequencies (*Figure 8A*). Interestingly, when we used the same number and strength of synapses in the mature and young abGC model, the young abGCs could follow the input even at high frequencies (*Figure 8A*, right panel, dashed pink line) whereas the mature GCs could only follow frequencies below 20 Hz, mainly due to the activation of slow AHP currents. However, it is known, that young abGCs have a lower number of synapses than mature GCs as

indicated by lower spine densities (*Zhao et al., 2006*) and lower miniature excitatory post-synaptic current (mEPSC) frequencies but similar amplitudes (*Mongiat et al., 2009*). Therefore, we reduced the number of synapses in the young abGC model accordingly by a factor of two (15 synapses, same strength). Interestingly, in this reduced input configuration, abGCs had a similar synaptic input/output relation as the mature GCs (*Figure 8A*, right panel, solid pink line compare with solid blue line).

In order to test the input tuning of young abGCs, we then investigated their integration of temporally delayed inputs. In these simulations, the phase of the distal inputs (OML) was shifted with respect to the proximal synapses (MML) by a time difference Δt, and the more realistic 'reduced input' (15 synapses) configuration was used for young abGCs. Interestingly, we found that whereas mature GCs were not able to integrate considerably delayed inputs (Δt > 15 ms), young abGCs performed better (*Figure 8B*, upper graph), despite receiving input from less synapses. However, when the frequency exceeded 20 Hz young abGCs performed slightly worse than mature GCs in following the input frequency (*Figure 8B* lower right graphs). The higher the frequency, the worse was the performance correlated with Δt as the young cells' activity became saturated. In summary, at low frequencies in the theta range, young abGCs were able to integrate synaptic inputs with a broader time window than mature GCs. This is in agreement with reports of broader abGC tuning to their input activity (*Aimone et al., 2010*; *Johnston et al., 2016*; *Rangel et al., 2013*).

## Discussion of GC modeling results

### Mouse and rat mature GC model

A source of possible issues in former mature GC modeling studies lies in the fact that morphologies, biophysical mechanisms and electrophysiological data were not always used together in a consistent way concerning the animal/cell age or the species, that is mouse or rat. The widely used model from Aradi and Holmes (*Aradi and Holmes, 1999*) had originally been fitted on a mature rat morphology and rat experiments. However, since then it has been used with newborn GC morphologies (*Tejada et al., 2012*), fitted on mouse data (*Ferrante et al., 2009*) or used with mouse morphologies to reproduce mouse (*Platschek et al., 2016*) or even rat experiments (*Krueppel et al., 2011*; *Chiang et al., 2012*). Thus, there was a need for developing a consistent compartmental GC model, which would be specific for a given species and GC maturation phase. We took great care to fit and use our mature mouse GC model only with mature mouse morphologies and corresponding electrophysiological experiments. As the resulting detailed compartmental model provided accurate results by mimicking mature mouse GC behavior, we used it to develop a young mouse abGC and a mature rat GC model by implementing differences suggested by literature concerning ion channel expression and electrophysiology as compared to mature mouse GCs.

Interestingly, despite the thicker ML in the rat DG, the total dendritic length and surface of reconstructed mouse (*Schmidt-Hieber et al., 2007*) and rat (*Beining et al., 2017*) GCs were not significantly different from each other. This phenomenon can be explained by the increased branching in mouse GCs (~16 branch points in mouse vs ~10 in rat). We first challenged our model to reproduce electrophysiological data from rat by simply replacing the mouse with rat morphologies. However, the combined rat $R_{in}$ measurements from literature implied a steeper I-V relationship than in mouse (*Staley et al., 1992*; *Schmidt-Hieber et al., 2004*; *Mateos-Aparicio et al., 2014*; *Pourbadie et al., 2015*) and this difference could not be explained by morphology alone. Increasing the Kir conductance resulted in F-I curves matching experimental data. These results suggest that rat GCs display a reduced excitability due to incorporation of additional Kir channels. As an alternative, this could also be achieved by other leak channels such as K2P channels; however, the rat I-V curve from experimental data in *Figure 4B* showed pronounced inward rectification, further supporting Kir channels as an underlying mechanism. In line with this, the increased leak conductance in the rat GC model improved the fitting of simulated bAP attenuation to physiological recordings obtained from rat experiments (*Krueppel et al., 2011*), as the attenuation was too weak in the unmodified rat model (data not shown).

## Young abGC model

As the neuronal circuit of the DG is continuously remodeled by adult neurogenesis, the role of adult-born neurons in hippocampal processing gains relevance (*Rangel et al., 2013*; *Johnston et al., 2016*). As these neurons transiently exhibit unique intrinsic and synaptic properties they are subject to many studies (see reviews *Aimone et al., 2010*; *Rangel et al., 2013*; *Johnston et al., 2016*). Many special features have been associated with abGCs, such as increased excitability due to reduced Kir2 channel expression (*Mongiat et al., 2009*), increased synaptic plasticity due to higher NMDAR-2b expression (*Ge et al., 2007*), low inhibition due to developmentally delayed input from both feedforward and feedback GABAergic inhibitory loops (*Marín-Burgin et al., 2012*; *Pardi et al., 2015*; *Temprana et al., 2015*) and prolonged calcium transients due to different buffering capacities (*Stocca et al., 2008*). These special features of abGCs are most prominent during a critical time window (4–6 weeks after cell birth), when abGCs are supposed to exert their special role in learning and memory (*Ming and Song, 2011*). In our attempt to create the first compartmental model of abGCs, we focused on their special intrinsic, non-synaptic properties known to exist at the start of the critical time window, namely increased input resistance and weaker Na/K peak conductances (*Mongiat et al., 2009*). To implement these changes we used data on ion channels which are known to be upregulated during postnatal development (*Table 3*) assuming that adult-born is similar, delayed at the most, as postnatal GC development (*Espósito et al., 2005*; *Zhao et al., 2006*; *Snyder et al., 2012*). Even though a lower expression (or alternative splicing) of BK channels is only visible at P14 or earlier, we also had to reduce BK channels in our young abGC model because the fast AHP, which is mainly regulated by BK channels in GCs was reported to be reduced in young abGCs (*Yang et al., 2015*), an observation we also found in our raw traces (*Figure 7B*, left) from *Mongiat et al. (2009)*. The parameters of the young abGC model were fitted best when we reduced the beta4-subunit-associated BK current (gabk) by 100%. Thus, the abGC model predicts that the beta4 subunit is less expressed or not associated with BK channels in young abGCs.

To investigate the impact of the special intrinsic properties of young abGCs on their synaptic integration, we subjected both young and mature GC models to a broad range of synaptic input stimulation frequencies ranging from 10 to 75 Hz using T2N. In line with experimental data (*Mongiat et al., 2009*; *Pardi et al., 2015*), we found that diminished glutamatergic input onto abGCs was compensated by their enhanced excitability when GABAergic inhibition was absent. Both populations of GCs responded in a similar fashion over a wide range of stimuli, which is also in agreement with electrophysiological recordings (*Pardi et al., 2015*). Furthermore, despite their weaker excitatory input, we found that young abGCs were more efficiently activated by temporally separated (>15 ms) incoming activity from medial and lateral perforant path inputs as compared to mature GCs. In future models, both feedforward and feedback inhibitory GABAergic inputs as well as realistic proportional numbers of immature and mature GCs could be incorporated to obtain a comprehensive realistic model of the DG network.

## Model limitations

Our model was mainly fitted on raw voltage and current traces that had been the basis of the findings in *Mongiat et al. (2009)*. The advantage of such traces compared to data extracted from literature is that the same analyses (e.g. of spike width) can be performed on traces from both experiment and model, thereby increasing the accurateness of reproducing experimental data. Furthermore, the diversity of responses can directly be assessed and taken into account. However, similar to many other physiological recordings in ex vivo slices, the experiments were performed at room temperature. In our sensitivity analysis we found that temperature has a crucial impact on spiking behavior and adaptation (see above and *Figure 6A*), Therefore, our model might have limited predictability on some aspects of GC behavior at body temperature. In future studies, using raw traces from GCs at body temperature, our GC

model could be adapted to physiological temperatures or be used to investigate temperature-sensitivity of GC ion channels.

Kir channels significantly shape the resting membrane potential (*Day et al., 2005*; *Stegen et al., 2012*). A portion of the current that is described by our Kir2 model is probably also mediated by G-protein coupled Kir channels (Kir3 or GIRK) and by ATP-sensitive Kir channels (Kir6 or KATP) in real GCs. Both are expressed and functional in GCs (*Karschin et al., 1996*; *Pelletier et al., 2000*; *Tanner et al., 2011*) but are not part of our GC model as this would have required models of the G-protein and ATP molecular machinery (*Enkvetchakul et al., 2000*; *Proks and Ashcroft, 2009*). GIRKs might be involved in neuromodulation through their activation by G-protein coupled receptors (e.g. 5-HT$_{1A}$ serotonin, GABA$_B$ and D$_2$ dopamine receptors). KATP channels are involved in controlling the resting membrane potential (*Baukrowitz and Fakler, 2000*) and a part of the slow AHP (*Andrade et al., 2012*) due to their opening upon ATP depletion after long-lasting spiking phases, which might explain the vulnerability of our model to high and prolonged current injections. Furthermore, in experiments, the Na$^+$ ionic drive drops during strong and prolonged current injections as the Na$^+$/K$^+$ pump activity becomes saturated (*Nørby et al., 1983*; *Zahler et al., 1997*; *Forrest et al., 2012*). Hence, an implementation of models for the Na$^+$/K$^+$ pump, KATP channels and cellular ATP handling might further improve the GC model.

To model sodium channel isoforms Na$_v$1.2 and Na$_v$1.6, which are expressed in GCs, we used a unifying Na$_v$ model which had been developed based on AP measurements in mouse GCs (*Schmidt-Hieber and Bischofberger, 2010*). In the experimental data on which we fitted our mouse GC models (*Mongiat et al., 2009*), GCs displayed an initial high maximal rate of voltage rise during an AP which slowly decreased with increasing current injections whereas in the model the rate was low at the beginning and increased with higher current injections (*Figure 3—figure supplement 4*, mature GCs and young abGCs in mouse). Furthermore, in the model, the amplitude of the second spike was lower indicating only partial recovery from inactivation (*Figure 3D*). In addition, the model generated rather large bAP amplitudes compared to the bAP data from (*Krueppel et al., 2011*). However, the latter only has one data point for the somatic voltage amplitude, so we could not estimate the standard deviation of this measurement. Furthermore the protocol with which the bAPs were elicited was not reported there. In our simulations, we used a very short (2.5 ms) and strong current pulse to elicit the bAPs. A different, longer pulse that partly inactivates Nav channels might result in lower amplitudes. Nonetheless, developing detailed models of Na$_v$1.2 and 1.6 and their controlling mechanisms might help improving these aspects of the GC model. It is possible that young abGCs have a slightly different Nav composition with different activation kinetics. However, there exists no data on the exact Nav composition, hence we could only apply the Nav channel of the mature GC as it is. Future abGC models should take this into consideration.

An important issue is disentangling the different mechanisms of Ca$^{2+}$ buffering and thereby the interdependency of Ca$^{2+}$ channels, Ca$^{2+}$ internal stores and Ca$^{2+}$-dependent potassium channels since Ca$^{2+}$ influences AP repolarization (through BK channels) and ISIs (through SK and the slow AHP). By incorporating a phenomenological Ca$^{2+}$ buffer model reproducing realistic local Ca$^{2+}$ dynamics we avoided using unphysiological high BK or Ca$^{2+}$ channel densities (*Aradi and Holmes, 1999*; *Santhakumar et al., 2005*; *Hayashi and Nonaka, 2011*; *Jaffe et al., 2011*) or adding unknown Ca$^{2+}$- and voltage-dependent channels (*Mateos-Aparicio et al., 2014*). This represents a significant improvement over former GC models. However, we have not explicitly modeled Ca$^{2+}$ from the internal Ca$^{2+}$ stores. The implementation of such an internal store mechanism might be important for modeling synaptic plasticity (*Jedlicka and Deller, 2017*) but also for detailed modeling of fast, medium and slow AHP generating potassium channels (*Kaufmann et al., 2010*; *Piwonska et al., 2008*; *Shruti et al., 2012*; *Wang et al., 2016*). Another useful extension of the model would be an implementation of explicit spines. Spines are especially important during synaptic activation generating strong local depolarization (*Gulledge et al., 2012*; *Tønnesen et al., 2014*). Therefore, their explicit model will be needed to simulate realistically the effects of the depolarization on spine-localized ion channels and plasticity mechanisms.

In contrast to dendrites of CA1 pyramidal cells, dendrites of GCs have not yet been studied using systematic cell-attached recordings. Therefore, dendritic ion channels of GCs and their location-dependent properties have not yet been fully characterized. Thus it is possible that dendritic and somatic channels may have different expression, kinetics and voltage-dependence profiles. Indeed, several channels show location-dependent gradients in their conductance, kinetics or voltage-dependence (for reviews see *Migliore and Shepherd, 2002*; *Lai and Jan, 2006*; *Narayanan and Johnston, 2012*). Even somato-dendritic gradients in intracellular milieu (e.g. in $Ca^{2+}$ or $Cl^-$ concentration) may contribute to location-dependent channel properties in neurons. Therefore, future experiments should address potential somato-dendritic differences and variability in GC ion channel properties and their impact on GC electrophysiology. Similarly, it is currently discussed that neurons could achieve a specific electrophysiological profile with different configurations of ion channels and channel densities (see the discussion of degeneracy in the main text). Therefore, it might be possible that in a different model configuration ion channels could have different impact on GC behavior, e.g. Kv4/Kv1.4 channels contributing more strongly to $R_{in}$ and I-V relationship, or SK channels regulating spike frequency adaptation rather than Kv7 channels. Taken together, future GC models should incorporate channel degeneracy and location-dependent channel properties to study possible functional role and synergy of such channels. (*Rathour and Narayanan, 2014*; *Drion et al., 2015*; *Mishra and Narayanan, 2017*).

# Detailed description of GC morphologies and ion channels

## Reconstructed dendritic morphologies

Eight morphologies of mature mouse GCs (*Schmidt-Hieber et al., 2007*) were converted to *NEURON* models using the *TREES toolbox* (*Cuntz et al., 2010*). To each morphology a synthetic axon (length: 1350 µm, diameter 0.45 µm) was added where no axon was provided. Three of those morphologies are shown in *Figure 2A*. The biophysical model for mouse mature GCs was first fitted on the basis of these morphologies. In order to accommodate the layer dependent mechanisms of the biophysical model, it was necessary to define the following dendritic regions: the granule cell layer (GCL), and the inner, middle and outer molecular layer (IML, MML and OML, respectively). We assumed a molecular layer (ML) thickness of 188 µm (*Zhou et al., 2009*; *Dokter et al., 2015*) and subdivided it with a ratio of (0.2 : 0.4 : 0.4) as the IML is smaller compared to MML and OML in mouse due to the lack of commissural fibers. We rotated the morphologies to align the distal dendritic tips and assigned all dendrites within 75 µm of the tips to the OML, all dendrites within 75 µm of the OML border to the MML and the dendrites within 38 µm of the MML border to the IML. The remaining dendritic segments between the IML border and the soma were assigned to the GCL. For the rat GC model, reconstructed morphologies from a previous study were used (*Beining et al., 2017*). We only chose reconstructions that belonged to the mature population, were untreated (contralateral side) and had a completeness of over 90% resulting in five morphologies. Dendritic regions in these morphologies were anatomically assigned based on the slices they were reconstructed from (see *Beining et al., 2017*).

## Morphological models

To further test the robustness of the biophysical model against morphological variations, 15 synthetic mouse and rat morphologies of mature GCs were generated with a self-written morphological GC model based on the minimum spanning tree algorithm available in the *TREES toolbox* (*Cuntz et al., 2010*). The rat morphological model has already been published (*Beining et al., 2017*) and reproduces morphological data of mature rat GCs with great detail (*Figure 4A*, right side). For the mouse morphological model (*Figure 2B*), we used the GC morphologies mentioned above (*Schmidt-Hieber et al., 2007*) to fit the parameters, which resulted in several changes compared to the rat model. Since the volume of the mouse DG is considerably smaller, the thickness of the ML was reduced to 188 µm in the model to mimic

the ML thickness of the real morphologies including the ratios between the layers (38 : 75 : 75 μm). In parallel, the threshold for pruning short terminal segments was reduced to 20 μm. Furthermore, mouse GCs were much more branched (compare morphologies in *Figures 3* and *4*), reaching a similar total dendritic length despite a smaller ML. Hence, we doubled the total number of target points (50), and changed their distribution to obtain more points in the MML, where branching was extensive in real mouse morphologies (*Schmidt-Hieber et al., 2007*). Fitting a quadratic diameter taper function to the morphologies of Schmidt-Hieber (*Schmidt-Hieber et al., 2007*) resulted in an offset of 0.396 μm and a scaling factor of 0.1, which was used to taper the dendritic diameter of the synthetic trees. The soma diameter was set to $10.25 \pm 0.5$ μm. Analogously to the real GC morphologies, a synthetic axon was added.

## Ion channels
We performed an extensive literature research on the existence and subcellular distribution of different channel isoforms in mature GCs and found 16 ion channel isoforms (*Figure 2A*) together with their coarse subcellular distribution. Briefly, an 8-state sodium ($Na_v$) channel model from (*Schmidt-Hieber and Bischofberger, 2010*) was adapted, a leaky inward-rectifier channel (Kir) model was developed based on experimental data (*Lopatin et al., 1995*; *Dhamoon et al., 2004*; *Yan and Ishihara, 2005*; *Panama and Lopatin, 2006*; *Ishihara and Yan, 2007*; *Liu et al., 2012*), A-type potassium channels were represented by models of $K_v1.1$ (*Christie et al., 1989*), $K_v1.4$ (*Wissmann et al., 2003*) and $K_v4.2$ (*Barghaan et al., 2008*) channel isoforms, the delayed-rectifier channel $K_v3.4$ was fitted on data from (*Rudy et al., 1991*; *Schröter et al., 1991*; *Rettig et al., 1992*; *Miera et al., 1992*; *Riazanski et al., 2001*; *Desai et al., 2008*), and the M-type potassium channel $K_v7.2/3$ was taken from (*Mateos-Aparicio et al., 2014*). Furthermore, we included an HCN channel (*Stegen et al., 2012*) adapted to a lower threshold for activation, a $Ca_v2.2$ (N-type) calcium ($Ca^{2+}$) channel (*Fox et al., 1987*) adapted to a more realistic, slower inactivation time constant (100 ms instead of 1–10 ms), a T-type $Ca_v3.2$ $Ca^{2+}$ channel model from (*Burgess et al., 2002*), two L-type $Ca_v1.2$ and $Ca_v1.3$ $Ca^{2+}$ channel models (*Evans et al., 2013*) transferred from *GENESIS* (*Bower and Beeman, 1998*), the $Ca^{2+}$-dependent big and small conductance potassium channels BK and SK from (*Jaffe et al., 2011*) and (*Solinas et al., 2007*) with modifications (see below for more details), and an improved $Ca^{2+}$ buffering model (see below for more details). We then looked for the kinetics of these channels and chose ion channel models that followed these kinetics (or could be modified appropriately) or, if not available, fitted own channel models to these data. This was done by performing least squares fitting on the activation and inactivation curves and time constants provided by the literature cited in *Table 1*. Generally, all channels were incorporated according to their expression strength found by protein immune staining, followed by fine-tuning of the channel densities to fit the active properties of mature GCs. The original literature, the incorporated channel densities and the origin of the channel models are summarized in *Table 1*. A general review on axonal targeting of voltage-dependent potassium channels can be found in *Gu and Barry (2011)*. The reversal potentials for potassium and sodium were calculated from the solutions which were used in the experiments (*Mongiat et al., 2009*) ($E_K$ = −93 mV, $E_{Na}$ = 87.76 mV) and were kept constant throughout the simulations.

## Passive membrane properties
The axial resistance was chosen to be 200 Ωcm in the dendrites and the soma and 100 Ωcm in the axon. The specific membrane capacitance $C_m$ was 0.9 pF/cm². Spines were implicitly modeled by scaling the passive conductance and $C_m$ in the IML by a factor of 1.45, and in the MML and OML by a factor of 1.9 (mature mouse and rat GC model). We combined the function of two-pore channel potassium (K2P) channels such as TWIK and TREK into a single passive channel as they behave in a voltage-independent manner (i.e. linearly) over a wide voltage range and are expressed in GCs (*Lesage et al., 1997*; *Hervieu et al., 2001*; *Talley et al., 2001*; *Gabriel et al., 2002*; *Aller and Wisden, 2008*; *Yarishkin et al., 2014*).

However, it should be noted that modeling leak currents exactly (i.e. non-linearly) may affect the output of neurons (*Huang et al., 2015*).

## Passive leak channel and Kir channel

For the passive GC model, a passive leak channel model was introduced representing the cell's channels with linear current-voltage relationships, such as 2-pore potassium (K2P) channels that are known to be strongly expressed in GCs (*Hervieu et al., 2001*; *Gabriel et al., 2002*; *Yarishkin et al., 2014*). Moreover, we incorporated a model of the inward-rectifying Kir2 channel, which also contributes to the cell's properties at resting potential. Kir channels in GCs are expressed as several variants (Kir2.1–4) (*Karschin et al., 1996*; *Prüss et al., 2005*), which can assemble to heteromeric channels with mixed properties (*Dhamoon et al., 2004*). Hence, because the stoichiometry of Kir channel subunits is not known in GCs yet, we aimed to incorporate a unifying Kir2.x model. We used a model for a Kir 2.1 channel that described low- and high-affinity spermine block as well as $Mg^{2+}$ block modes (*Yan and Ishihara, 2005*). Since later studies showed that both modes exist in parallel with the high-affinity block being a substate of the channel that still shows low conductance (*Liu et al., 2012*), we first changed the high-affinity block state to be a substate. To account for other Kir2 isoforms and putative Kir2.1–3 heteromers, which have different sensitivities to spermine block and are thus less rectifying (especially Kir2.3) (*Panama and Lopatin, 2006*), we set the high-affinity fractional subconductance to 0.25 (which corresponds to the fractional conductance being susceptible to the low-affinity spermine block) compared to 0.09–0.15 in the original studies with homomeric Kir2.1 (*Yan and Ishihara, 2005*; *Ishihara and Yan, 2007*). Then, we added the low-affinity state as another substate to the model (six states in total) and fitted an exponential back and forward rate as the low-affinity mode had originally been described without kinetics, being assumed to be instantaneous (*Yan and Ishihara, 2005*). Furthermore, the unblock rates of the high-affinity states were slowed down to match the values from literature (*Lopatin et al., 1995*; *Panama and Lopatin, 2006*). Another issue with the original Kir2.1 model was that it was based on measurements in equal extra- and intracellular potassium (i.e. $E_K = 0$ mV) and different $Mg^{2+}$ concentrations, two factors which strongly influence Kir rectification. As the dependence between $E_K$ and $V_{1/2}$ of the Kir channel block has a slope of exactly 1 (*Panama and Lopatin, 2006*), we shifted the activation and inactivation of the model by −93 mV, which was the $E_K$ in our biophysical model. Furthermore, as the $Mg^{2+}$ block of the Kir channel was weakened in the experiments (*Mongiat et al., 2009*) due to a drastic difference in the $Mg^{2+}$ driving force ($E_{Mg} = -14.38$ mV and $E_K = -93$ mV vs. $E_{Mg} = -88$ mV and $E_K = 0$ mV in the original study), we adapted the $Mg^{2+}$ block in the Kir model by shifting the $Mg^{2+}$ inactivation by only $0.5 * E_K$ and reducing the influence of intracellular $Mg^{2+}$ concentration by a factor of 8. The resulting model could well reproduce the steady-state I–V relationships (*Figure 3A*, *Figure 4B* and *Figure 7A*) as well as the slow unblock at hyperpolarized potentials (*Figure 3—figure supplement 3*).

## Sodium channels

We implemented the 8-state sodium ($Na_v$) channel model from Schmidt-Hieber as it had been directly fitted on mature GCs (*Schmidt-Hieber and Bischofberger, 2010*) However, we found that distributing the densities according to the spatial functions used in the model was not compatible with our morphologies since the spiking behavior depended strongly on the somatic and axonal geometry, which varied significantly between cells. Thus, we chose region-dependent densities with the highest density in the axon initial segment (*Schmidt-Hieber and Bischofberger, 2010*). The dendritic sodium channel was removed because GC dendrites were reported to exhibit predominatly passive properties (*Schmidt-Hieber et al., 2007*; *Krueppel et al., 2011*). In order to compensate for the resulting smaller excitatory drive, somatic and axonal channel densities were increased compared to the original. Also, the $Na_v$ channel activation curve was shifted by +10 mV, as the −35.3 mV (axon) and −29.4 mV (soma) half-activation midpoints measured in *Schmidt-Hieber and Bischofberger (2010)* were

significantly lower than in many other studies that measured Nav 1.2 and 1.6 kinetics (*Smith et al., 1998*; *Oliva et al., 2014*; *James et al., 2015*). This shift also reproduced the spike threshold from the electrophysiology data much better (see *Figure 3E*). Furthermore, the inactivation kinetics had originally only been fitted between −50 and 20 mV thereby omitting the reproduction of deinactivation kinetics between −120 and −50 mV, which were slower than the reported values in literature (*Rush et al., 2005*; *Mercer et al., 2007*). Hence, we adapted the inactivation rates to fit the recovery kinetics from inactivation, too.

## $K_v$ channels

*Table 1* summarizes the literature that the different $K_v$ models were taken from or fitted onto. Importantly, $K_v1$ channels at the axon initial segment control action potential (AP) waveforms and synaptic efficacy (*Kole et al., 2007*) and form heteromers comprising α- and auxiliary subunits. In the DG $K_v1.1/1.4$, heteromers are formed together with $K_v\beta1/2$ (*Rhodes et al., 1997*; *Monaghan et al., 2001*) and KCNE1/2 (*Tinel et al., 2000*; *Kanda et al., 2011*) subunits. However, these interactions have only been modeled in detail in the $K_v4$ channel (*Amarillo et al., 2008*; *Barghaan et al., 2008*); we therefore implemented a standard $K_v1.1$ and $K_v1.4$ model from (*Christie et al., 1989*) and (*Wissmann et al., 2003*), which might ignore functional impacts such as the calcium-dependent modification of inactivation kinetics through $K_v\beta1$ (*Jow et al., 2004*).

## HCN channels

The existence of HCN channels in GCs remains controversial (*Stegen et al., 2012*) which might partly be explained by the use of different species and by the fact, that HCN channels are blocked differentially by the $Mg^{2+}$ present in the pipette in various concentrations (*Vemana et al., 2008*). We implemented an HCN channel (*Stegen et al., 2012*) but left-shifted the voltage-dependence by 10 mV since $V_{1/2}$ was more hyperpolarized than −90 mV for all HCN channel isoforms found in the literature (*Altomare et al., 2001*; *Bräuer et al., 2001*; *Chen et al., 2001*; *Surges et al., 2006*; *Postea and Biel, 2011*).

## Calcium channels

The $Ca_v1.2$ and $Ca_v1.3$ (L-type) $Ca^{2+}$ channel models were taken from (*Evans et al., 2013*) and transferred from *GENESIS* (*Bower and Beeman, 1998*) to NEURON. As $Ca_v1.3$ compared to $Ca_v1.2$ is known to show only partial voltage-dependent inactivation (VDI; see *Bell et al., 2001*; *Koschak et al., 2001*) we restricted VDI of $Ca_v1.3$% to 85% of the total conductance (see *Appendix 2—figure 2*). For the $Ca_v2.2$ (N-type) Ca-channel we used the activation and inactivation kinetics from *Fox et al. (1987)*, but set the inactivation time constant to 100 ms (compared to 1–10 ms between 0 and 50 mV in the original model). The resulting inactivation kinetics of $Ca_v2.2$ at these voltages were then more similar to experimental results (*Fox et al., 1987*; *Huang et al., 2010*) and were in the range of inactivation time constants of other models (*Wolf et al., 2005*; *Hemond et al., 2008*; *Evans et al., 2013*; *Papoutsi et al., 2013*). However, it should be noted that the deinactivation kinetics (below 0 mV) are much slower in real channels being in the range of seconds (*Zhu et al., 2015*), which might be important when $Ca_v2.2$ is inactivated by a long depolarization and then reactivated shortly thereafter.

The expression and distribution of T-type Ca-channels ($Ca_v3.1–3$) was controversial in the literature: Whereas Martinello et al. (EM immune stainings) reported $Ca_v3.2$ to be mainly expressed in GC dendrites (*Martinello et al., 2015*), McKay et al. (fluorescent immune staining) found $Ca_v3.2$ to be expressed exclusively in the soma (*McKay et al., 2006*). Similarly, McKay et al. found $Ca_v3.3$ strongly expressed in GC somata (*McKay et al., 2006*), whereas Talley et al. reported very weak $Ca_v3.3$ mRNA expression in the DG and McRory et al. reported a complete lack of $Ca_v3.3$ in the hippocampus (*McRory et al., 2001*). Since one explanation might be that $Ca_v3.3$ expression is largely reduced from juvenile to the adult age (*McRory et al., 2001*; *Yunker et al., 2003*), we decided to only implement a $Ca_v3.2$ model

from *Burgess et al. (2002)* to model T-type Ca$^{2+}$ currents. We did not use a separate Ca$_v$3.1 channel as it has kinetics similar to Ca$_v$3.2 (*Cain and Snutch, 2010*). Moreover, since no consensus could be found about the subcellular distribution, we incorporated the model into all compartments with increased densities in the soma and dendrite (*McKay et al., 2006*; *Martinello et al., 2015*).

## Calcium buffer

Since calcium (Ca$^{2+}$) activates small-conductance and big-conductance potassium (SK and BK) channels and plays a crucial role for synaptic plasticity, we developed an improved Ca$^{2+}$ buffering model. Submembrane shell models of Ca$^{2+}$ buffering and dynamics in neurons were used in previous compartmental models of granule cells (*Yuen and Durand, 1991*; *Aradi and Holmes, 1999*; *Santhakumar et al., 2005*; *Hayashi and Nonaka, 2011*; *Mateos-Aparicio et al., 2014*). They were based on the assumption that Ca$^{2+}$ is mainly active within a thin shell beneath the cell membrane but rapidly buffered outside of that shell. However, as recently reported (*Anwar et al., 2014*), many of these shell models were found to contain an error that was introduced in early GC models and led to incorrect Ca$^{2+}$ levels in thin dendrites. Hence, we implemented a Ca$^{2+}$ shell model (using a shell depth of 0.05 μm) corrected to varying diameters in the morphology (*Anwar et al., 2014*).

Furthermore, earlier compartmental models of GCs used shell models with fast exponential Ca$^{2+}$ decay times of 9 ms (*Yuen and Durand, 1991*; *Aradi and Holmes, 1999*; *Santhakumar et al., 2005*; *Mateos-Aparicio et al., 2014*). This was an estimate originally implemented into the compartmental GC model of *Yuen and Durand, 1991* to reproduce spike adaptation and represents averaged and simplified Ca$^{2+}$ dynamics which, in reality, extends over several time scales in nerve cells (*Blaustein, 1988*; *Yuen and Durand, 1991*). Slower dynamics of 100 ms have only been used in motorneurons so far (*Traub and Llinás, 1977*), but a carefully calibrated Ca$^{2+}$ imaging study in rat and mouse mature GCs revealed decay time constants of 230 ± 30 ms and 280 ± 30 ms, respectively, as well as considerably lower Ca$^{2+}$ peak levels (*Stocca et al., 2008*) as compared to former models (*Aradi and Holmes, 1999*; *Santhakumar et al., 2005*; *Hayashi and Nonaka, 2011*; *Mateos-Aparicio et al., 2014*). Furthermore, other studies have reported that the membrane of GCs comprises micro- and nanodomains of Ca$^{2+}$ channels clustered with SK or BK channels at distances between 13 and 150 nm and showed that the Ca$^{2+}$ rise and decay can be very large and nearly instantaneous in these domains (*Marrion and Tavalin, 1998*; *Müller et al., 2007*; *Fakler and Adelman, 2008*; *Kaufmann et al., 2010*). In our model, we aimed to reproduce overall Ca$^{2+}$ increase and decay on the one side, but also the local Ca$^{2+}$ increase in micro- and nanodomains. Hence, we introduced a phenomenological model of Ca$^{2+}$ buffering by dividing the Ca$^{2+}$ influx with a constant, which was analogous to the so called Ca$^{2+}$ binding ratio representing the ratio of buffer-bound Ca$^{2+}$ ions versus free ions. Following the literature we assumed a lower Ca$^{2+}$ binding ratio of 10 in the axon compared to 50 in the dendrite (*Jackson and Redman, 2003*; *Stocca et al., 2008*). A somatic Ca$^{2+}$ binding ratio has not been reported, yet, but might be high due to a high amount of fixed and mobile buffers such as mitochondria (*Duchen, 1999*). Hence, we set the Ca$^{2+}$ binding ratio to 200 in the soma. To model the clustering of Ca$^{2+}$ channels with BK and SK channels, we additionally supplied BK and SK channels with unmodified (i.e. no Ca$^{2+}$ binding ratio) instantaneous local [Ca$^{2+}$]$_i$ from the respective clustered Ca$^{2+}$ channel (N-type Ca$^{2+}$ channels for BK and L-type Ca$^{2+}$ channels for SK channels; see *Marrion and Tavalin, 1998*). In this way, the model was taking into account that these channels are in close proximity to Ca$^{2+}$ channels and therefore their Ca$^{2+}$-activation is not affected by the intracellular Ca$^{2+}$ buffers (*Müller et al., 2007*; *Fakler and Adelman, 2008*). We set the Ca$^{2+}$ decay time constant to 43 ms in the axon (*Jackson and Redman, 2003*) and to 240 ms in all other compartments (*Stocca et al., 2008*). To fine-tune the Ca$^{2+}$ channel density distributions of the four used Ca$^{2+}$ channels (Ca$_v$1.2, Ca$_v$1.3, Ca$_v$2.2 and Ca$_v$3.2) we additionally considered the contribution of each isoform to the Ca$^{2+}$ current at 100 mV (*Eliot and Johnston, 1994*) and peak calcium levels following an AP (*Jackson and Redman, 2003*; *Stocca et al., 2008*).

## Calcium-dependent potassium channels

$Ca^{2+}$-dependent potassium channels such as BK and SK form micro- and nanodomains with specific $Ca^{2+}$ channels in GCs (reviewed in [*Fakler and Adelman, 2008*]): N-type $Ca^{2+}$ channels form nanodomains with BK (*Marrion and Tavalin, 1998*; *Loane et al., 2007*) and L-type $Ca^{2+}$ channels form microdomains with SK channels (*Marrion and Tavalin, 1998*). In these clusters, the local $Ca^{2+}$ concentration can be very fast and high during $Ca^{2+}$ channel opening. Thus, as aforementioned, an additional local $[Ca^{2+}]$ was directly calculated for BK and SK channels from the respective locally clustered $Ca^{2+}$ channel without applying the $Ca^{2+}$ binding ratio. For simplification and fast computation this local $[Ca^{2+}]$ was assumed to be instantaneous, that is, without a rise or decay time. To account for SK having a higher distance of ~150 nm to its L-type $Ca^{2+}$ channels in the microdomains, compared to 13–50 nm for BK channels we divided the local SK $[Ca^{2+}]$ by 3.

The BK channel model from (*Jaffe et al., 2011*) simulated BK α-subunits with or without β4-subunits which make the channel kinetics much slower and resistant against the BK blocker iberiotoxin. Pharmacological studies with iberiotoxin suggest that wildtype GCs do not contain pure α-BK channels (*Shruti et al., 2012*), but prolonged application of high toxin concentrations were shown to also block wild-type GC BK channels (*Müller et al., 2007*), suggesting, that BK channels might be expressed with different stochiometries of α- and β4-subunits resulting in partial iberiotoxin-resistances and intermediate kinetics (*Wang et al., 2014*). As there exist no studies on such intermediate kinetics, we implemented both the α- and α β4-subunit model of BK.

We adapted the SK2 channel model from *Solinas et al. (2007)* which is based on the kinetic model by *Hirschberg et al. (1998)* to model all SK channel isoforms expressed in GCs (SK1-3) as they have similar kinetics and calcium-dependencies. We found that the activation kinetics of the model following $Ca^{2+}$ transients were too fast and the inactivation kinetics to slow, compared to experimental measurements (*Hirschberg et al., 1998*; *Xia et al., 1998*), hence we refitted the constant and the $Ca^{2+}$-dependent rates to the experimental data.

## Fitting of ion channel densities

The initial relative distribution of all channel densities was taken from the qualitatively assessed channel expression pattern (see *Table 1*) and subsequently fine-tuned by hand. Fine-tuning of the models 'passive and active channel densities was done with Matlab by carefully adapting densities to mimic raw voltage and current traces (*Mongiat et al., 2009*) (control vs. $BaCl_2$ application) and further experimental data on GC physiology (e.g. $Ca^{2+}$ dynamics and physiology at dendrites) from literature (*Riazanski et al., 2001*; *Müller et al., 2007Müller et al., 2007*; *Schmidt-Hieber et al., 2008Schmidt-Hieber et al., 2007*; *Stocca et al., 2008Stocca et al., 2008*; *Schmidt-Hieber and Bischofberger, 2010*; *Krueppel et al., 2011Krueppel et al., 2011*). Thereby, each experiment was recreated according to the information provided by the publications using T2N and standard NEURON mechanisms (IClamp, SEClamp, Exp2Syn etc., see code for more details).

## Aradi and Holmes mature GC model

In order to assess the performance of our mature GC model, we compared it to the widely used GC model developed by *Aradi and Holmes (1999)*. We implemented the original (https://senselab.med.yale.edu/ModelDB/showModel.cshtml?model=116740) Aradi and Holmes biophysical model into our T2N framework and ran the same simulations with the same morphologies used in our model (*Figure 3—figure supplement 1*, *Figure 4—figure supplement 1*, *Figure 5—figure supplement 1*).

## Young adult-born GC (abGC) model

We turned our biophysical model of mature GCs into a model of young (28 dpi) abGCs by reducing several channel densities that have been reported to be less expressed in developing GCs (*Table 3*). By reducing the $Na_v$ channel density, a smaller $Na^+$ drive was achieved as observed in the phase plots of young abGCs (*Figure 7F*). Reducing $K_v2.1$ and $K_v4.2$ in the model decreased the medium AHP that otherwise would induce too long and hyperpolarized ISIs. Consequently, $Ca_v1.3$ also had to be reduced to avoid unrealistically strong dendritic depolarization caused by the lacking hyperpolarization mediated by $K_v4.2$. Furthermore, we reduced BK channel density to eliminate the prominent fast AHP not found in the experimental data from abGCs (*Mongiat et al., 2009*; *Yang et al., 2015*).

## Synaptic integration in abGCs

For these simulations, we randomly distributed 30 synapses equally distributed over dendrites in the MML and OML (schemes in *Figure 8*). The synapses had an exponential rise and decay with dynamics mimicking that of real EC-GC synapses (exponential rise time constant: 0.2 ms, exponential decay time constant: 2.5 ms, reversal potential at 0 mV, synaptic weight 0.65 nS each). All synapses were either activated synchronously (*Figure 8A*) with a frequency ranging from 10 to 100 Hz or with a delay between MML and OML synapses (*Figure 8B*) ranging from −25 to +25 ms. The relation between input and output was analyzed by calculating the ratio of the output versus the input frequency. Hence a ratio of 1 means the cell having the same spiking frequency as the input and 0 meaning the cell does not spike at all.

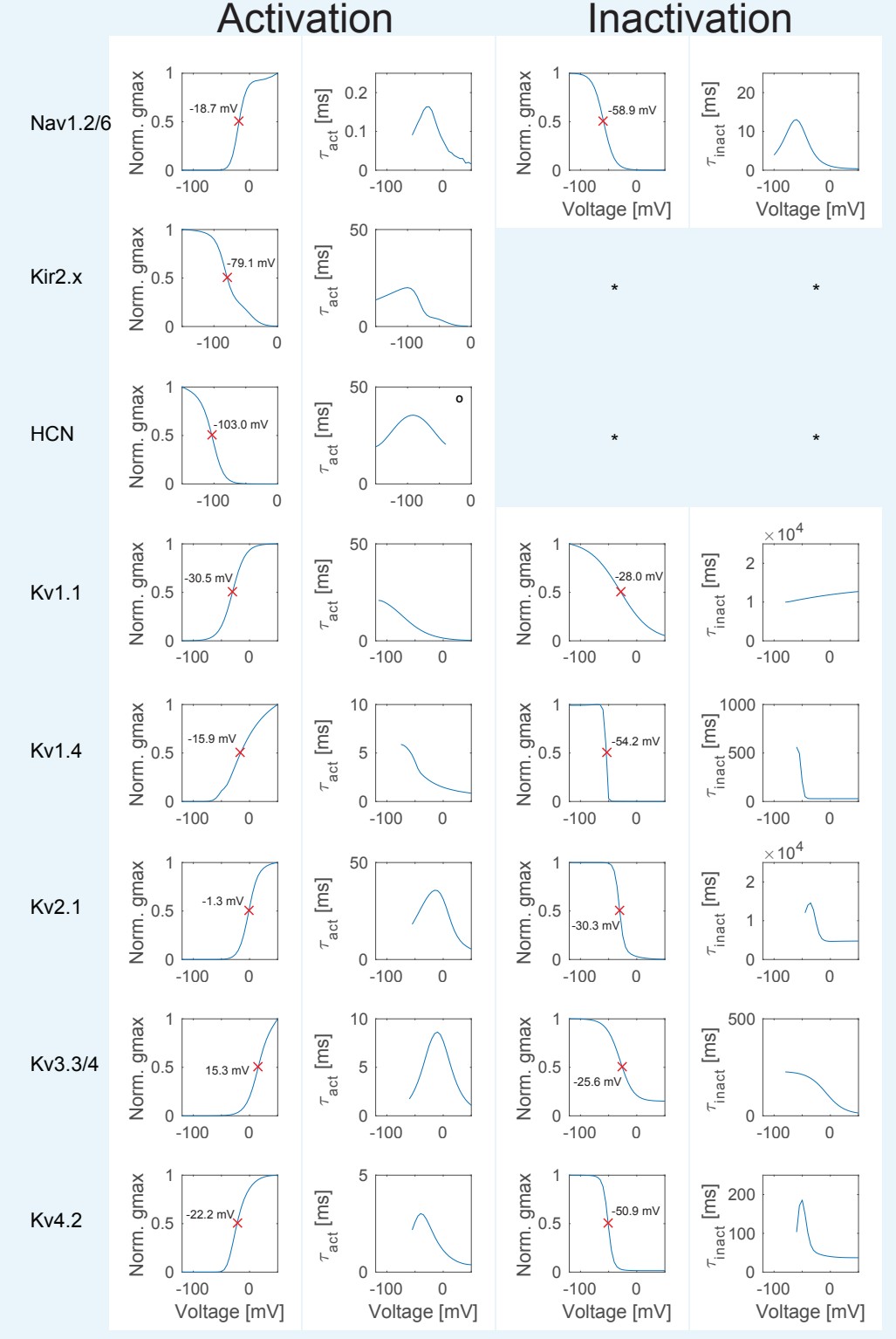

**Appendix 2—figure 1.** Overview of the ion channel activation and inactivation kinetics in the GC model. The illustrated voltage-dependent kinetics were automatically calculated and plotted with a function of the T2N package, which applied voltage step protocols to a single compartment comprising only the ion channel of interest. First column: Activation curves of all ion channels used in the GC model. The red crosses denote the half-activation voltage, which is additionally inserted as text in each case. Second column: Curve of the activation time constant at different voltages obtained with a monoexponential fit to the rise in conductance.

Degree symbol (°) denotes that only the fast inactivation component was fitted. Third column: Inactivation curves of all ion channels used in the GC model. The red crosses denote the half-inactivation voltage, which is additionally inserted as text in each case. Fourth column: Curve of the inactivation time constant at different voltages obtained with a monoexponential fit to the decay in conductance. Asterisk (*) denotes cases where no inactivation occurred, e.g. for the hyperpolarization-activated ion channels Kir2.x and HCN.

DOI: https://doi.org/10.7554/eLife.26517.025

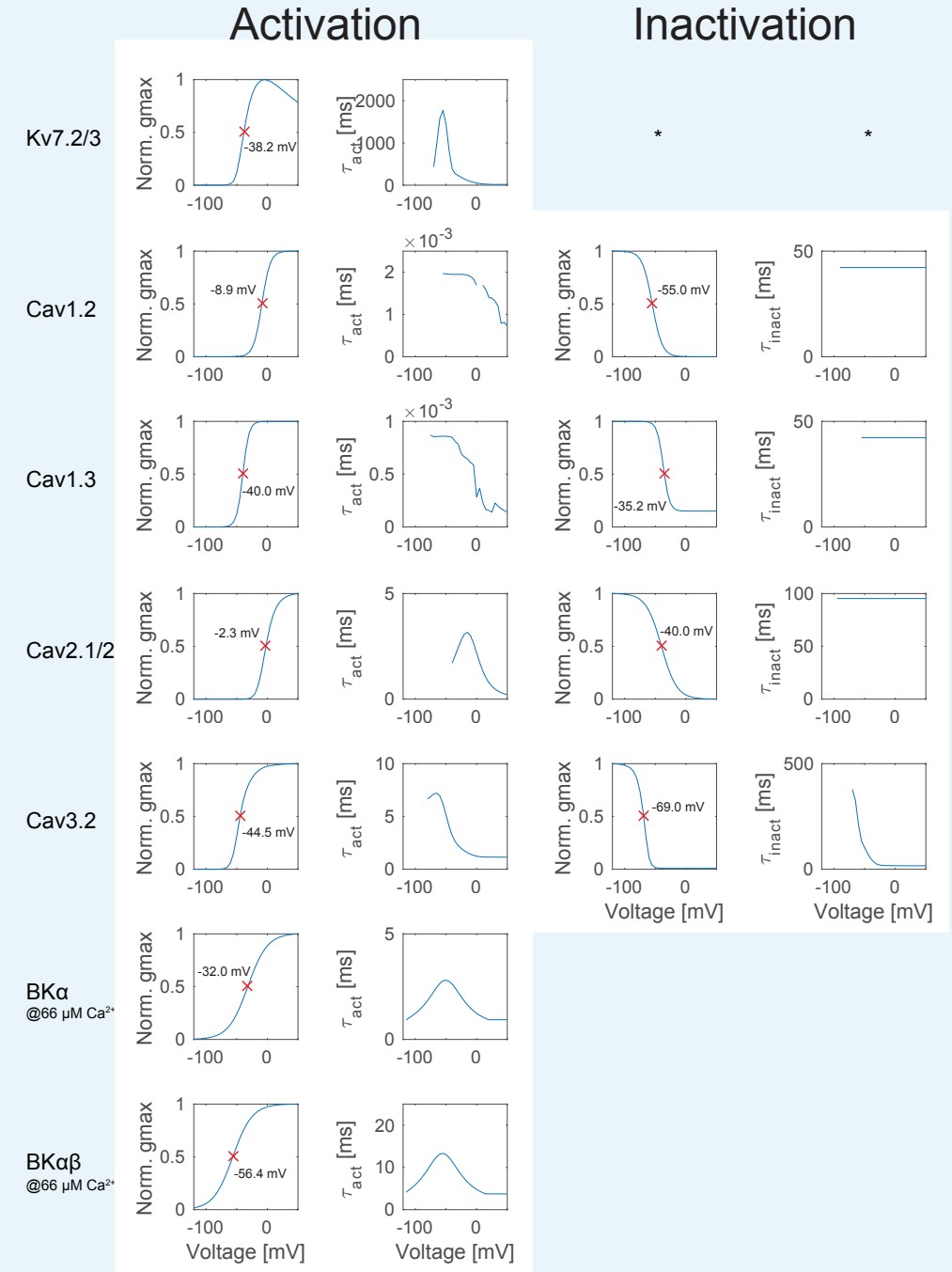

**Appendix 2—figure 2.** Overview of the ion channel activation and inactivation kinetics in the GC model (continued from *Appendix 2—figure 1*).

DOI: https://doi.org/10.7554/eLife.26517.026

