## [Decision Letter]

Thank you for submitting your article "Robust electrophysiological modeling demonstrated for mature and adult-born dentate granule cells of mouse and rat" for consideration by *eLife*. Your article has been reviewed by three peer reviewers, one of whom, Frances K Skinner (Reviewer #1), is a member of our Board of Reviewing Editors, and the evaluation has been overseen by Eve Marder as the Senior Editor. The following individuals involved in review of your submission have agreed to reveal their identity: Rishikesh Narayanan (Reviewer #2); Marianne J Bezaire (Reviewer #3).

The reviewers have discussed the reviews with one another and the Reviewing Editor has drafted this decision to help you prepare a revised submission.

Summary:

The authors present T2N, a tool that provides an interface to control NEURON with Matlab and TREES toolbox in generating compartmental models. The authors introduce their useful tool and then proceed to develop a well-constrained model using the tool. With the model cells, the authors present a number of findings and make predictions. This work sets a new benchmark for detailed cell modeling, especially for dentate gyrus and granule cell modeling, and provides a contrast between its performance and the performance of previous granule cell models.

While all the reviewers felt that this represented a potentially helpful tool, they also all felt that the work was not presented as a 'Tools and Resources" paper, and as such, would make it difficult for potential users to appreciate, assess and use themselves.

Essential revisions:

In summary, there are three essential revisions that the authors need to consider. An overview of each revision is provided first, and detailed comments from the reviewers to consider for each point are provided after the overview.

1) A rewrite of the paper is needed, mostly Results and Discussion (and Materials and methods in coordination) so that the paper is actually a Tools and Resource paper. That is, the paper should mainly be about the tool and its usage, and not mainly about the subsequent model results and interpretations. The authors need be clear about what the tool brings forth relative to what one would need to do otherwise etc. A title change should be considered in light of this.

2) Technical aspects for use of the tool for potential users needs to be included. That is, some sort of step by step, tool usage setup, mini tutorials and examples in some way.

3) In demonstrating the usage of the tool with GC models (e.g., morphological changes, dendritic remodeling in pathology/development of rat/mouse, adult-born neurons), the authors need to be clear about where and how the tool is used in their developed models. That is, the authors need to describe what it is the tool is allowing them to do (in their demonstrated use of it) rather than solely presenting their interpretation of their model results.

Please separately present tool usage and model interpretation to avoid confusion, and ensure that the emphasis is on demonstrating the tool usage.

Note that while several of the detailed comments below may refer to model interpretation, the authors need to ensure that their revised paper emphasizes and is mainly about the demonstrated tool usage and not about the model results and their interpretation per se. This way, it can be clear that the present paper is a 'Tools and Resource' one.

Detailed comments from reviewers to take into consideration when doing Essential Revision 1:

a) While the GC modeling work is interesting, I found myself reading it and wondering where the T2N tool fit in. Presumably, the modeling work could've been done without T2N but it would've been alot more complicated, harder and longer? Is it easier to do sensitivity analyses, plot and visualize etc. etc.? Is it easier to test relationships between ion channels and AP features as they do? If so, how is this the case via T2N tool? Is the T2N tool intended for cell types besides GCs? Presumably it is (otherwise it would be extremely limiting). In the outlook, it would seem that it might only be for considering GCs?. Are there any other tools that do what T2N does? What would a user do if T2N did not exist?

b) In essence, the authors talk about T2N for one page and then say that "In the following, we show at the example of the dentate GC how to build a robust compartmental model using T2N." But they actually don't explain it via T2N that I could find, and they don't simply build a model – they use it to examine epilepsy, young cells etc.

c) The Discussion starts by talking about the model results, and the tool second. This should be reversed and/or the model results should be presented in light of the tool somehow I would think if this is a 'Tools and Resources' paper.

d) There is a lot of repetition in Results and Materials and methods regarding the models, but at the same time barely any specifics regarding the tool or how to use the tool itself. This possibly stems from the authors not presenting the work as 'tools and resources'. I appreciate that there is a manual, but there should at least be some overall explanations and mini examples that the user could try/test for the tool itself?

e) The Discussion is over lengthy and several parts of it are significantly redundant with the Results section. This could be significantly reduced.

f) Focus on the tool: The manuscript is within the "Tools and Resources" section of *eLife*. The focus of the authors is the technique that they are developing, with DG employed only as an example for the demonstration of the utility of this tool. However, a significant portion of the Results and the Discussion sections, including the limitations presented there, are all very specific to the DG, and not with reference to the methodology developed by the authors. The authors should place emphasis of their manuscript on the tool that they are developing to fit into the "tools and resources" section of *eLife*. Specifically they should focus on the steps involved in the use of the tool, discuss about limitations of the tool (rather than of the DG model) from the perspective of generalizability to modeling any type of neuron (especially focusing on incorporating measurements that are not available in DG neurons; see below), future directions for how the tool could be further developed (say for well-tuned network models or for astrocyte-neuron interactions, etc). It is important to emphasize the novelties of the DG model, but the focus shouldn't stray away from the main focus of the article.

Detailed comments from reviewers to take into consideration when doing Essential Revision 2:

The problems discussed by the authors, that many models behave poorly outside of the scope for which they were created, and that model cells often contain a mix of constraints from very different experimental conditions and animals, combined in sometimes inconsistent ways, are quite real in the world of detailed modeling. The field can and should have higher expectations at this point, and this tool provides a way for us to get there. From the inclusion of variable morphologies to the impressive & precise incorporation of ion channels within the model, this tool shows promise for aiding modelers to tackle these problems in our models. This approach will allow us to more fully harness the resources within online databases such as NeuroMorpho and IonChannelGenealogy. The use of sterotyped scripts will help reduce the burden on the modeler, as will the automatic parallelization.

The T2N and TREES tools appear to have an elegant way to organize the code and have MATLAB and NEURON interact. This reviewer sees a lot of potential with these tools. The author was surprised to find that this manuscript did not seem to be written as a Tools & Resources paper. The info as to how to access the code and software is not provided until the Materials and methods. Tutorials are not provided, and there are not many technical, quantitative details about the implementation of the model, the parallelization, or the tuning. The high level of detail and careful attention to constraints that went into developing this model is not fully represented in the Abstract and Introduction.

a) Especially given the "Tools and Resources" designation of this manuscript:

– More technical details should be provided in the publication – how do you recreate the experimental conditions for your tests? In Results fourth paragraph – what fitting tool or strategy did you use?

– Each of the results headings can be thought of as a use case, for which a tutorial would be appropriate (in an appendix or as part of included documentation of T2N)

– Say whether you can use the T2N in a more exploratory manner, for cell types where channel expression data is not known but we have single cell current sweep data?

– Share how flexible is the code, for using the models produced from TREES and T2N in a larger network simulation?b) The code should be thoroughly tested on more machines (and operating systems) before being highlighted in a publication. I ran into several.

c) Would like more visibility into the ion channel models, like characterization of the ion channel behavior (activation curves, etc), maybe a table of the ion channel kinetics (time constants, voltage of half act/inact, etc)d) Feels like technical details of the approach are missing. The authors say it's "not a genetic algorithm", and it's "similar to a multiple objective approach", but don't take us through exactly what they've done to tune the parameters. Should also clarify that they found a single solution that works across many morphologies, not a set of solutions with different parameter combinations for each morphology.e) Add code to ModelDB, include link in paper with accession #.

Detailed comments from reviewers to take into consideration when doing Essential Revision 3:

a) In some parts, the authors seem to be making the point about "the importance of using detailed data-driven ion channel composition, diverse dendritic trees as well as multiple different electrophysiology experiments for tuning and generating compartmental models which would be stable across many morphologies and conditions", and in other parts, as a different approach to more standardized methods with automated parameter fitting as in BBP etc.. Is this meant to be the point of the tool? If so, it does not seem appropriate for a tools and resources paper. Models have different goals and are built to address different questions, and this is true for compartmental ones too I think.

b) Essentially, the authors focus on the model results after using the tool, emphasizing issues regarding GCs that are possible to determine with their tool. This seems somewhat circular to me. That is, they say that "our work might suggest that morphologic robustness arises naturally in models in which parameters have been tuned using multiple different experiments and morphologies. This conclusion is also supported by a comparison of our biophysical model to an earlier widely used GC model which failed to reproduce electrophysiological data…" (see my next comment), but it may be that if some other biological aspect besides morphology or in addition to it was focused on, robustness would've also been observed?

c) The authors refer to a classical GC model (Aradi and Holmes). What is meant/intended by classical? That it is the first, considered the best at present for those modeling GCs? Besides needing to explain what they mean by classical, it seems a bit unfair to show it doesn't 'match', as presumably building and using this model when Aradi/Holmes did provided some insight and understanding of GC functioning? (I didn't go back to look at that paper's details).

In other words, the authors should present this classical model in a more holistic sense. Presumably they are showing that "most published modelsbehave poorly when used outside of the scope for which they were created." as they state in their Introduction. This may not be a huge problem with the classical model (and others) so long as the model (with its limitations and caveats) was clearly presented at the time, and that some insight/understanding/hypothesis-generation etc. was achieved at the time with the model.

d) Morphology of adult-born neurons: To match physiological properties of adult-born neurons the authors have changed channel properties. However, it is well-established that maturing neurons have significantly shorter dendritic arborization and lesser spine density as well (Zhao et al., 2006). As mentioned by the authors, the lesser dendritic extent and the reduced surface area caused by the lack of spines would increase the excitability of the cell as well. The authors have accounted for spine density differences when they assess synaptic integration in the mature and immature GCs. But, why did the authors not consider matching morphological profiles of adult-born neurons at various ages to understand the physiology of adult-born neurons? Given the several differences between adult-born and mature neurons in the DG, that might have been more appropriate within the framework that the authors are proposing rather than assuming that all excitability changes are mediated by changes in ion channel densities. I believe that performing these additional set of experiments with immature morphologies (which is certainly within the capabilities of the framework proposed here) would further emphasize the utility of the tool that the authors are reporting here, especially with reference to the scenario where different statistics of morphology are observed with certain physiological/pathological conditions. In this case, the authors might also want to discuss if their framework would be capable of maintaining structural integrity of real or synthetic morphologies as they mature (Narayanan and Chattarji, 2010; Dhupia et al., 2015; Bozelos et al., 2016) so as to enable causal links between morphological characteristics and physiological measurements.

e) Please note that the following are not necessarily concerns about the tools that have been developed or reported, but are constraints from the perspective of physiological relevance of models.

i) Dendritic ion channels and their properties: Unlike CA1 pyramidal neurons (Magee, J Neuroscience, 1998; Hoffman et al., Nature, 1997; Colbert et al., J Neuroscience 1997; Magee and Johnston, J Physiology, 1995), the dendritic ion channel profiles of DG GC neurons are not well characterized through systematic location-dependent cell-attached recordings. This is important data because channel physiology is not just a function of the main and auxiliary subunits expressed, but is dependent on the relative expression profiles of different subunits, the phosphorylation state of the different residues on each of these subunits, and structural interactions across channels that might alter functionality (Anderson et al., Nat Neuroscience, 2010; Heath et al., J Neuroscience, 2014; An et al., Nature, 2000; Gasparini and Magee, J Physiology, 2002). Several studies that the authors have cited and have used models from also strongly emphasize the critical importance of intracellular milieu in determining the specific physiological properties, which is not directly determinable only from knowledge of the subunits expressed. Additionally it is impossible to assume that the dendritic and somatic channels have the same kinetics and voltage-dependence profiles; several channels show significant gradients in their conductances/kinetics/voltage-dependence and these properties and play important physiological roles in location-dependent input processing (Magee, J Neuroscience, 1998; Hoffman et al., Nature, 1997; Colbert et al., J Neuroscience 1997; Magee and Johnston, J Physiology, 1995; Migliore and Shepherd, Nature Reviews Neuroscience, 2002; Lai and Jan, Nature Reviews Neuroscience, 2006; Narayanan and Johnston, J Neurophysiology, 2012). Therefore, it is important that models also account for these differences and variability in channel properties and location-dependent measurements (Rathour and Narayanan, 2014), rather than assuming that the somatic channel properties (kinetics and voltage-dependence) extend to the dendrites as well.

ii) The authors might want to add a discussion paragraph that expands on details of how their framework will be able to accommodate such gradients in kinetics, voltage-dependence and other properties of channels and receptors, and how their conclusions on cross-morphology robustness would be affected by such gradients. The framework of degeneracy (below) might therefore be an essential one in accounting for variability in gradients of channel conductances and properties towards matching location-dependent physiological measurements and input processing (Rathour and Narayanan, 2014). It also might be appropriate to emphasize the importance of intracellular milieu in determining location-dependent channel properties in different neurons, as the authors are envisaging a more general applicability of their model rather than being focused only on DG neurons.

iii) Degeneracy: The authors lay emphasis on morphological variability, but ignore another important form of cell-to-cell variability in ion channel expression profiles in a location-dependent manner (Marder and Goaillard, 2006; Marder and Taylor, Nat Neuroscience, 2011; Marder, 2011, Rathour and Narayanan, 2014) except for brief references in the discussion! The authors should discuss the implications for variability in ion channels, their properties and location-dependent expression profiles. Importantly, perhaps in the future, the authors could incorporate a stochastic sampling algorithm (Foster et al., 1993) that has been employed across several studies cited above for building a population of heterogenous models that spans both morphological variability (that the authors focus here on) and channel variability. A discussion on this would be helpful, because currently the critical roles of channel variability and degeneracy have been left undiscussed but are too important for the framework that the authors are considering.

iv) Please note that we are not requesting the authors for simulations showing that they could obtain similar physiological outcomes with distinct combinations of morphological and biophysical properties. We just suggest that the authors might want to consider a discussion on these future directions (in terms of building on the basic framework reported here). This is especially important because the equivalence that the authors are drawing for pharmacological and overexpression studies, and the conclusions of the single parameter sensitivity analyses would critically depend on the specific conductance values for each channel in a system that expresses variability and degeneracy (Taylor et al., J Neuroscience, 2011; Rathour and Narayanan, 2014; O'Leary et al., Neuron, 2014). A discussion on such variable dependence within the "Sensitivity analysis reveals critical ion channels in mouse and rat GCs" section or in the Discussion section might be appropriate.

---

## [Author Response]

Essential revisions:In summary, there are three essential revisions that the authors need to consider. An overview of each revision is provided first, and detailed comments from the reviewers to consider for each point are provided after the overview.1) A rewrite of the paper is needed, mostly Results and Discussion (and Materials and methods in coordination) so that the paper is actually a Tools and Resource paper. That is, the paper should mainly be about the tool and its usage, and not mainly about the subsequent model results and interpretations. The authors need be clear about what the tool brings forth relative to what one would need to do otherwise etc. A title change should be considered in light of this.

We extensively rewrote the paper as a Tools and Resource paper. The main text is now focusing on the tool and its usage. We placed the details about model results and interpretations into Appendix 2. We modified the title to emphasize that the paper is introducing a new tool for the computational neuroscience community.

2) Technical aspects for use of the tool for potential users needs to be included. That is, some sort of step by step, tool usage setup, mini tutorials and examples in some way.

We have now included 9 tutorials explaining step by step how to use T2N to run simulations in datasets of reconstructed and synthetic morphologies.

3) In demonstrating the usage of the tool with GC models (e.g., morphological changes, dendritic remodeling in pathology/development of rat/mouse, adult-born neurons), the authors need to be clear about where and how the tool is used in their developed models. That is, the authors need to describe what it is the tool is allowing them to do (in their demonstrated use of it) rather than solely presenting their interpretation of their model results.Please separately present tool usage and model interpretation to avoid confusion, and ensure that the emphasis is on demonstrating the tool usage.Note that while several of the detailed comments below may refer to model interpretation, the authors need to ensure that their revised paper emphasizes and is mainly about the demonstrated tool usage and not about the model results and their interpretation per se. This way, it can be clear that the present paper is a 'Tools and Resource' one.

In the revised paper (both in Results and Discussion), we put emphasis on the tool usage by first mentioning the respective functions of the T2N tool and only then describing their usage in the case of dentate granule cell (GC) modeling. Furthermore, we separated the presentation of the tool usage and model interpretation by putting the details of the GC model and GC simulations into Appendix 2. In this way, we provide all the details, which are of interest for dentate gyrus experts (but not directly relevant for all users of T2N) without keeping them in the main text.

Detailed comments from reviewers to take into consideration when doing Essential Revision 1:a) While the GC modeling work is interesting, I found myself reading it and wondering where the T2N tool fit in. Presumably, the modeling work could've been done without T2N but it would've been alot more complicated, harder and longer? Is it easier to do sensitivity analyses, plot and visualize etc. etc.? Is it easier to test relationships between ion channels and AP features as they do? If so, how is this the case via T2N tool? Is the T2N tool intended for cell types besides GCs? Presumably it is (otherwise it would be extremely limiting). In the outlook, it would seem that it might only be for considering GCs?. Are there any other tools that do what T2N does? What would a user do if T2N did not exist?

The T2N tool is intended also for other cell types besides GCs. We mention this now in the revised manuscript (Discussion section). We have also rewritten the Outlook paragraph to make it clear (subsection “GC modeling and degeneracy”). The reviewer is right that in principle the modeling work could have been done without T2N but T2N made it easier. Indeed, sensitivity analyses, plots, visualizations are much easier to do with T2N than other commonly used software tools. Therefore, we have added a description of strengths and unique features of T2N in the Discussion section.

b) In essence, the authors talk about T2N for one page and then say that "In the following, we show at the example of the dentate GC how to build a robust compartmental model using T2N." But they actually don't explain it via T2N that I could find, and they don't simply build a model – they use it to examine epilepsy, young cells etc.

As mentioned above in our response 3, now we explain first the T2N tools and then we proceed to describe their usage in building the GC model and running GC simulations.

c) The Discussion starts by talking about the model results, and the tool second. This should be reversed and/or the model results should be presented in light of the tool somehow I would think if this is a 'Tools and Resources' paper.

We followed the suggestion of the reviewer and reversed the description of model results and the tool usage in the Discussion. In addition, a large part of the detailed discussion of model results is now in Appendix 2.

d) There is a lot of repetition in Results and Materials and methods regarding the models, but at the same time barely any specifics regarding the tool or how to use the tool itself. This possibly stems from the authors not presenting the work as 'tools and resources'. I appreciate that there is a manual, but there should at least be some overall explanations and mini examples that the user could try/test for the tool itself?

Now we present 9 new tutorials (containing usage examples) in Appendix 1. We have also shortened the Results, and the Materials and methods by moving GC model details into Appendix 2. We have also added the tool description in the new version of the Results.

e) The Discussion is over lengthy and several parts of it are significantly redundant with the Results section. This could be significantly reduced.

We have modified and shortened the Discussion by removing its GC-specific details and by adding paragraphs requested by the reviewers (see below).

f) Focus on the tool: The manuscript is within the "Tools and Resources" section of eLife. The focus of the authors is the technique that they are developing, with DG employed only as an example for the demonstration of the utility of this tool. However, a significant portion of the results and the Discussion sections, including the limitations presented there, are all very specific to the DG, and not with reference to the methodology developed by the authors. The authors should place emphasis of their manuscript on the tool that they are developing to fit into the "tools and resources" section of eLife. Specifically they should focus on the steps involved in the use of the tool, discuss about limitations of the tool (rather than of the DG model) from the perspective of generalizability to modeling any type of neuron (especially focusing on incorporating measurements that are not available in DG neurons; see below), future directions for how the tool could be further developed (say for well-tuned network models or for astrocyte-neuron interactions, etc). It is important to emphasize the novelties of the DG model, but the focus shouldn't stray away from the main focus of the article.

As mentioned above, the focus of the revised manuscript is now on the T2N. In line with the suggestion of the reviewer, we have added a short discussion of T2N limitations and future directions. Furthermore, we describe the generalizability of T2N for modeling other neuron types. The novelty of the dentate GC model is mentioned in the main text of the manuscript but its details are now in Appendix 2.

Detailed comments from reviewers to take into consideration when doing Essential Revision 2:The problems discussed by the authors, that many models behave poorly outside of the scope for which they were created, and that model cells often contain a mix of constraints from very different experimental conditions and animals, combined in sometimes inconsistent ways, are quite real in the world of detailed modeling. The field can and should have higher expectations at this point, and this tool provides a way for us to get there. From the inclusion of variable morphologies to the impressive & precise incorporation of ion channels within the model, this tool shows promise for aiding modelers to tackle these problems in our models. This approach will allow us to more fully harness the resources within online databases such as NeuroMorpho and IonChannelGenealogy. The use of sterotyped scripts will help reduce the burden on the modeler, as will the automatic parallelization.The T2N and TREES tools appear to have an elegant way to organize the code and have MATLAB and NEURON interact. This reviewer sees a lot of potential with these tools. The author was surprised to find that this manuscript did not seem to be written as a Tools & Resources paper. The info as to how to access the code and software is not provided until the Materials and methods. Tutorials are not provided, and there are not many technical, quantitative details about the implementation of the model, the parallelization, or the tuning. The high level of detail and careful attention to constraints that went into developing this model is not fully represented in the Abstract and Introduction.

We now provide the information about the code availability in the Discussion section and subsection “Data sharing”. Tutorials are now described in Appendix 1. We mention in the Abstract that the novel GC model is a highly-detailed model. We added details about parallelization and tuning (see below).

a) Especially given the "Tools and Resources" designation of this manuscript:– More technical details should be provided in the publication – how do you recreate the experimental conditions for your tests? In Results fourth paragraph – what fitting tool or strategy did you use?

The technical details on recreating the experimental conditions and fitting strategy are now provided in Appendix 2.

– Each of the results headings can be thought of as a use case, for which a tutorial would be appropriate (in an appendix or as part of included documentation of T2N)

Done. Thank you for this suggestion.

– Say whether you can use the T2N in a more exploratory manner, for cell types where channel expression data is not known but we have single cell current sweep data?

We have added this information in subsection “T2N facilitates creation of compartmental models with detailed channel composition”.

– Share how flexible is the code, for using the models produced from TREES and T2N in a larger network simulation?

We describe the potential usage of the code for large-scale network simulations in subsection “T2N facilitates the use of real or synthetic morphologies from different species”; “T2N supports prediction of clinically relevant ion channel alterations in multiple neuronal morphologies”; Discussion section paragraph two; subsection “Morphologically robust compartmental modeling” and in the Abstract and provide one tutorial on how to build up networks (Appendix 1–Tutorial 9).

b) The code should be thoroughly tested on more machines (and operating systems) before being highlighted in a publication. I ran into several.

We have now tested the T2N on following machines: Windows Vista 32 Bit, Windows 7 64 Bit, Windows 10 64 Bit, macOS 10.12 (Sierra). We are currently testing T2N on Linux machines.

c) Would like more visibility into the ion channel models, like characterization of the ion channel behavior (activation curves, etc), maybe a table of the ion channel kinetics (time constants, voltage of half act/inact, etc)

We have included these details in the form of graphs of activation and inactivation kinetics, see Appendix 2 Figure 1 and 2.

d) Feels like technical details of the approach are missing. The authors say it's "not a genetic algorithm", and it's "similar to a multiple objective approach", but don't take us through exactly what they've done to tune the parameters. Should also clarify that they found a single solution that works across many morphologies, not a set of solutions with different parameter combinations for each morphology.

We used hand-tuning tofit multiple objectives and datasets, which are summarized in the Table 2. In subsection “Morphologically robust compartmental modeling”, we clarify now that we found a single solution, which works across many morphologies. There, we also mention that we do not exclude that there exists a set of solutions with distinct parameters for different morphologies.

e) Add code to ModelDB, include link in paper with accession #.

We provide the link of the model in the first paragraph of the Discussion section and subsection “Data sharing”. Here is the accession code: 231862. The model is currently private, the access code is: BeiningGCmodel. The model will be made public after acceptance of the paper. We also uploaded a pure NEURON version of the GC model (all morphologies and biophysics but only two protocols implemented) on ModelDB (accession # 231818, currently private, access code: BeiningGCmodel).

Detailed comments from reviewers to take into consideration when doing Essential Revision 3:a) In some parts, the authors seem to be making the point about "the importance of using detailed data-driven ion channel composition, diverse dendritic trees as well as multiple different electrophysiology experiments for tuning and generating compartmental models which would be stable across many morphologies and conditions", and in other parts, as a different approach to more standardized methods with automated parameter fitting as in BBP etc.. Is this meant to be the point of the tool? If so, it does not seem appropriate for a tools and resources paper. Models have different goals and are built to address different questions, and this is true for compartmental ones too I think.

We agree with the reviewer that different models have different goals and address different questions. It is not the point of our paper and tool to question this or to disregard valid predictions and insights of previously published compartmental models or to criticize automated parameter fitting as performed in BBP. Our main point is the emphasis on using many different morphologies in combination with carefully constrained ion channel models. We clarify this now in subsection “Morphologically robust compartmental modeling” and “GC modeling and degeneracy”.

b) Essentially, the authors focus on the model results after using the tool, emphasizing issues regarding GCs that are possible to determine with their tool. This seems somewhat circular to me. That is, they say that "our work might suggest that morphologic robustness arises naturally in models in which parameters have been tuned using multiple different experiments and morphologies. This conclusion is also supported by a comparison of our biophysical model to an earlier widely used GC model which failed to reproduce electrophysiological data…" (see my next comment), but it may be that if some other biological aspect besides morphology or in addition to it was focused on, robustness would've also been observed?

We concur with the reviewer that it has to be determined whether variation of other biological aspects would also lead to robust GC simulation results. We are currently studying this question as part of a new project in our group and we have added this point to the paragraph on degeneracy in subsection “GC modeling and degeneracy”.

c) The authors refer to a classical GC model (Aradi and Holmes). What is meant/intended by classical? That it is the first, considered the best at present for those modeling GCs? Besides needing to explain what they mean by classical, it seems a bit unfair to show it doesn't 'match', as presumably building and using this model when Aradi/Holmes did provided some insight and understanding of GC functioning? (I didn't go back to look at that paper's details).In other words, the authors should present this classical model in a more holistic sense. Presumably they are showing that "most published modelsbehave poorly when used outside of the scope for which they were created." as they state in their Introduction. This may not be a huge problem with the classical model (and others) so long as the model (with its limitations and caveats) was clearly presented at the time, and that some insight/understanding/hypothesis-generation etc. was achieved at the time with the model.

Yes, Aradi and Holmes is currently the most frequently used model of GCs (e.g. Schneider, Bezaire and Soltesz Front Neural Circuits 2012; Yu et al., J Neurophysiol 2013; Liu, Cheng and Lien, 2014; Jedlicka et al., 2015; Platschek et al., 2016). Absolutely, we agree that the model by Aradi and Holmes provided many important insights about GC function and we have added this point in the revised manuscript. However, as also mentioned by the reviewers, now we should try to build models, which perform well “outside of the scope for which they were created” and our “tool provides a way for us to get there.”

d) Morphology of adult-born neurons: To match physiological properties of adult-born neurons the authors have changed channel properties. However, it is well-established that maturing neurons have significantly shorter dendritic arborization and lesser spine density as well (Zhao et al., 2006). As mentioned by the authors, the lesser dendritic extent and the reduced surface area caused by the lack of spines would increase the excitability of the cell as well. The authors have accounted for spine density differences when they assess synaptic integration in the mature and immature GCs. But, why did the authors not consider matching morphological profiles of adult-born neurons at various ages to understand the physiology of adult-born neurons?

The reviewer is right that we have modeled (implicitly) the lower spine density in young abGCs. We have not studied morphological differences of abGCs since we focused on 4 weeks old abGCs that don’t display any further significant alterations in their dendritic morphology (see our dendrite analyses in Beining, Jungenitz et al., 2016). It would be certainly insightful to model also younger abGCs with reduced dendrite arborization and a different ion channel composition. However, since we used electrophysiology data from 4 week old cells (Mongiat et al., 2009) because these cells exhibit most interesting functional differences (e.g. higher excitability and synaptic plasticity), we restricted our modeling to this particular cell age group.

Given the several differences between adult-born and mature neurons in the DG, that might have been more appropriate within the framework that the authors are proposing rather than assuming that all excitability changes are mediated by changes in ion channel densities. I believe that performing these additional set of experiments with immature morphologies (which is certainly within the capabilities of the framework proposed here) would further emphasize the utility of the tool that the authors are reporting here, especially with reference to the scenario where different statistics of morphology are observed with certain physiological/pathological conditions. In this case, the authors might also want to discuss if their framework would be capable of maintaining structural integrity of real or synthetic morphologies as they mature (Narayanan and Chattarji, 2010; Dhupia et al., 2015; Bozelos et al., 2016) so as to enable causal links between morphological characteristics and physiological measurements.

Due to the unavailability of raw electrophysiological traces (see our previous response), we can now only provide a solid model for 4 weeks old abGCs but we would love to develop well constrained models for other cell age stages (e.g. based on data from published papers or raw traces from the Schinder group). However, we expect that this would require a lot more time and it would probably deserve a separate publication since now the revised manuscript is more about T2N than about the GC models. However, we mention the future development of models for younger abGCs at different ages in subsection “Predictions of the GC model”. Thank you for mentioning the issue and the publications on the relationship between morphology and function. We have incorporated them in the revised manuscript.

e) Please note that the following are not necessarily concerns about the tools that have been developed or reported, but are constraints from the perspective of physiological relevance of models.i) Dendritic ion channels and their properties: Unlike CA1 pyramidal neurons (Magee, J Neuroscience, 1998; Hoffman et al., Nature, 1997; Colbert et al., J Neuroscience 1997; Magee and Johnston, J Physiology, 1995), the dendritic ion channel profiles of DG GC neurons are not well characterized through systematic location-dependent cell-attached recordings. This is important data because channel physiology is not just a function of the main and auxiliary subunits expressed, but is dependent on the relative expression profiles of different subunits, the phosphorylation state of the different residues on each of these subunits, and structural interactions across channels that might alter functionality (Anderson et al., Nat Neuroscience, 2010; Heath et al., J Neuroscience, 2014; An et al., Nature, 2000; Gasparini and Magee, J Physiology, 2002). Several studies that the authors have cited and have used models from also strongly emphasize the critical importance of intracellular milieu in determining the specific physiological properties, which is not directly determinable only from knowledge of the subunits expressed. Additionally it is impossible to assume that the dendritic and somatic channels have the same kinetics and voltage-dependence profiles; several channels show significant gradients in their conductances/kinetics/voltage-dependence and these properties and play important physiological roles in location-dependent input processing (Magee, J Neuroscience, 1998; Hoffman et al., Nature, 1997; Colbert et al., J Neuroscience 1997; Magee and Johnston, J Physiology, 1995; Migliore and Shepherd, Nature Reviews Neuroscience, 2002; Lai and Jan, Nature Reviews Neuroscience, 2006; Narayanan and Johnston, J Neurophysiology, 2012). Therefore, it is important that models also account for these differences and variability in channel properties and location-dependent measurements (Rathour and Narayanan, PNAS, 2014), rather than assuming that the somatic channel properties (kinetics and voltage-dependence) extend to the dendrites as well.

We share the concern of this reviewer about the complexities of channel modeling. We have added a brief note on these issues in Appendix 2 (subsection “Predictions of the GC model”), which discusses the details of GC models. We cite also the Rathour and Narayanan paper since it is relevant in this context.

ii) The authors might want to add a discussion paragraph that expands on details of how their framework will be able to accommodate such gradients in kinetics, voltage-dependence and other properties of channels and receptors, and how their conclusions on cross-morphology robustness would be affected by such gradients. The framework of degeneracy (below) might therefore be an essential one in accounting for variability in gradients of channel conductances and properties towards matching location-dependent physiological measurements and input processing (Rathour and Narayanan, PNAS, 2014). It also might be appropriate to emphasize the importance of intracellular milieu in determining location-dependent channel properties in different neurons, as the authors are envisaging a more general applicability of their model rather than being focused only on DG neurons.

We followed the suggestion of the reviewer. We added a brief paragraph addressing these issues (see subsection “GC modeling and degeneracy” and in Appendix 2).

iii) Degeneracy: The authors lay emphasis on morphological variability, but ignore another important form of cell-to-cell variability in ion channel expression profiles in a location-dependent manner (Marder and Goaillard, Nat Rev Neuroscience, 2006; Marder and Taylor, Nat Neuroscience, 2011; Marder, PNAS, 2011, Rathour and Narayanan, PNAS, 2014) except for brief references in the discussion! The authors should discuss the implications for variability in ion channels, their properties and location-dependent expression profiles. Importantly, perhaps in the future, the authors could incorporate a stochastic sampling algorithm (Foster et al., 1993) that has been employed across several studies cited above for building a population of heterogenous models that spans both morphological variability (that the authors focus here on) and channel variability. A discussion on this would be helpful, because currently the critical roles of channel variability and degeneracy have been left undiscussed but are too important for the framework that the authors are considering.

We agree with the reviewer that the relationship between morphological and channel variability in the context of degeneracy is an extremely important topic deserving a lot of attention in the field. There were hints about degeneracy and cell-to-cell variability in ion channel expression in the submitted manuscript but now we expanded the discussion of degeneracy – see subsection “GC modeling and degeneracy” and, accordingly, also included more citations. In addition, we have incorporated the comments of the reviewer on stochastic sampling algorithm into the revised manuscript (subsection “T2N limitations and future directions” and “GC modeling and degeneracy”).

iv) Please note that we are not requesting the authors for simulations showing that they could obtain similar physiological outcomes with distinct combinations of morphological and biophysical properties. We just suggest that the authors might want to consider a discussion on these future directions (in terms of building on the basic framework reported here). This is especially important because the equivalence that the authors are drawing for pharmacological and overexpression studies, and the conclusions of the single parameter sensitivity analyses would critically depend on the specific conductance values for each channel in a system that expresses variability and degeneracy (Taylor et al., J Neuroscience, 2011; Rathour and Narayanan, 2014; O'Leary et al., Neuron, 2014). A discussion on such variable dependence within the "Sensitivity analysis reveals critical ion channels in mouse and rat GCs" section or in the Discussion section might be appropriate.

As we described above, we mention now in the revised paper that T2N can be used in the future to study degeneracy by searching for distinct combinations of morphological and biophysical properties to achieve similar physiological outcomes. Moreover, now we discuss the results of our single parameter sensitivity analysis in this context (subsection “Example of sensitivity analysis performed with T2N revealing critical ion channels in mature mouse and rat GCs” and Appendix 2).